**EMBO** *reports*

# YAP1 and QSER1 are key modulators of embryonic signaling pathways in the mammalian epiblast

Elizabeth Abraham [1], Thomas Roule [2], Aidan Douglas[1], Emily Megill[1], Olivia M Pericak[1,3], Jordan E Howe [1], Carmen Choya-Foces [1], Joanne F Garbincius [1], Henry M Cohen [1], Paula Roig-Flórez[1], Mikel Zubillaga[1], Mark D Andrake [4], Seonhee Kim [3], John W Elrod [1], Naiara Akizu [2] & Conchi Estaras [1,5✉]

## Abstract

**YAP1 signaling is essential for development but its specific roles in early embryogenesis remain poorly understood. To shed light on this, we analyze YAP1's role in regulating the pluripotency of the mammalian epiblast, using scRNAseq approaches. Conditional deletion of *Yap1* in the mouse epiblast (*Sox2*-Cre) alters the expression of signaling genes, including *Nodal*, *Wnt3*, and *Fgf8*. Accordingly, *Yap1* loss leads to enhanced differentiation of the epiblast toward primitive streak lineages, as evidenced by the upregulation of T/*Brachyury* and *Eomes* genes. A proximity labeling assay in human pluripotent stem cells, followed by biochemical assays and molecular modeling predictions, reveals that YAP1 cooperates with QSER1 protein to regulate lineage genes. Our analysis shows that YAP1:TEAD4 enhancers recruit QSER1 to prevent RNA Polymerase II recruitment. QSER1 depletion, similar to YAP1, increases NODAL gene expression and leads to hyperactive NODAL signaling during human embryonic stem cells differentiation. Overall, our findings define a role of YAP1 in the epiblast in vivo and uncover an interplay with QSER1 controlling the activity of developmental signaling pathways in pluripotent cells.**

**Keywords** Pluripotency; Epiblast; YAP1; QSER1; Nodal Signaling
**Subject Categories** Development; Signal Transduction; Stem Cells & Regenerative Medicine

## Introduction

The main transcriptional effectors of the Hippo signaling pathway are YAP1 and its homolog TAZ. When YAP1 and TAZ are phosphorylated by Hippo kinases, they are degraded in the cytoplasm. When dephosphorylated, YAP1 and TAZ are stabilized and translocated into the nucleus where they interact with TEAD1-4 DNA binding proteins to induce the expression of genes that promote proliferation and growth (Wu and Guan, 2021). Thus, loss-of-function mutations in the core kinases or overexpression of YAP1 and TAZ upregulates the expression of genes that induce cell proliferation, cell survival, and tissue growth. Notably, TAZ knockout (KO) mice complete embryonic development, whereas YAP1 KO results in early embryonic lethality, underscoring a more prominent role of YAP1 in embryogenesis that TAZ cannot compensate (Hossain et al, 2007; Morin-Kensicki et al, 2006). Overall, the results of these experimental approaches established the Hippo:YAP1 signaling pathway as a universal regulator of organ size (Zhong et al, 2024).

However, emerging lines of research underscored that the physiological roles of YAP1 may be unrelated to cell proliferation or growth in some developmental contexts. During liver development, YAP1 overexpression causes overgrowth (Dong et al, 2007), but conditional deletion of YAP1/TAZ in the liver lineage does not prevent this organ from reaching its normal size (Kowalczyk et al, 2022; Lu et al, 2018). Instead, endogenous YAP1 activity is required for cholangiocyte differentiation (Kowalczyk et al, 2022). Similarly, *Yap1* KO mouse embryos can initiate gastrulation and develop until ~E8 without noticeable changes in size, suggesting that YAP1 is not required for embryonic growth or proliferation of totipotent, multipotent, and pluripotent stem cells (Morin-Kensicki et al, 2006). Instead, during this embryonic period, YAP1 regulates zygotic genome activation in totipotent cells and specification of trophectodermal cells in multipotent cells (Nishioka et al, 2009; Yu et al, 2016). However, whether YAP1 regulates the differentiation of pluripotent stem cells in vivo, is still unknown.

The natural progression of the pluripotent cells of the epiblast, triggers gastrulation (Ghimire et al, 2021). During gastrulation, the pluripotent epiblast differentiates into each of the three germ-layer lineages, breaking the symmetry of the embryo and establishing the body plan. The formation of the primitive streak (PS) in the posterior part of the epiblasts marks the beginning of gastrulation. It gives rise to mesoderm and endoderm layers, while the anterior

[1]Department of Cardiovascular Sciences, Aging + Cardiovascular Discovery Center, Lewis Katz School of Medicine, Temple University, Philadelphia, PA 19140, USA. [2]Raymond G. Perelman Center for Cellular and Molecular Therapeutics, The Children's Hospital of Philadelphia, Philadelphia, PA 19104, USA. [3]Department of Neural Sciences, Center for Neural Development and Repair, Temple University, Lewis Katz School of Medicine, Philadelphia, PA 19140, USA. [4]Molecular Modeling Facility, Program in Cancer Signaling and Microenvironment, Fox Chase Cancer Center, Philadelphia, PA 19111, USA. [5]Cancer Epigenetics Institute, Fox Chase Cancer Center, Philadelphia, PA 19111, USA.
✉E-mail: conchi.estaras@temple.edu

epiblast commits to ectodermal lineages (Bardot and Hadjantonakis, 2020; Ghimire et al, 2021; Wang et al, 2023). The activity of key signaling pathways, including WNT, NODAL and BMP, enables selective activation of gene-expression programs necessary for germ-layer specification (Morgani and Hadjantonakis, 2020). Before gastrulation, NODAL signaling maintains pluripotency of the epiblast (Camus et al, 2006; Mesnard et al, 2006; Vallier et al, 2009; Xiao et al, 2006) while WNT signaling is initially low (Liu et al, 1999; Yoon et al, 2015). Upon the onset of gastrulation, upstream signals from extraembryonic tissues—including BMP4—stimulate *Wnt3* expression in adjacent epiblast cells (Bardot and Hadjantonakis, 2020). WNT signaling activation stabilizes β-CATENIN, which partners with Nodal transcriptional effectors SMAD2/3 to drive PS gene expression, including *Eomes* and *T/Brachyury*, in the posterior part of the epiblast (Funa et al, 2015; Estarás et al, 2015; Gadue et al, 2006). Conversely, in the anterior epiblast, WNT and NODAL activity are actively suppressed to permit ectodermal differentiation, an inhibition largely mediated by WNT and NODAL antagonists secreted by extraembryonic tissues (Arkell et al, 2013; Morgani and Hadjantonakis, 2020; Robertson, 2014). Thus, as development progresses, NODAL signaling becomes progressively confined to the posterior epiblast to induce the PS in concert with WNT (Funa et al, 2015; Estarás et al, 2015; Gadue et al, 2006), enabling ectoderm commitment in the anterior region. Overall, precise regulation of WNT and NODAL signaling is necessary to ensure correct lineage differentiation and anterior–posterior patterning during gastrulation.

Using human pluripotent stem cell (hPSC) models, we and others have shown that YAP1 regulates germ-layer differentiation by repressing the activity of WNT and NODAL signaling (Beyer et al, 2013; Hsu et al, 2018; Stronati et al, 2022). Furthermore, our recent studies revealed that *Yap1*-depleted embryos have altered proportion of germ-layer derivatives, compared to controls (Abraham et al, 2025), suggesting that *Yap1* loss disrupts epiblast differentiation in vivo. Thus, in the present study we investigate the molecular mechanisms by which YAP1 regulates cell-fate decisions of the pluripotent epiblast. Single-cell transcriptomic profiling of embryos at E7 revealed that conditional YAP1 deletion led to upregulation of key gastrulation signaling genes, including *Nodal*, *Wnt3*, and *Fgf8* in the epiblast. Furthermore, E7 mutant embryos display an increased proportion of cells in the PS, compared to controls, and increased expression of PS genes, including *T/Bra* and *Eomes*. We then conducted a proximity-based proteomic screen in hESCs, which identified QSER1 as a high-confidence nuclear interactor. Accordingly, by ChIP-seq analysis we found that QSER1 co-localizes with YAP1 and TEAD4 on a subset of developmental enhancers, including those regulating *NODAL* and *SMAD2*. Functional and structural analyses revealed that YAP1 serves as a bridge between QSER1 and TEAD4, facilitating QSER1 recruitment and stabilizing a TEAD-YAP1-QSER1 trimeric complex in the chromatin. Finally, we found that loss of QSER1 on YAP1:TEAD4 enhancers increased RNA polymerase II occupancy on these sites and phenocopies YAP inhibition, leading to ectopic NODAL expression and mesoderm induction during hESC differentiation. Altogether, our findings define a role in YAP regulating epiblast differentiation in vivo and reveal a previously unrecognized YAP1–QSER1 module that functions to fine-tune gene expression during lineage differentiation.

# Results

## Conditional deletion of *Yap1* (*Yap1* cKO) in the epiblast and analysis of E7 embryos

To investigate the functional role of YAP1 in epiblast differentiation in vivo, we conditionally deleted *Yap1* in mouse embryos using a Sox2-Cre driver (Hayashi et al, 2002). *Sox2* gene expression initiates at ~E5.5 in the epiblast, and *Cre*-mediated recombination occurs throughout the epiblast and its derivatives, excluding earlier arising extraembryonic lineages (Hayashi et al, 2002). We performed single-cell RNA-seq analysis on freshly isolated embryos at E7 prior to genotyping of Cre, flox, and sex alleles, using a methodology optimized in our lab (Abraham et al, 2024) (Fig. EV1A–D). We sequenced a pool of three *Yap1* cKO (*Yap1*^Flox/Flox^: Sox-Cre) and four heterozygous control littermates (*Yap1*^Flox/+^: Sox2-Cre) (Figs. 1A and EV1C). A total of 1829 control cells and 1897 *Yap1* cKO cells were initially sequenced. After excluding cells that did not meet specific data quality standards, these numbers were reduced to 788 and 862, respectively (see Methods for details).

Sequenced cells were manually annotated based on the expression of known markers (Argelaguet et al, 2019; Mittnenzweig et al, 2021; Pijuan-Sala et al, 2019). We identified six embryonic (epiblast, primitive streak (PS), nascent mesoderm, cardiac mesoderm, blood progenitor, and endoderm) and three extraembryonic populations (extraembryonic endoderm, anterior visceral endoderm (AVE), and extraembryonic ectoderm) in the E7 embryos, based on the expression of known markers (Fig. 1B and see Methods for details). Importantly, *Yap1* cKO embryos showed a significant reduction in Yap1 mRNA levels across embryonic clusters compared to controls (Figs. 1E and EV1E). Consistent with reduced YAP1 activity in the embryos, we observed typical YAP1 target genes and Hippo regulators transcriptionally affected. These include *Ccn2* (also known as *Ctgf*), which was significantly downregulated in the primitive streak and nascent mesoderm, and *Ccn1* (also known as *Cyr61*), which was reduced in the epiblast. Regulators of the Hippo pathway, including *Amotl2* (Hirate et al, 2013), *Ptpn14* (Blakely et al, 2024), and *Wwc2* (Hermann et al, 2021), were also downregulated across multiple lineages, such as the epiblast, primitive streak, and cardiac mesoderm (Dataset EV1 and Fig. EV1F). These results support the effective disruption of YAP1 activity in the cKO embryo. Importantly, the expression of the *Wwtr1* gene, which codifies for the YAP1 homolog TAZ, was not significantly affected by *Yap1* deletion (Fig. EV1E). As anticipated, *Yap1* expression in extraembryonic clusters remained similar between *Yap1* cKO embryos and controls (Fig. EV1E). Cell cycle analysis revealed no significant differences in the *Yap1* cKO compared to control embryos (Fig. EV1G). In agreement, control and *Yap1* cKO E7 embryos display similar cell count and epiblast perimeter (Fig. EV1H). We conclude that our experimental approach effectively captured the major cell populations of E7 embryos, achieved efficient conditional *Yap1* deletion, and showed that *Yap1* loss did not alter cell cycle dynamics compared to controls.

## YAP1 represses Nodal and Wnt signaling pathway genes in the epiblast

Differential expressed genes (DEGs) analysis between *Yap1* cKO and control embryos revealed varying numbers of genes impacted by *Yap1* deletion across embryonic clusters (Fig. 1C and Dataset EV1). Within the embryonic populations, the epiblast cluster

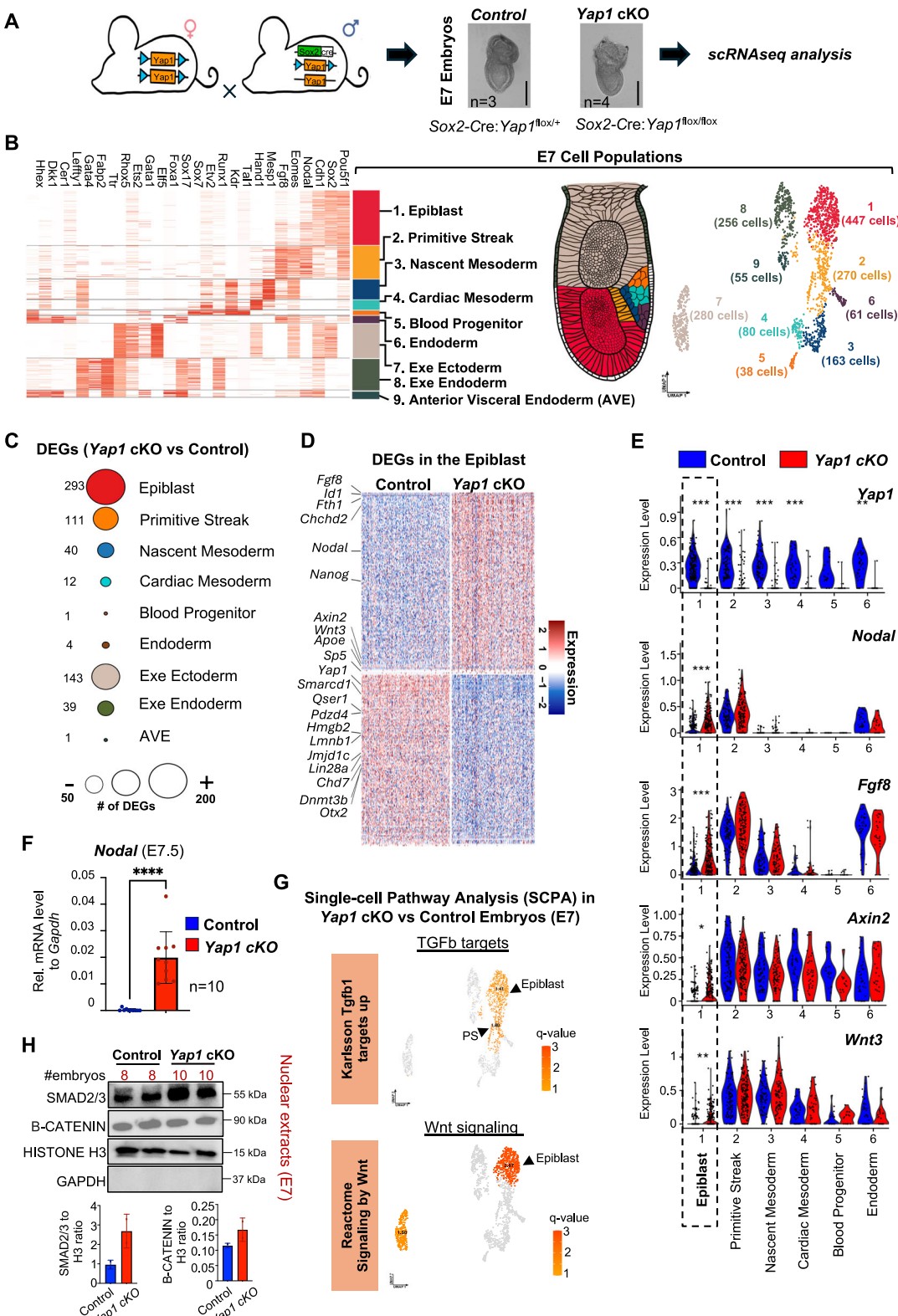

exhibited the highest number of DEGs (293; 151 upregulated, 142 downregulated, adj. $p < 0.05$), followed by the PS (111) and the nascent mesoderm (40) (Fig. 1C). These findings align with previous studies showing a significant role of YAP1 in the

transcriptome of pluripotent stem cells in vitro (Beyer et al, 2013; Estarás et al, 2015; Estarás et al, 2017; Hsu et al, 2018). Among the downregulated genes in the epiblast, we identified key pluripotent regulators such as *Jmjd1c* (Xiao et al, 2017), *Dnmt3b* (Dai et al,

◀

**Figure 1.  Nodal and Wnt signaling genes are activated in the epiblast of *Yap1 cKO* embryos compared to controls.**

(A) Mice scheme shows the breeding strategy to obtain embryos with conditional deletion of *Yap1* in the epiblast (Sox2cre). Blue arrowheads indicate LoxP alleles. E7 heterozygous control (Sox2cre:YAPflox/+) and *Yap1* cKO (Sox2cre:Yap1flox/flox) embryos were processed for scRNAseq analysis. Bright-field images show representative embryos of the indicated genotype. The number of embryos processed for sequencing is indicated. Scale bar 250 µm. (B) Heatmap showing expression of lineage markers used to annotate cell populations in the E7 scRNAseq datasets. On the right, a schematic of an E7 mouse gastrula and a UMAP of E7 scRNAseq showing the detected cell populations with the number of cells in parentheses, color-coded to match the heatmap. (C) Dot plot depicts the number of differentially expressed genes (DEGs) in each cluster, with the exact count indicated to the left of each dot. Note that the epiblast cluster contains the highest number of DEGs (abs(Log2FC)>0.25, adj. *p*-value < 0.05). (D) Heatmap shows DEGs in the epiblast of *Yap1* cKO versus control embryos. Relevant genes for pluripotency and differentiation are shown. (E) Violin plots shows scRNAseq expression levels of indicated genes across clusters in control and *Yap1* cKO. The dotted box highlights the epiblast cluster. *Yap1* expression is significantly reduced in *Yap1* cKO cells across multiple lineages, including epiblast (adjusted $p = 7.8 \times 10^{-58}$), primitive streak ($1.2 \times 10^{-20}$), nascent mesoderm ($4.2 \times 10^{-17}$), cardiac mesoderm ($1.2 \times 10^{-5}$), and endoderm ($3.8 \times 10^{-4}$). Epiblast expression of *Nodal* (adjusted $p = 1.0 \times 10^{-5}$), *Fgf8* ($7.1 \times 10^{-9}$), *Axin2* ($4 \times 10^{-3}$), and *Wnt3* ($1.5 \times 10^{-4}$) is significantly altered in *Yap1* cKO embryos. Adjusted *p*-values were calculated using a Wilcoxon rank-sum test with Benjamini–Hochberg correction (*$p < 0.05$, **$p < 0.001$, ***$p < 0.0001$). Each dot represents a single cell from E7 scRNAseq data. (F) Graph shows RT-qPCR analysis of the *Nodal* gene in E7.5 control and *Yap1* cKO embryos ($n = 10$). Data presented as mean ± SEM. Statistical analysis: Student's t-test, ****$p < 0.0001$. (G) Single-cell pathway enrichment analysis (SCPA) was performed on DEGs of *Yap1* cKO compared to control. The UMAP plot shows the enrichment of two terms related to the Nodal/TGFb and Wnt pathway. Significant q-values (>1.4) are displayed in orange with the names of populations. The complete list of Wnt and TGFb terms enriched are shown in Fig. EV1I. (H) Western blot of nuclear extracts of E7 control and *Yap1* cKO embryos. The number of embryos pooled per lane is indicated above each lane, along with the markers analyzed. Error bars represent mean ± SD. Uncropped blots are found in Fig. EV1J.

2016), and *Qser1* (Dixon et al, 2021). Conversely, upregulated genes included the critical signaling pathway genes *Nodal*, *Wnt3*, and *Fgf8* (Fig. 1D–F). Along with Wnt and Nodal, Fgf signaling also promotes epiblast differentiation toward PS cell-fates (Conlon et al, 1994; Guo and Li, 2007; Liu et al, 1999; Sun et al, 1999). Consistent with this, we observed upregulation of downstream targets of these pathways, including *Axin2*, *Id1*, and *Nanog*. *Axin2* is a typical β-CATENIN target gene, thought to fine-tune Wnt signaling activity in the epiblast (Preprint: Hernández-Martínez et al, 2024). *Nanog* and *Id1* are direct targets of SMAD2/3 involved in the repressing of neuroectodermal fates (Vallier et al, 2009; Wang et al, 2012; Xu et al, 2008) and regulating epiblast differentiation timing and cardiac mesoderm differentiation (Cunningham et al, 2017; Malaguti et al, 2019; Yu et al, 2018), respectively. Furthermore, single-cell Pathway Analysis (SCPA) retrieved TGFβ/NODAL and WNT signaling-related terms specifically enriched in the epiblast cluster of the *Yap1* cKO embryos, compared to controls (Figs. 1G and EV1I). Accordingly, we detected an accumulation of nuclear SMAD2/3, the transcriptional effectors of Nodal signaling, in nuclear extracts of mutant embryos, compared to controls, further suggesting an increase in the activity of the NODAL pathway upon *Yap1* loss (Figs. 1H and EV1J (nuclear) and Fig. EV1K (total extracts)). However, we failed to detect nuclear accumulation of the WNT effector β-CATENIN in the mutant embryos, compared to controls (Figs. 1H and EV1J (nuclear) and Fig. EV1K (total extracts)), which could be due to a milder activation of the pathway combined with the dynamic nature of β-CATENIN nuclear localization, as previously noted by others (Martyn et al, 2019).

Overall, these data show that conditional deletion of *Yap1* in the epiblast affects the expression of genes important for pluripotency and differentiation. Furthermore, we conclude that the predominant role of YAP1 in the epiblast is to repress the expression of signaling genes regulating PS differentiation.

## Conditional YAP1 deletion leads to an expanded primitive streak (PS) domain at E7

To test how YAP1 deletion in the epiblast affects E7 populations, we analyzed the proportion of cells in each cluster in *Yap1* cKO embryos and controls. Among the embryonic clusters, the PS was

the only cluster with significant changes. *Yap1* cKO embryos displayed a significant increase in the number of PS cells, compared to controls (Fig. 2A). Two of the extraembryonic clusters also displayed significant differences in the number of cells (Fig. EV2A). However, the extraembryonic clusters do not express the *cre* allele and preserve YAP1 expression (see Fig. EV1E). Therefore these differences may be due to a non-autonomous effect of *Yap1* deletion in the epiblast.

To visualize the PS domain in gastrulating embryos, we analyzed the expression pattern of *T/Brachyury*, a general marker of the PS (Bulger et al, 2024; Rivera-Pérez and Magnuson, 2005; Stott et al, 1993), using whole-mount immunostaining. At the onset of gastrulation, *T/Brachyury* is initially expressed in the proximal posterior epiblast and expands both anteriorly and distally as development progresses (Rivera-Pérez and Magnuson, 2005). In *Yap1* cKO embryos, we observed a significant increase in *T/Brachyury* expression levels, along with a distal expansion of its expression domain compared to control embryos (Figs. 2B–D and EV2B). Therefore, we conclude that YAP1 deletion leads to enhanced PS differentiation of the epiblast.

## YAP1 deficiency disrupts embryo patterning

Premature or excessive differentiation toward PS may result in severe consequences for gastrulation patterning. For instance, it may lead to an imbalance in the proportion of populations derived from the PS and/or a depletion in the pool of epiblast cells left for ectodermal commitment.

Indeed, older *Yap1* cKO embryos display severely under-developed headfolds and neural plate structures (Fig. EV2C,D) that are not compatible with life beyond E8.5 (Abraham et al, 2025; Morin-Kensicki et al, 2006). These abnormalities are consistent with impaired ectoderm development and disruption of anterior structures (Stiles and Jernigan, 2010).

To further determine the developmental consequence of epiblast *Yap1* deletion, we analyzed the proportion of cell populations in the scRNAseq of E7.75 embryos we previously generated (Abraham et al, 2025). Remarkably, a comparison of epiblast-derived populations between control and *Yap1* cKO at E7.75 revealed an increase in the proportion of hematopoietic stem cell and endoderm populations alongside a significant reduction in

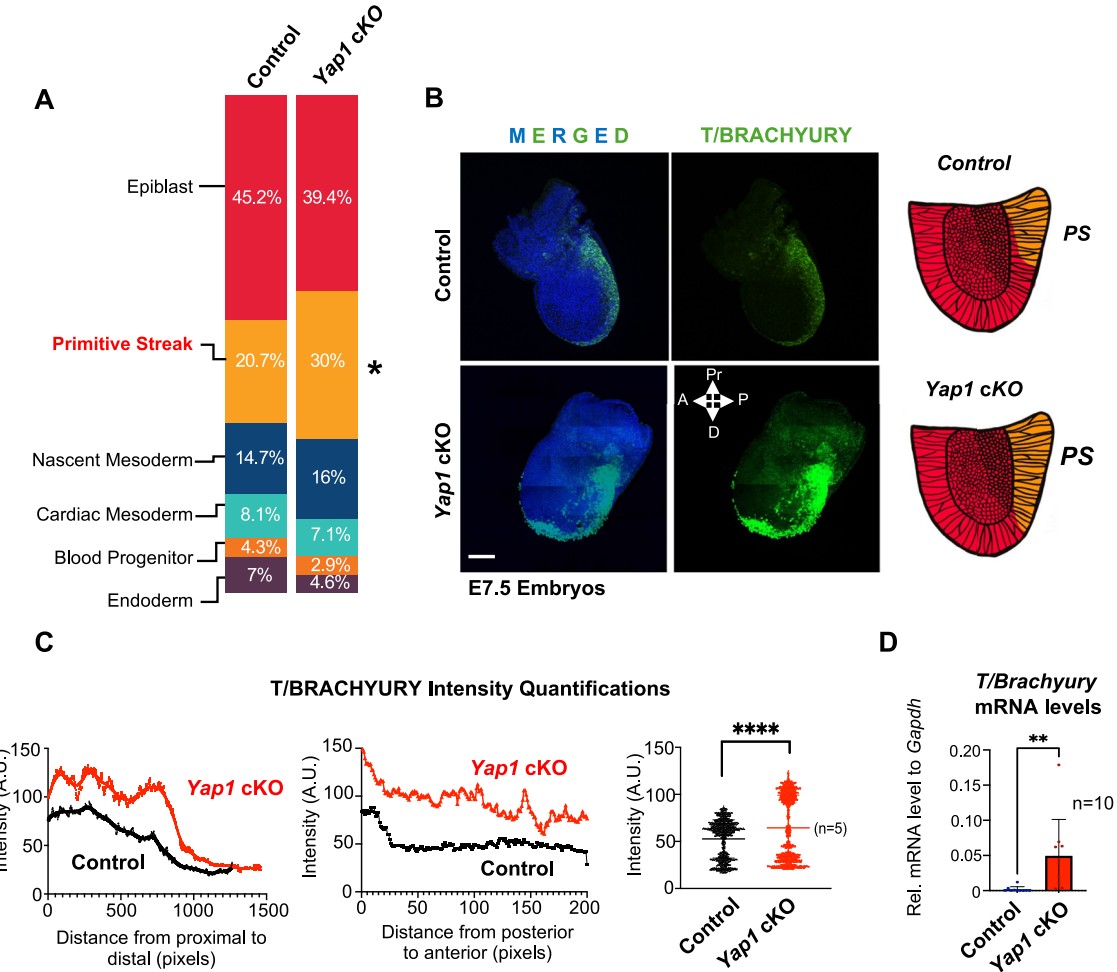

**Figure 2. Yap1 loss in the epiblast increases differentiation toward the primitive streak (PS).**

(A) Bar graph showing the percentage of embryonic cell populations detected by scRNA-seq analysis in control and *Yap1* cKO embryos. Statistical significance was assessed using the Chi-square test (*$p < 0.05$). Only embryonic populations are shown. See Fig. EV2A for analysis including all populations. (B) Representative images of whole-mount immunostaining for the PS marker BRACHYURY (T/BRA) (green) in E7.5 control and *Yap1* cKO embryos. DAPI (blue) marks nuclei. On the right, a scheme summarizing results; compared to controls, *Yap1* cKO embryos have expanded the PS domain. Scale bar 250 µm Pr: proximal, A: anterior, P: posterior, D: Distal. (C) Graphs show quantifications of T/BRA signal intensity along the proximal to distal axis of the embryo (left), the posterior to anterior axis (middle), and the overall intensity of the immunostaining (right). An in-house developed Matlab script was applied to quantify fluorescence. The experiment was replicated with three separate litters ($n = 3$). Data are presented as mean ± SEM. Statistical analysis: Student's t-test, ****$p < 0.0001$. (D) RT-qPCR of *T/Bra* in E7.5 control and *Yap1* cKO embryos ($n = 10$). Data are presented as mean ± SEM. Statistical analysis: Student's t-test, **$p = 0.0097$.

neural and surface ectoderm populations in *Yap1* cKO embryos (Fig. EV2E). Other PS populations, including presomitic and somitic mesoderm, were also underrepresented in the *Yap1* cKO embryos (Fig. EV2E). To confirm these results, we performed qPCR and Western blot analyses of the anterior primitive streak marker *Eomes* (Arnold et al, 2008), endoderm marker *Foxa2* (Burtscher and Lickert, 2009) and neuroectoderm marker *Otx2* (Acampora et al, 1995) in E7.75 embryos. Consistent with the morphological analysis and scRNAseq data, *Eomes* and *Foxa2* were upregulated and *Otx2* downregulated in *Yap1* cKO (Fig. EV2F,G).

Overall, we conclude that conditional *Yap1* deletion leads to imbalanced differentiation of the epiblast. Specifically, *Yap1* loss in the epiblast promotes the expansion of anterior primitive streak derivatives while compromising neural committed cells. This shift

is consistent with increased NODAL and WNT signaling activities in the mutant embryos, as suggested by our transcriptomic analysis of the epiblast at E7.

## Analysis of YAP1 nuclear interactors in human pluripotent stem cells

In many cell contexts, YAP1 regulates transcription of genes controlling proliferation and cell growth, such as *Myc* or *Ctgf* (LeBlanc et al, 2021). However, in the mouse epiblast (Fig. 1) and in hESCs (Estarás et al, 2017; Stronati et al, 2022), YAP1 also regulates genes linked to differentiation, like *Wnt3* and *Nodal*. This suggests the existence of a pluripotent-specific YAP1-transcriptional network. To gain insight into this, we performed a proximity biotinylation assay to identify YAP1-associated nuclear proteins

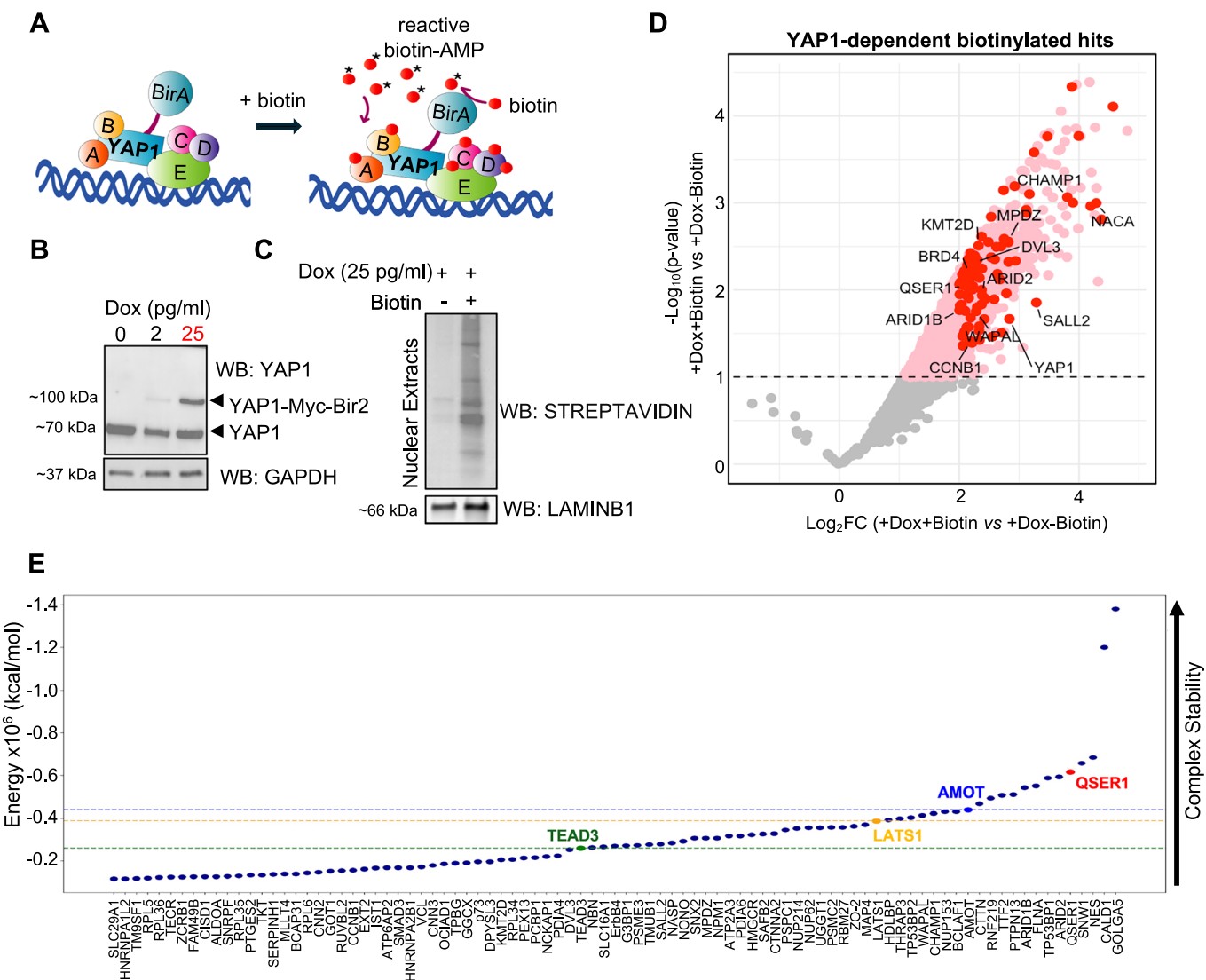

**Figure 3. BioID2 assay identifies QSER1 in the proximity network of YAP1 in hESCs.**

(A) Scheme of BioID2 assay to identify potential nuclear interactors of YAP1 in hESCs. (B) A doxycycline-inducible YAP1-Myc-Bir2 clonal cell line was generated in H1 hESCs. Western blot of YAP1 shows expression of construct following the indicated treatments. 25 pg/ml of Doxycycline (Dox) was used for experiments (highlighted in red). Note that the expression of the YAP1-Myc-Bir2 construct at 25 pg/ml (~90 kDa band) is similar to the expression of endogenous YAP1 (~70 kDa band). GAPDH was used as a loading control. (C) Western blot of YAP1-Myc-Bir2 against Streptavidin shows biotinylated nuclear proteins in the presence and absence of biotin, as indicated. LAMINB1 was used as a loading control of nuclear extracts. (D) Volcano plot showing protein hits detected in the BioID2 assay. Gray dots indicate proteins not significantly enriched in +Biotin+Dox vs. −Biotin+Dox condition. Pink dots represent proteins significantly enriched (Log2FC > 2 and adjusted $p$-value < 0.05) in the +Biotin+Dox vs. −Biotin+Dox comparison, totaling 311 hits. A secondary filter was applied to the 311 hits: red dots highlight proteins also enriched (Log2FC > 1.5) in +Biotin+Dox compared to both +Biotin−Dox and −Biotin−Dox conditions. Only proteins with LFQ values in at least two of three replicates were included in the analysis. The name of some hits is shown; the full list is provided in (E). (E) Computational protein docking analysis of YAP1 with 83 candidate interactors identified by BioID2. The plot displays the predicted potential energies of YAP1 complexes with each candidate, which serve as a proxy for the stability of the protein complex. Energies for three known YAP1 interactors (TEAD3, LATS1, and AMOT) are included as positive controls. The QSER1–YAP1 complex is highlighted and ranks as the fifth most stable predicted interaction.

in hESCs. Although key differences exist between mouse epiblast and human hESCs (Ghimire et al, 2021; Handford et al, 2024; Rossant and Tam, 2022), they share important functional similarities, including their pluripotent state and capacity to give rise to all three germ layers (Brons et al, 2007; Warrier et al, 2017). Accordingly, the phenotype of YAP1 deletion in hESCs (Stronati et al, 2022) closely mirrors the phenotype of *Yap1* cKO in the epiblast (Figs. 1 and 2).

We carried out a proximity biotinylation (BioID2 (Kim et al, 2016)) assay on nuclear extracts of hESCs using YAP1 as bait (Fig. 3A). For this, we created an H1 inducible hESC clonal line that expresses YAP1 fused to Bir2 under the control of a Doxycycline responsive promoter. Doxycycline and biotin concentrations were optimized to observe a stable nuclear YAP1-MYC-BIR2 expression (Figs. 3B and EV3A) and effective biotinylation of nuclear proteins, respectively (Figs. 3C and EV3B). Furthermore, we confirmed that

the YAP1-MYC-BIR2 fusion protein retains the ability to interact with TEAD4, by co-immunoprecipitation experiments (Fig. EV3C) and ChIP-qPCR approaches (Fig. EV3D), to further confirm the functionality of our YAP1-MYC-BIR2 construct.

Mass spectrometry analysis of purified streptavidin-bound peptides was performed in four experimental conditions, as shown in Fig. EV3E. The analysis of the potential hits was performed as follows. We first compared biotin-treated YAP1-MYC-BIR2 samples (+dox, +biotin) to untreated YAP1-MYC-BIR2 samples (+dox, +biotin) and found 311 hits significantly enriched ($n = 3$, Log2FC > 2, FDR < 10%) (Fig. 3D, highlighted as pink dots in volcano plot). Then, we filtered out these 311 hits list, by discarding the proteins that were still biotinylated in the absence of YAP1-MYC-BIR2 construct (−dox, +biotin) or the absence of YAP1-MYC-BIR2 construct and biotin (−dox, −biotin) (see Methods for details). After these filters, we obtained a subset of 83 proteins (Fig. 3D, highlighted as red dots in volcano plot), as bonafide biotinylated targets of YAP1.

Among the 83 biotinylated hits, we found transcriptional co-activators and co-repressors reported to interact with YAP1, including the epigenetic reader BRD4 (Zanconato et al, 2018), the subunit of the chromatin remodeler complex SWI/SNF ARID2 (Chang et al, 2018), and the methyltransferase KMT2D (Fasciani et al, 2020). Additionally, we detected other well-known interactors of YAP1, such as the subunit of the Crumbs cell polarity complex MPDZ (Liu et al, 2021), and DVL3, a core component of the Wnt signaling pathway, that shuttles between the nucleus and cytoplasm and regulates YAP1's intracellular location (Barry et al, 2013) (Dataset EV2 and Fig. 3D,E). The analysis also retrieved new potential interactors, including the zing finger protein SALL2 (Kloet et al, 2018), the cohesin regulatory protein WAPL (Kueng et al, 2006), or the splicing and transcription regulator SNW1 (Wu et al, 2011). QSER1, a recently characterized protein, whose activity has been tightly associated with the protection of DNA methylation sites in hESCs was also among the 83 gene list (Dixon et al, 2021; Zhao et al, 2023) (Fig. 3D,E and Dataset EV2). Nonetheless, our mass spectrometry analysis detected few TEAD peptides, which did not reach the threshold for significant enrichment (see raw values in Dataset EV2 and Fig. EV3E). Other published BioID/TurboID datasets report similarly low TEAD recovery when YAP1 was used as bait (Zhu et al, 2019), which could be due to limited accessible lysines, rigid structure of TEAD or/and geometric constraints due to the tight YAP1:TEAD interactions (Mesrouze et al, 2018; Minde et al, 2020; Zhu et al, 2019).

## Computational docking predicts high stability for a QSER1:YAP1 complex

To prioritize potential YAP1 interactors among the hits identified in the BioID2 assay for downstream analysis, we performed computational docking to evaluate whether these proteins could plausibly form stable complexes with YAP1. Structural models from AlphaFold were used to simulate binding interactions using HDock (Yan et al, 2020; Yan et al, 2017a; Yan et al, 2017b), and each complex was assessed for stability through energy minimization (Case et al, 2023b). This analysis allowed us to rank candidate interactors based on predicted structural compatibility, with lower energies associated to more stable complexes (see Methods for details). As a reference, we included known YAP1 interactors—

TEAD3 (Zhao et al, 2008), LATS1 (Hao et al, 2008), and AMOT (Zhao et al, 2011)—to interpolate benchmark energy values (Fig. 3E). Among these positive controls, the AMOT:YAP1 complex exhibited the lowest energy. Notably, 10 of the 83 candidate proteins displayed even lower predicted energies than AMOT, including transcriptional regulators such as ARID1B and SNW1. Among these, the QSER1:YAP1 complex ranked within the top five most stable predictions. Considering its predicted binding stability along with QSER1's prominent yet poorly understood role in pluripotency (Dixon et al, 2021) we chose to investigate this potential functional interaction in greater detail. While docking identified CALD1 and GOLGA5 as top-scoring candidates, and functional relationships between YAP1 and both protein families have been reported (Kim et al, 2025; Kokate et al, 2022; Wang et al, 2017), neither CALD1 nor GOLGA5 is known to play a role in transcriptional regulation, yet additional studies would be required to assess the role of these candidates in this context.

To validate the predicted interaction between QSER1 and YAP1, we performed co-immunoprecipitation (Co-IP) assays in the YAP1-MYC-BIR2 hESCs lines, which confirmed that YAP1 interacts with endogenous QSER1 (Fig. EV3C). Immunofluorescence analysis further revealed that both proteins co-localize in the nucleus of hESCs (Fig. EV3F–H). In addition, we conducted size-exclusion chromatography experiments using nuclear extracts of hESCs, which showed that QSER1 and YAP1 co-elute in over-lapping fractions of high molecular weight, suggesting they are part of same nuclear complexes (Fig. EV3I). Together, these results provide strong evidence that QSER1 and YAP1 interact within the nucleus of hESCs.

## QSER1 is an intrinsically disordered protein enriched at promoters and enhancers in hESCs

Previous finding identified QSER1 through a methylation-based screen as a TET-interacting partner in hESCs (Dixon et al, 2021) and found a role for QSER1 in protecting DNA methylation, particularly at EZH2-bound loci (Dixon et al, 2021). However, ChIP-seq analyses of a FLAG-tagged QSER1 protein in hESCs revealed broad binding to enhancer regions, beyond EZH2 co-occupancy (Dixon et al, 2021), suggesting additional regulatory roles yet to be defined.

To further investigate the binding profile of QSER1 in the chromatin, we performed ChIP-seq analysis (Figs. 4 and EV4A) using a validated QSER1 antibody (Figs. EV3G and 6B). We confirmed that QSER1 binds extensively on the chromatin of hESCs, as we detected an average of 12,461 peaks, near 7942 genes. Importantly, a strong correlation was found between our datasets and Flag-tagged QSER1 ChIP-seq datasets previously published (Dixon et al, 2021) (Fig. EV4A), although our ChIPseq retrieved less peaks, probably due to differences in the antibodies used (QSER1 versus Flag). The genomic distribution of our QSER1 ChIP-seq datasets revealed that around ~65% of our QSER1 peaks lie on promoter regions, while the rest are spread among gene bodies and intergenic regions (Fig. EV4B). Supporting its proposed role in safeguarding DNA methylation (Dixon et al, 2021), ~48% of QSER1 peaks are located at CpG islands. However, the remaining peaks are found outside of CpG-rich regions, suggesting that QSER1 may have additional functions (Fig. EV4C). To further characterize the chromatin context of QSER1 binding, we

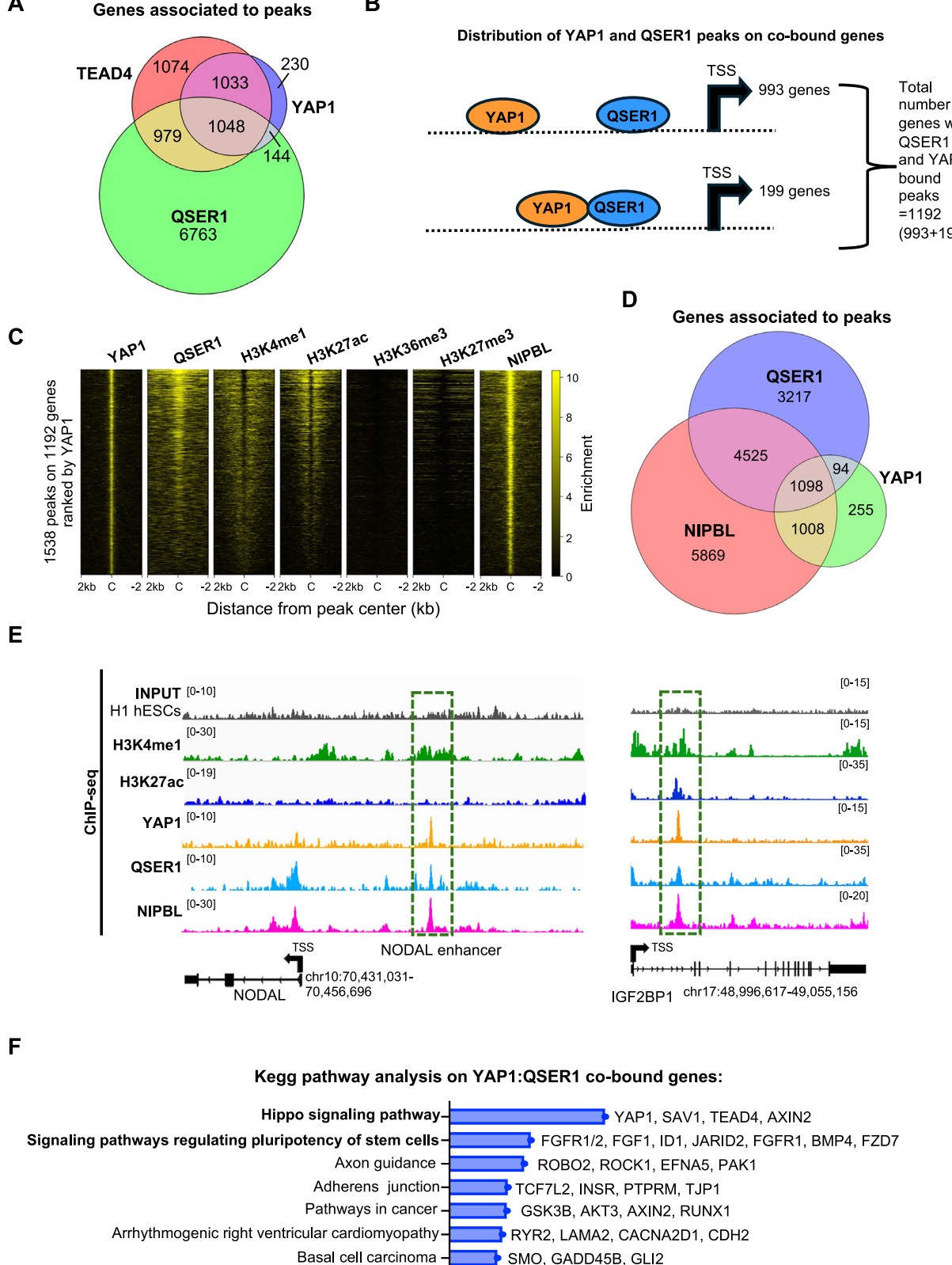

**A** Genes associated to peaks

**B** Distribution of YAP1 and QSER1 peaks on co-bound genes

**C**

**D** Genes associated to peaks

**E**

**F** Kegg pathway analysis on YAP1:QSER1 co-bound genes:

Hippo signaling pathway — YAP1, SAV1, TEAD4, AXIN2
Signaling pathways regulating pluripotency of stem cells — FGFR1/2, FGF1, ID1, JARID2, FGFR1, BMP4, FZD7
Axon guidance — ROBO2, ROCK1, EFNA5, PAK1
Adherens junction — TCF7L2, INSR, PTPRM, TJP1
Pathways in cancer — GSK3B, AKT3, AXIN2, RUNX1
Arrhythmogenic right ventricular cardiomyopathy — RYR2, LAMA2, CACNA2D1, CDH2
Basal cell carcinoma — SMO, GADD45B, GLI2
Cell adhesion molecules — NTNG1, CTNAP2, NLGN1, CADM1
Rap1 signaling pathway — DOCK4, FLT4, PDGFB, ADCY2
TGF-beta signaling pathway — TGIF1, SMAD2, SMAD3, INHBA, NODAL, ACVR2A

Log(adj.p-value)

◄ **Figure 4. QSER1 binds YAP1: TEAD4 enhancers near developmental genes.**

(A) Venn diagram shows the number of genes bound by QSER1, YAP1, and TEAD4 in hESCs (from ChIP-seq datasets). (B) Scheme shows YAP1 and QSER1 ChIP-seq peak distribution on co-bound genes in hESCs. Overlapping ovals denote perfectly aligned peaks. Ovals separated by a dotted line represent distant annotated to the same gene. (C) Analysis of chromatin features associated to QSER1:YAP peaks. Heatmaps depict the correlation of QSER1:YAP co-bound sites with indicated histone modifications (source; ENCODE, Bernstein datasets). Peaks are organized from high to low YAP1 signal (C = center of the peak, +/−2 kb). (D) Venn diagram shows the number of genes bound by QSER1, NIPBL (cohesin complex), and YAP1 (from ChIP-seq datasets) in hESCs. (E) IGV genome browser captures show YAP1 and QSER1 binding distribution at the indicated genes. H3K27ac and H3K4me1 histone marks are also shown. The coordinates of each genomic region are shown below each gene. TSS; transcriptional start site. The black arrow indicates the transcription direction. (F) Graph shows all enriched terms retrieved from Kegg Pathway analysis of the 1192 QSER1:YAP co-bound genes. Examples of genes included in each term are also indicated.

performed an overlap analysis using histone modification datasets from H1 hESCs (Fig. EV4D,E). QSER1 peaks showed strong enrichment at regions marked by H3K27ac, a signature of active enhancers (Rada-Iglesias et al, 2011), while a smaller subset overlapped with the repressive mark H3K27me3 (Rada-Iglesias et al, 2011) (Fig. EV4D–F). The co-occupancy with distinct histone marks suggests the existence of functionally diverse QSER1-bound chromatin pools.

To gain insight into QSER1 potential activities, we used the professional services of a Molecular Modeling facility to analyze QSER1's predicted structure through AlphaFold tools (Abramson et al, 2024). The results revealed that QSER1 is largely intrinsically disordered, with only two folded domains in the C-terminus: one of unknown function and a "bivalent Mical/EHBP Rab binding" (bMERB) domain (Rai et al, 2016), a mostly helical domain, which is likely to mediate protein–protein interactions, including those with Rab GTPases (Fig. EV4G,H). None of these domains are predicted to have enzymatic functions or catalytic activities. Notably, we found no evidence of a canonical DNA-binding domain, suggesting that QSER1 is likely recruited to chromatin through interactions with other DNA-bound proteins.

Given QSER1's extensive intrinsically disordered regions and lack of predicted catalytic or DNA-binding domains, we propose that QSER1 functions as a scaffold or co-regulator that facilitates protein–protein interactions within chromatin-associated complexes.

## QSER1 co-localizes with YAP1:TEAD4 at developmental enhancers in hESCs

Next, we intersected the QSER1 ChIP-seq with YAP1 and TEAD4 ChIP-seq datasets we previously reported (Estarás et al, 2017). Results revealed substantial co-occupancy with 48.5% of YAP1-bound genes (1192 out of 2455) and 49% of TEAD4-bound genes (2027 out of 4134) also displaying QSER1 binding (Fig. 4A and Dataset EV3). Nonetheless, QSER1 and YAP1 typically occupied distinct binding sites annotated to the same gene locus (Fig. 4B). This is consistent with the general chromatin distribution of both proteins, with QSER1 peaks typically binding closer to promoters and YAP1 peaks closer to enhancers (Fig. EV4B). Accordingly, most of the YAP peaks on the YAP1:QSER1 co-bound genes were enriched in typical enhancer marks H3K4me1 and H3K27ac (Rada-Iglesias et al, 2011), and were co-occupied by cohesin, as shown by the profile of the NIPBL cohesin subunit (Kagey et al, 2010) (Fig. 4C–E), suggesting that QSER1 may be brought into spatial proximity with YAP1 through chromatin looping. As stated above, genome-wide, ~50% of QSER1 peaks overlapped CpG islands. In

contrast, only 12% of QSER1:YAP1 co-bound peaks overlapped CpG islands (Fig. EV4C). Together, these findings suggest that (1) QSER1's role at YAP:TEAD-bound enhancers is likely independent of DNA methylation, and (2) that QSER1 recruitment to these sites may be mediated by TEAD or YAP1 tethering.

A KEGG pathway analysis (Kanehisa et al, 2016) of the 1192 co-bound genes revealed that the top significant enriched terms were "Signaling pathways in pluripotency" and "Hippo signaling pathway" (Fig. 4F). Other significant categories include "Pathway in cancer", which contain multiple Wnt and pluripotency genes, and "TGF-β signaling pathway" (Fig. 4F). In these categories we found, in addition to *NODAL*, multiple TGF-β signaling genes like *INHBA*, which encoding homodimers form the Activin A ligand (Abdel Mouti and Pauklin, 2021) and its associated Activin receptor type IIA (*ACVR2A*), which both have important roles in early development (Namwanje and Brown, 2016). We also found *TGIF1*, a critical inhibitor of the pathway (Powers et al, 2010), and *SMAD2* and *SMAD3*, which are the NODAL/ACTIVIN/ TGF-β signaling effectors (Zinski et al, 2018). Among WNT-target genes we identified critical regulators of WNT/β-CATENIN signaling, including the gene codifying for the *GSK3b* kinase (Lian et al, 2013), and *AXIN2*, a typical β-CATENIN target gene involved in restricting WNT activity (preprint: Hernández-Martínez et al, 2024). We also found several members of the Frizzled family receptors, including *FZD7*, that mediate classical and non-classical WNT signaling (He et al, 2021), and N-CADHERIN (codified by *CDH2* gene), a key molecule involved in epithelial-to-mesenchymal transitions, including those that occur during epiblast to PS transition in the embryo (Scheibner et al, 2021). Moreover, critical pluripotency and germ-layer (i.e., *CDX2*, *SOX2*, *OTX2, KDM2B, PRDM14*), Hippo pathway (i.e., *TEAD4, AMOTL2, WWC1*), and Fibroblast Growth Factor-related genes (i.e., *FGFR1/2*, *FGFBP3, FGF1*) were also targeted by both proteins (Dataset EV4). Other categories, such as "Axon guidance", "arrhythmogenic right ventricular cardiomyopathy" or "basal cell carcinoma" were also enriched, which likely reflect co-bound genes with broader or other context-specific functions. Taking together, we conclude that QSER1 and YAP1 co-occupy a common genetic program in hESCs enriched in pluripotency and lineage genes along with specific TGF-β/NODAL and WNT signaling genes.

## YAP1 recruits QSER1 to co-occupied developmental enhancers

YAP1 and QSER1 bind to 1192 genes in hESCs. From these, YAP1 and QSER1 peaks perfectly overlap on the same regulatory regions of 199 genes, while an additional 993 genes display YAP1 and

QSER1 peaks at non-overlapping regulatory regions (Fig. 4B). Here, we investigated the functional relationship between YAP1 and QSER1 on these genes. To do so, we analyzed the following three groups: (1) YAP1 and QSER1 co-occupied sites, such as those near *NODAL*, *CDX2* and *OTX2* genes; (2) YAP1 and QSER1 distal binding sites, like *SMAD2*, *SMARCA2*, and *INHBA*; and (3) no YAP binding, which are genes bound by QSER1 that do not have YAP1 binding sites associated to the same gene, as seen in *PDX1* and *PAX6* (Figs. 5A–D and EV5A,B). Then, we assessed QSER1 binding dynamics across these three types of regions in the presence and absence of YAP1, using ChIP-qPCR approaches. In YAP1 KO hESCs, we observed a significant decrease in QSER1 binding at regions co-occupied by YAP1, including *NODAL*, *CDX2*, *OTX2*, *SOX13*, and *SHB* genes, compared to WT (Fig. 5B). In contrast, QSER1 binding at non-YAP1 target genes remained unaffected (Fig. EV5B). Interestingly, YAP1 deletion also reduced QSER1 levels on *SMAD2*, along with other distal QSER1-bound sites analyzed (Fig. 5D). This further suggest a crosstalk between distal YAP1 and QSER1 bound regions, likely facilitated by cohesin-mediated distal interactions (Fig. 4C,D). On the other hand, QSER1 depletion (Fig. EV3F) did not affect YAP1 binding on the analyzed regions (Fig. EV5C). Altogether, these data show that YAP1 controls QSER1 recruitment on co-bound genes.

## Molecular modeling suggests YAP1 acts as a bridge between TEAD4 and QSER1

To test whether YAP1 binding to TEAD proteins is required for QSER1 recruitment, we treated human pluripotent stem cells with GNE-7883 (Hagenbeek et al, 2023), a pan-TEAD inhibitor that allosterically disrupts the interaction between YAP1/TAZ and all TEAD paralogs by binding to the conserved TEAD lipid pocket. As observed with YAP1 depletion, treatment with GNE-7883 led to decreased CTGF and increased *NODAL* expression, indicating effective YAP1:TEAD complex disruption (Fig. EV5D).

To investigate whether inhibition of YAP1 binding to TEAD affects QSER1 chromatin binding, we performed ChIP-qPCR at the NODAL enhancer, a region where QSER1, YAP1, and TEAD4 have been detected. GNE-7883 treatment significantly reduced both YAP1 and QSER1 occupancy at this site (Fig. EV5E), supporting a model in which YAP1:TEAD complexes facilitate the recruitment of QSER1 to target enhancers.

To further explore the structural basis of this interaction, we used molecular modeling to predict the configuration of the YAP1:TEAD:QSER1 complex on the NODAL enhancer, using the DNA sequence encompassing the TEAD binding motif (see Methods for details). The resulting AlphaFold 3 modeling yielded a strong ipTM score of 0.68, consistent with a high-confidence trimeric complex on the DNA (Fig. 5E). These structural predictions suggest that YAP1 functions as a molecular bridge: its residues 46–101 wrap around the second TEAD4 domain, and residues 50–60 are positioned in the interface between the bMERB domain of QSER1 and the YAP1 binding domain of TEAD4, acting like a bridge between QSER1 and TEAD4 (Figs. 5E and EV5F).

Because of weaknesses in ipTM scoring (Preprint: Dunbrack, 2025) particularly found in protein targets possessing large regions of intrinsically disordered residues (i.e., QSER1 N-terminal residues 1 to 1320), we trimmed the interacting domains of each of the proteins to optimize the ipTM score (see boundaries enumerated in

Methods). The resulting ipTM scoring for the TEAD4-YAP1-QSER1 complex showed a mean ipTM score of 0.61 for 25 models (standard deviation of 0.126) and strong conformational uniformity with a YAP1 β-strand bridging an exposed β-strand of TEAD4 and a β-strand of QSER1 (Fig. EV5F). In contrast, in the absence of YAP1, the TEAD4-QSER1 complex only yielded weak mean ipTM score of 0.31 (standard deviation of 0.27) for 25 models, which generated much more conformational heterogeneity in the location of QSER1 binding, and the resulting interfaces were weaker in general. This aligns with our ChIP-qPCR data, which show that YAP1 depletion leads to a marked reduction in QSER1 occupancy, while QSER1 loss does not affect YAP1 binding, which remains anchored via TEAD4 (Figs. 5B–D and EV5C).

Taken together, these structural predictions suggest that YAP1 functions as a molecular bridge, and in the absence of YAP1, QSER1 binding to YAP1:TEAD sites is compromised due to the loss of a stabilizing interface (Fig. 5F).

## QSER1 depletion increase RNA polymerase II occupancy on YAP1-co-bound enhancers

In mouse embryos, a regulatory region known as the Proximal Epiblast Enhancer (PEE) is critical for NODAL expression in the PS during gastrulation (Vincent et al, 2003). Our previous studies identified that the PEE region is conserved in the human genome and that YAP1 represses this enhancer in hESCs (Stronati et al, 2022). In this study, we identified that QSER1 co-occupies this key enhancer of *NODAL*, located ~13 kb upstream of the TSS in human cells. Thus, we speculate that QSER1 acts as a transcriptional co-repressor of YAP1 on this enhancer, to prevent excessive NODAL activation and subsequent premature differentiation of hESCs toward mesoendodermal lineages. To assess this, we analyzed RNA Polymerase II (RNAPII) binding to the *NODAL* enhancer. Our ChIP-qPCR analysis revealed that QSER1 depletion is sufficient to increase RNAPII abundance on the enhancer of *NODAL* (Figs. 6A–C and EV6A). Furthermore, we extend these analyses to the regions analyzed in Fig. 5B. Our ChIP-qPCR analysis revealed a similar trend across most regions examined: QSER1-bound sites displayed increased RNAPII occupancy upon QSER1 depletion (Figs. 6C and EV6B), suggesting that QSER1 may act to restrain transcriptional activity at the YAP1 co-occupied enhancers.

## QSER1 depletion induces NODAL signaling in hESCs

Consistent with increased RNAPII occupancy at the *NODAL* enhancer, QSER1 depletion led to elevated *NODAL* transcript levels following mesoderm-inducing signals in hESCs (Fig. 6D and see Methods for details) and a similar transcriptional upregulation was observed for *WNT3* in response to QSER1 depletion (Fig. 6D).

To further investigate the role of QSER1 in hESC differentiation, we applied 2D-directed differentiation strategies. Mesoderm induction normally requires the combined activity of WNT and NODAL/ACTIVIN effectors, β-catenin and SMADs. Treatment with low cytokine concentrations, or ACTIVIN alone, is generally insufficient to induce differentiation (Estarás et al, 2015; Estarás et al, 2017). However, we previously demonstrated that in the absence of YAP1, ACTIVIN treatment alone efficiently induces mesodermal markers, including BRACHYURY (Figs. 6E and EV6C,D; and (Estarás et al, 2017; Hsu et al, 2018)) due to elevated

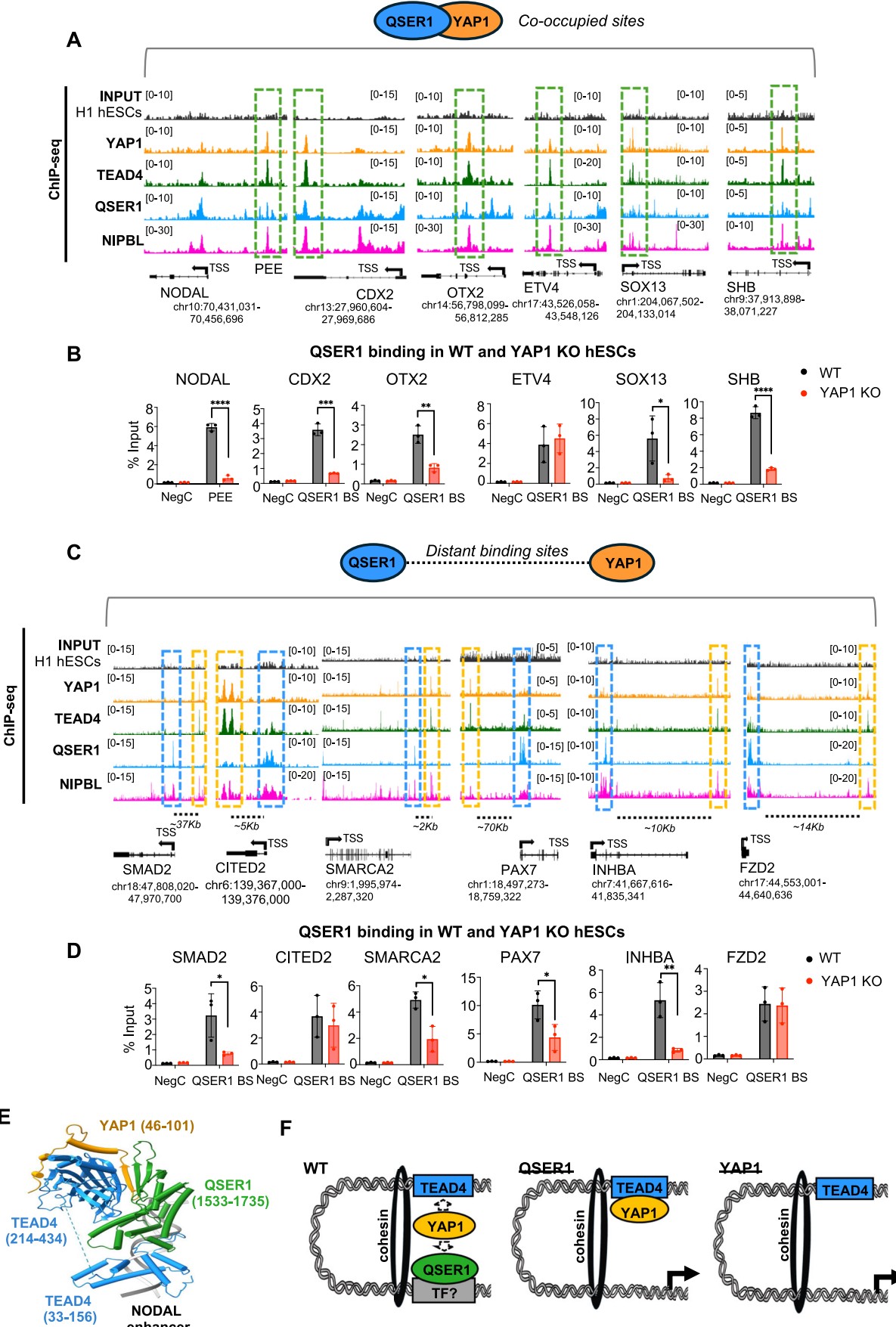

◀ **Figure 5.   YAP1 tethers QSER1 to TEAD4 enhancers.**

(A) IGV genome browser captures show peak distribution of QSER1, YAP1, TEAD4, and NIPBL on indicated genes. Dotted green boxes enclose the co-occupied regions of QSER1 and YAP1. The coordinates of each genomic region are shown below each gene. TSS; transcriptional start site. The black arrow indicates the transcription direction. (B) Graphs show ChIP-qPCR analysis of QSER1 protein on the indicated genomic regions in WT and YAP1 KO hESCs. QSER1 BS: QSER1 binding site, depicted in (A). NegC: Negative control region ($n = 3$, independent biological replicates). Data presented as mean ± SEM. Statistical analysis: Student's t-test, ****$p = 0.0001$, ***$p = 0.0002$, **$p = 0.0040$, *$p = 0.0394$. (C) IGV genome browser views showing the distal binding profile of YAP1, TEAD4, QSER1, and NIPBL on indicated genes. Blue and yellow boxes enclose QSER1 and YAP1 bound regions, respectively. Genomic coordinates corresponding to each locus are indicated below the gene tracks. TSS; transcriptional start site. Black arrow indicates transcription direction. (D) Graphs show ChIP-qPCR analysis of QSER1 protein on the distal binding sites in WT and YAP1 KO hESCs. QSER1 BS: QSER1 binding site, depicted in (A). NegC: Negative control region ($n = 3$, independent biological replicates). Data presented as mean ± SEM. Statistical analysis: Student's t-test, **$p = 0.0078$ and *$p = 0.0388$. (E) AlphaFold3 Modeling of a complex of TEAD4-YAP1-QSER1 proteins with a cognate TEAD4 binding site on the NODAL enhancer. The domain labeling shows the boundaries of the interacting protein fragments in parentheses. Helices are shown as tubes and ß-strands as arrows for simplicity, and the flexible linker between the two TEAD4 domains is replaced by a dotted line. The two DNA strands are shown as gray ribbons and the main TEAD4 DNA binding helix is bound in the major groove of the cognate site. (F) Scheme summarizing interpretation of results. QSER1 is brought in proximity to TEAD-bound enhancers through YAP1. It is likely that the presence of cohesin facilitates these distal interactions. Since QSER1 does not have a DNA binding domain it is possible that binds DNA through unidentified transcription factors (represented by the gray box; TF?).

endogenous expression of NODAL and WNT ligands in YAP-depleted cells (Estarás et al, 2015; Estarás et al, 2017).

To test whether QSER1 has a comparable role in mesoderm induction, we transiently depleted QSER1 in a knock-in NODAL-citrine hESC line, which expresses the mature NODAL protein fused to a citrine tag (Liu et al, 2022) (Fig. EV6E). QSER1 depletion led to elevated NODAL protein levels, consistent with increased NODAL enhancer activity and transcript accumulation. Moreover, similar to YAP-deficient cells, ACTIVIN treatment was sufficient to enhance mesoderm induction in siQSER1 cells, as evidenced by higher expression of the mesodermal marker BRACHYURY, compared with controls (Figs. 6E–F and EV6C,F,G).

Together, these phenotypes reinforce the conclusion that QSER1 and YAP1 regulate a common transcriptional program during early lineage specification, restricting gene induction in response to developmental signals.

## QSER1 expression is enriched in pluripotent cells

We examined the expression pattern of Qser1 across the early cell types of the E7 embryo. Unlike Yap1, which is homogenously distributed across pluripotent and differentiated cell types (Fig. 1), Qser1 is predominantly expressed in the epiblast, with reduced expression in more differentiated cell clusters (Fig. 6I,J). A similar trend is observed in hESCs, where a significantly reduction in the mRNA levels of Qser1 occurred during differentiation toward ectoderm, mesoderm and endoderm cell-fates, compared to pluripotent hESCs (Fig. 6K). Furthermore, we identified a mild, although significant, reduction of Qser1 levels in the Yap1 cKO embryos compared to controls, further supporting the relevance of QSER1 in maintaining the epiblast (Fig. 6I,J). Overall, we conclude that QSER1 is a specific co-factor of YAP1 in pluripotent cells required for the fine-tune regulation of gastrulation signaling genes (Fig. 6L).

## Discussion

In this study, we investigated the role of YAP1 in the differentiation of the mouse epiblast as well as the functional relationship with a new interactor in pluripotent cells, QSER1. Overall, our findings fill a long-standing gap in the field on defining the role of YAP1 in the differentiation of pluripotent cells,

in vivo, and provide new mechanistic insights into how YAP1 regulates developmental genes.

Collectively, our findings support a model in which YAP1 plays a role in safeguarding the pluripotent epiblast state by modulating the threshold of mesendoderm-inducing signals. Specifically, YAP1 appears to actively suppress premature activation of NODAL and WNT pathways, thereby preventing ectopic or excessive primitive streak formation. This regulatory function is conserved across species and model systems, as evidenced by the convergence of in vivo mouse embryonic phenotypes with human stem cell–based models (Estarás et al, 2017; Rito et al, 2024; Stronati et al, 2022).

The PS is expanded in YAP1-deficient epiblasts at E7. This observation, together with impaired neuroectoderm differentiation, revealed by the cell population analysis at E7.75 (Abraham et al, 2025) and morphological phenotyping at E8.25—suggests that YAP1 functions as a gatekeeper of lineage differentiation in the epiblast. Notably, enhanced PS specification and loss of anterior identity mirrors phenotypes caused by ectopic activation of WNT or NODAL signaling in the early embryo. For instance, excessive WNT signaling—such as that caused by activating mutations in Lrp6 (a WNT co-receptor) or Ctnnb1 (encoding β-CATENIN), or by deletion of the WNT inhibitor Dkk1—has been shown to disrupt neuroectoderm development and result in head truncation phenotypes (Arkell et al, 2013). Similarly, NODAL overexpression—whether by disruption of antagonists like Lefty1/2, or transcriptional repressors like Tgif1/2—lead to a variety of gastrulation defects, from expanded PS formation to failure to properly induce proper anterior ectodermal fates (Dai et al, 2016; Gripp et al, 2000; Iratni et al, 2002; Meno et al, 1999; Powers et al, 2010; Wotton et al, 1999). Thus, we speculate that, by limiting NODAL and WNT activity, YAP1 likely plays a critical role in ensuring proper spatial restriction of signaling domains during early patterning. However, our study did not include functional rescue experiments to test whether inhibition of NODAL or WNT signaling is sufficient to reduce PS differentiation or increase neural fold formation in Yap1 mutant embryos. Additionally, YAP1 also modulates retinoic acid signaling at later stages in mesoderm populations of the embryos (Abraham et al, 2025). This suggest that YAP1 may regulate developmental signaling pathways in a cell-type dependent manner, contributing to embryonic development through multiple mechanisms. Future studies will be needed to determine the extent to which these pathways contribute to YAP1's associated phenotypes during gastrulation.

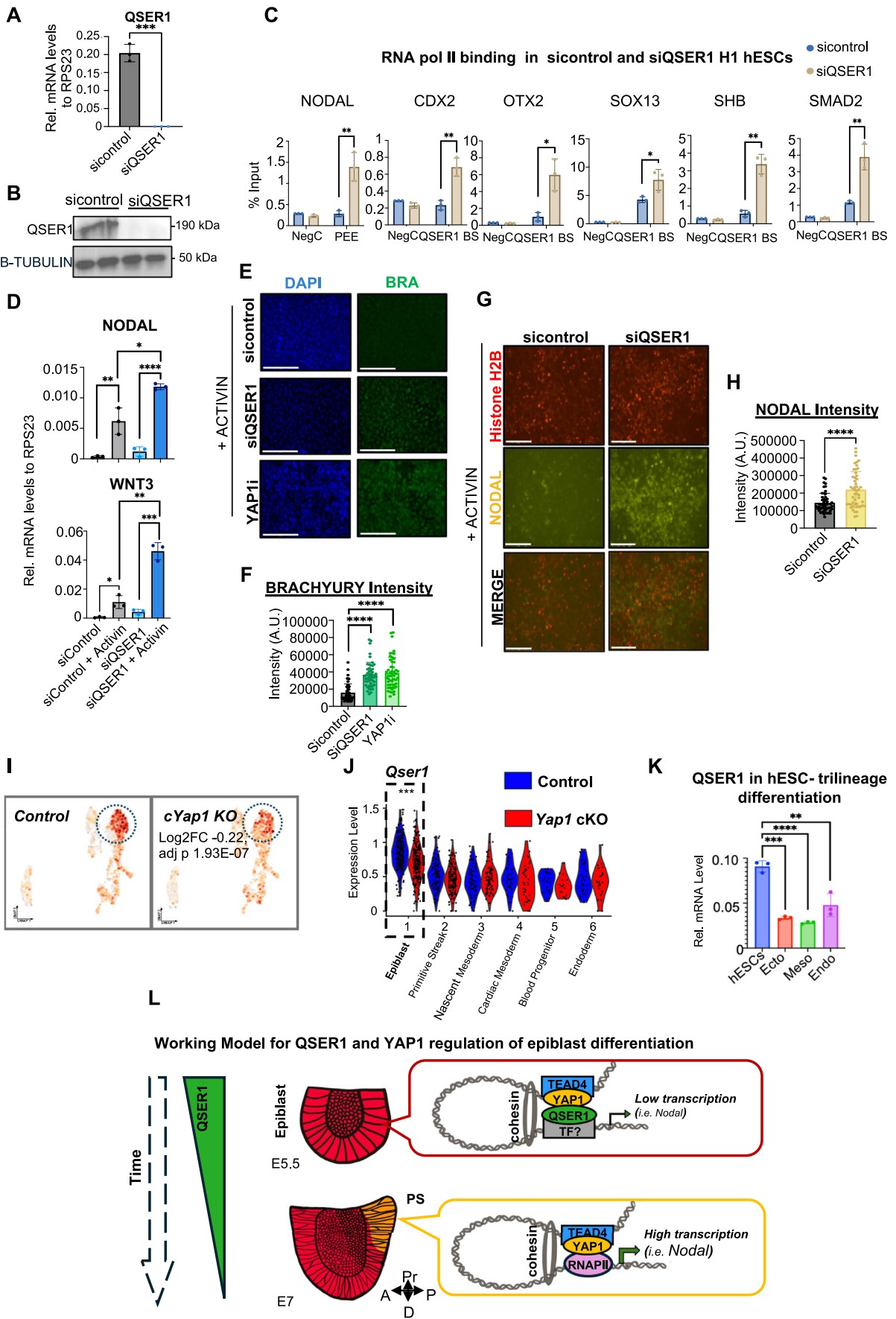

A

QSER1

B

sicontrol  siQSER1

QSER1 ────────── -190 kDa

B-TUBULIN ──────── -50 kDa

C

RNA pol II binding in sicontrol and siQSER1 H1 hESCs

● sicontrol
● siQSER1

NODAL    CDX2    OTX2    SOX13    SHB    SMAD2

D

NODAL

WNT3

E

DAPI    BRA

+ ACTIVIN

sicontrol / siQSER1 / YAP1i

F

BRACHYURY Intensity

G

sicontrol    siQSER1

Histone H2B / NODAL / MERGE

+ ACTIVIN

H

NODAL Intensity

I

Control    cYap1 KO

Log2FC -0.22,
adj p 1.93E-07

J

Qser1

■ Control
■ Yap1 cKO

Epiblast  Primitive Streak  Nascent Mesoderm  Cardiac Mesoderm  Blood Progenitor  Endoderm
1         2                 3                 4                 5                6

K

QSER1 in hESC- trilineage differentiation

hESCs  Ecto  Meso  Endo

L

Working Model for QSER1 and YAP1 regulation of epiblast differentiation

Time    QSER1

Epiblast
E5.5

cohesin  TEAD4 YAP1 QSER1 TF?    Low transcription (i.e. Nodal)

PS

E7

cohesin  TEAD4 YAP1 RNAPII    High transcription (i.e. Nodal)

Pr
A ◄►P
D

**Figure 6. QSER1 regulates developmental signaling pathways in hESC.**

(A) Graph shows QSER1 mRNA levels in hESCs transfected with siRNA control and siRNA against QSER1 for 72 h ($n = 3$, independent biological replicates). Data presented as mean ± SEM. Statistical analysis: Student's t-test, ***$p = 0.001$. (B) Western blot of QSER1 protein levels, same conditions as in (A). (C) Graphs show ChIP-qPCR analysis of RNA polymerase II protein on the indicated genomic regions in sicontrol and siQSER1 hESCs. QSER1 BS: QSER1 binding site NegC: Negative control region ($n = 3$, independent biological replicates). Data presented as mean ± SEM. Statistical analysis: Student's t-test, **$p = 0.0051$ (NODAL), **$p = 0.0031$ (CDX2), *$p = 0.0116$ (OTX2), *$p = 0.0343$ (SOX13), **$p = 0.0011$ (SHB), and **$p = 0.0037$ (SMAD2). (D) H1 hESCs were transfected with control or QSER1 siRNAs and left untreated or treated with Activin (=mesoderm inductor) for 24 h. Graphs show RT-qPCR analysis of NODAL and WNT3 genes ($n = 3$, independent biological replicates). Data presented as mean ± SEM. Statistical analysis: Student's t-test, Nodal (*$p = 0.0117$, **$p = 0.0098$, ****$p < 0.001$) and Wnt (*$p = 0.0154$, **$p = 0.0012$, ***$p = 0.0003$). (E) Representative images of hESCs treated with Activin (50 ng/mL) for 48 h and immunostained for BRACHYURY (BRA). The experimental groups are indicated; sicontrol (scramble siRNA), siQSER1 (siRNA against QSER1) or a YAP1 inhibitor (0.5 µM Dasatinib; YAP1i) were used. Scale bar, 50 µm. (F) Graph shows quantification of BRACHYURY immunostaining signal across the indicated experimental groups (50 cells were quantified from three biological replicates). Data presented as mean ± SEM. Statistical analysis: One-way ANOVA, ****$p < 0.0001$. (G) NODAL protein expression was visualized (live imaging) using an engineered dual-reporter line expressing NODAL-citrine and H2B-RFP (Liu et al, 2022). H2B-RFP is shown as control. Experimental groups are indicated. BF, bright field. Scale bar, 125 µm. (H) Graph shows quantification of intensity of citrine-Nodal levels (cNODAL) per cell (50 cells were quantified from three biological replicates). Data presented as mean ± SEM. Statistical analysis: Student's t-test, ****$p < 0.0001$. (I) UMAP from scRNAseq datasets of E7 embryos showing *Qser1* mRNA expression in control and cYap1 KO embryos. Dotted circles highlight the epiblast cluster (see Fig. 1). Differential *Qser1* expression in the epiblast of *Yap1* cKO versus control embryos is indicated (adj. $p = 1.93e-07$). (J) Violin plot of *Qser1* from scRNAseq of E7 embryos showing expression levels in all clusters Adjusted $p$-values were calculated using a Wilcoxon rank-sum test with Benjamini–Hochberg correction (***$p = 1.9e-07$). Each dot represents a single cell from E7 scRNAseq data. (K) WT H1 hESCs were differentiated toward ectoderm (ecto), mesoderm (meso), or endoderm (endo) fates followed by RNAseq analysis (Stronati et al, 2022). Graph shows the expression of QSER1 from these datasets in the indicated conditions ($n = 3$, independent biological replicates). Data presented as mean ± SEM. Statistical analysis: Student's t-test, ****$p < 0.0004$ and ***$p < 0.001$, **$p < 0.01$. (L) Cooperative mechanism of YAP1 and QSER1 modulating gene expression of signaling genes in the mammalian epiblast. Two developmental stages are shown. QSER1 expression decreases as the epiblast transitions to PS, which allows RNAPII recruitment and increased transcription of genes, including *Nodal*. PS: primitive streak. Pr: proximal, A: anterior, P: posterior, D: Distal.

Along these lines, we also observed potential non-autonomous effects of YAP1 deletion in extraembryonic populations. Extraembryonic tissues, including the anterior visceral endoderm (AVE) and extraembryonic ectoderm (ExE), are essential organizers of early embryonic patterning. These compartments act as key signaling hubs that influence epiblast behavior by secreting morphogens that regulate symmetry breaking, PS induction, and anterior–posterior (AP) axis formation. For instance, the ExE is a major source of BMP4, which induces *Wnt3* expression in adjacent epiblast cells to initiate PS formation, while the AVE secretes NODAL antagonists such as *Lefty1* and *Cer1* to restrict PS expansion and protect anterior fates (Bardot and Hadjantonakis, 2020; Morgani and Hadjantonakis, 2020; Perea-Gomez et al, 2001; Perea-Gomez et al, 2002). Although our genetic strategy using Sox2-Cre deletes *Yap1* specifically in the epiblast, we observed significant changes in the composition of extraembryonic lineages in *Yap1* cKO embryos. These alterations may reflect non-cell-autonomous effects of YAP1 loss in the epiblast. However, it is important to mention that our *Yap1*^[fl/fl] embryos (control and *cKO*) are likely to be heterozygous for *Yap1* in extraembryonic tissues due to constitutive paternal allele deletion, through Sox2-Cre:Yap1^[flox/+] male breeders (see Methods for details). Nonetheless, these observations raise the intriguing possibility that YAP1 may influence gastrulation not only through cell-intrinsic regulation, but also by modulating the epiblast–extraembryonic signaling axis. Future studies using lineage-specific Cre drivers to target YAP1 in extraembryonic compartments could help disentangle these effects and clarify whether YAP1 contributes to early patterning by affecting the formation or activity of extraembryonic cell populations.

Using a proximity labeling assay combined with in silico modeling and functional validation, we identified QSER1 as a novel co-factor of YAP1 in the regulation of a subset of developmental genes. QSER1 was recently characterized as a partner of TET enzymes and has been implicated in protecting CpG islands from DNA demethylation in hESCs (Dixon et al, 2021). Our findings extend the understanding of QSER1's functions, beyond DNA methylation. Our data showed that QSER1 depletion leads to NODAL de-repression and associates with increased RNA

Polymerase II occupancy at YAP1-bound enhancers, pointing to a QSER1 contribution maintaining low transcriptional activity on certain genomic regions. These observations are consistent with reported roles of QSER1 in cancer cells, where it has been linked to repression of apoptotic genes (Zhao et al, 2023). Our in silico analysis of the predicted structure revealed that QSER1 is an intrinsically disordered protein that lacks DNA-binding and catalytic domains. Based on these features and our findings, we hypothesize that QSER1 may act by stabilizing recruitment of transcriptional regulators, to fine-tune transcriptional levels, in accordance with the roles described for other intrinsically disordered proteins (Ferrie et al, 2022; He et al, 2024). Depending on the cellular context and the repertoire of interacting factors present, this aggregation-mediated buffering could differentially shape transcriptional outputs, supporting a model in which QSER1—like YAP1 (Estarás et al, 2017; Kim et al, 2015)—could contribute to either transcriptional repression or activation in a context-dependent manner.

Although YAP1 and QSER1 do not always co-occupy the same regulatory regions, our ChIP-qPCR analysis reveals that YAP1 influences QSER1 binding at both co-bound and distal sites, suggesting long-range regulatory effects. We propose that these distal regulatory regions, while separated linearly by several kilobases, are brought into spatial proximity through cohesin-mediated chromatin looping, enabling functional interaction between YAP1-bound and QSER1-bound elements. This model is supported by our findings showing extensive overlap between NIPBL, a core component of the cohesin complex (Kagey et al, 2010), and YAP1 peaks, suggesting that is frequently associated with loop anchor points, as previously reported (Galli et al, 2015). This also highlights a key technical limitation of conventional ChIP-seq, which relies on identifying DNA fragments in close proximity to the protein–DNA cross-linking site, and it often fails to detect secondary DNA anchor points mediated indirectly through protein–protein interactions. To resolve this, future studies using chromatin conformation capture techniques, such as Hi-C

(Hughes et al, 2014) or HiChIP (Mumbach et al, 2016), will be necessary to determine how many YAP1- and QSER1-bound enhancers are physically connected, and how this spatial organization contributes to transcriptional regulation during early lineage specification.

Finally, using hESC-based differentiation models, we showed that QSER1 depletion mimics the effects of YAP1 loss, and leads to enhanced NODAL and WNT signaling activities. Together, our results open new avenues to explore the role of QSER1 in in vivo models of gastrulation, and to investigate its broader functional interplay with YAP1 in other developmental contexts and disease states.

# Methods

### Reagents and tools table

| Reagent/Resource | Reference or Source | Identifier or Catalog Number |
|---|---|---|
| **Experimental models** | | |
| H1 hESCs (*H. sapiens*) | WiCell Research Institute | WAe001-A |
| YAP1 KO H1 hESCs (*H. sapiens*) | Estaras et al (2017) (Genes Dev) | Dr. Conchi Estaras |
| Nodal reporter (mCitrine::Nodal) hESC line (*H. sapiens*) | Liu et al (2022) (Nat Commun) | Dr. Aryeh Warmflash |
| Yap$^{flox}$ (*Yap1*$^{tm1.1Dupa}$/J) (*M. musculus*) | Jackson Research Laboratory | RRID:IMSR_JAX:027929 |
| Sox2$^{cre/+}$ B6.Cg-Edil3<Tg(Sox2-cre)1Amc >/J (*M. musculus*) | Jackson Research Laboratory | RRID:IMSR_JAX:00845 |
| **Recombinant DNA** | | |
| KA0717 plasmid | Estaras et al (2017) (Genes&Dev) | N/A |
| pPBCAG-rtTAM2-IN | Estaras et al (2017) (Genes&Dev) | N/A |
| pCAG-PBase | Estaras et al (2017) (Genes&Dev) | N/A |
| **Antibodies** | | |
| Alexa Fluor 488 Donkey anti-mouse | Thermo Fisher Scientific | A-21202 |
| Alexa Fluor 488 Donkey anti-rabbit | Thermo Fisher Scientific | A-21206 |
| Alexa Fluor 555 Donkey anti-goat | Thermo Fisher Scientific | A-21432 |
| Alexa Fluor 647 Donkey anti-rabbit | Thermo Fisher Scientific | A-31573 |
| Mouse anti-Beta-catenin | Santa Cruz | sc-7963 |
| DAPI | Cell Signaling Technologies | 4083S |
| Rabbit anti-BRACHYURY | Cell Signaling Technologies | D2Z37 |
| Mouse anti-Gapdh | Protein Tech | 60004-1 |
| Mouse anti-Histone H3 | Abcam | ab1791 |
| Mouse anti-Smad2/3 | Santa Cruz | sc-133098 |
| Mouse anti-Kdr/Flk1 | R&D Systems | FAB357P |

| Reagent/Resource | Reference or Source | Identifier or Catalog Number |
|---|---|---|
| Rabbit anti-Laminb1 | Abcam | ab16048 |
| Mouse anti-Yap1 | Santa Cruz | sc-101199 |
| Mouse anti-Tead4 | Abcam | ab5830 |
| IRDye 800 Streptavidin | LICOR | 926-32230 |
| Mouse anti-c-myc | Santa Cruz | sc-40 |
| Rabbit anti-Qser1 | Invitrogen | PA5-52181 |
| Rabbit anti-Nipbl | Bethyl Laboratories | A302-779A |
| Mouse anti-beta tublin | Santa Cruz | sc-5274 |
| Mouse DKXMO HRP XADS | Invitrogen | A16017 |
| Rabbit Igg HRP-linked | Cell Signaling Technologies | 7074P2 |
| **Oligonucleotides and other sequence-based reagents** | | |
| PCR primers and protocol | This study | Table EV1 |
| qPCR primers | This study | Table EV2 |
| **Chemicals, Enzymes and other reagents** | | |
| SiControl | Ambion Life Technologies | 4390843 |
| SiYAP1 | Ambion Life Technologies | s20366 |
| SiQSER1 | Ambion Life Technologies | s36436 |
| mTeSR1 media | Stem Cell | 85850 |
| Cultrex Reduced Growth Factor Basement Membrane Extract | R & D Systems | 3433-005-01 |
| UltraPure™ 0.5 M EDTA | Invitrogen | 15575020 |
| Accutase | Stem Cell | 7922 |
| Y-27632 dihydrochloride | Tocris | 1254 |
| DPBS, 1X, without Ca, Mg, Phenol Red, | Genesee | 25-508 |
| DPBS, 1X, with Ca and Mg | Quality Biological | 114-059-101 |
| DirectPCR Lysis Reagent | Viagen | 102-T 100 ml |
| SYBR Safe DNA gel stain | ThermoFisher Scientific | S33102 |
| GoTaq Green Master Mix | Promega | M712 |
| DSG (disuccinimidyl glutarate) | ThermoFisher Scientific | 20593 |
| Formaldehyde | ThermoFisher Scientific | 28908 |
| Pierce Protein A/G Magnetic Beads | ThermoFisher Scientific | 88803 |
| Qiaquick PCR purification | QIAGEN | 28106 |
| Fast SYBR™ Green Master Mix | Applied Biosystems | 4385612 |
| PowerUp SYBR Green Master Mix | Applied Biosystems | A25742 |
| NEBNext Ultra II DNA Library Prep Kit for Illumina | New England BioLabs | E7103 |
| GNE-7883 | MedChemExpress | HY-147214 |

| Reagent/Resource | Reference or Source | Identifier or Catalog Number |
|---|---|---|
| Dasatinib | Cell Signaling | 9052 |
| Puromycin | Sigma | P7255 |
| Geneticin™ Selective Antibiotic (G418 Sulfate) (50 mg/mL) | Gibco | 10131035 |
| Biotin | Sigma | B4639 |
| Doxycycline | Sigma | D5207 |
| Activin | StemCell Technologies | 78001 |
| Normal Donkey Serum | Jackson ImmunoResearch | 017-000-121 |
| Lipofectamine RNAiMAX Reagent | Invitrogen | 13778075 |
| Opti-MEM | Gibco | 31985-062 |
| Lab-Tek II Chamber Slide w/Cover RS Glass Slide Sterile | Nunc | 154534 |
| Bovine Serum Albumin | Sigma-Aldrich | A9418-100G |
| Triton™ X-100 | Sigma-Aldrich | 9036-19-5 |
| Fluoromount-G™ Mounting Medium | Invitrogen | 00-4958-02 |
| Prolong Glass Antifade Mountant | Invitrogen | P3698 |
| Quick RNA Miniprep Kit | Zymo | 11-328 |
| TRIzol | Invitrogen | 15596018 |
| Glycoblue | Invitrogen | AM9515 |
| iScript reverse transcription supermix for RT-qPCR | Biorad | 1708891 |
| RIPA Buffer | Santa Cruz | sc-24948 |
| Protein A/G PLUS-Agarose | Santa Cruz | sc-2003 |
| Protogel | National Diagnostics | EC-890 |
| Protogel Stacking Buffer | National Diagnostics | EC-893 |
| 4x Protogel Resolving Buffer | National Diagnostics | EC-892 |
| TEMED | Biorad | 161-0800 |
| Ammonium Persulfate | Biorad | 1610700 |
| 4–15% Mini-PROTEAN® TGX™ Precast Protein Gels | Biorad | 4561086 |
| PVDF Transfer Membranes, 0.45 µm | ThermoFisher Scientific | 88518 |
| SuperSignal™ West Pico PLUS Chemiluminescent Substrate | ThermoFisher Scientific | 34578 |
| Pierce™ IP Lysis Buffer | ThermoFisher Scientific | 87787 |
| SIGMAFAST protease inhibitor cocktail | Sigma-Aldrich | S8830-20TAB |
| Pierce™ BCA Protein Assay Kit | ThermoFisher Scientific | 23225 |

| Reagent/Resource | Reference or Source | Identifier or Catalog Number |
|---|---|---|
| Pierce™ 660 nm Protein Assay Reagent | ThermoFisher Scientific | 22660 |
| 1MDa Superose 6 Increase 10/300 GL column | Sigma-Aldrich | GE29-0915-96 |
| 3 kDa MWCO Amicon Ultra 0.5 centrifugal filter devices | Merck Millipore | UFC500396 |
| Gel filtration standards | Bio-Rad Laboratories | 1511901 |
| TrypLE™ Express Enzyme | Gibco | 12604013 |
| 10x Chromium Next GEM Single Cell 3′ GEM, Library & Gel Bead Kit v3.1 | 10x | PN-1000128 |
| **Software** | | |
| ImageJ 1.52 h | National Institute of Health | https://imagej.nih.gov/ |
| GraphPad Prism 10 | GraphPad Prism Software, Inc | https://www.graphpad.com/ |
| MATLAB | MathWorks | https://mathworks.com/ |
| Seurat R package version v4.3.0 and v3.9.9 | https://satijalab.org/seurat/ | |
| STAR v2.5.1b | https://doi.org/10.1093/bioinformatics/bts635. Dobin et al | |
| Cellranger v7.1.0 and v2.1.1 | 10x Genomics | |
| SnapTools (v1.4.7) and SnapATAC (v1.0.0) | UCSD | |
| ZEN 3.0 (blue edition) | Zeiss | |
| HOMER (v4.9.1) | http://homer.ucsd.edu/homer/ | |
| AlphaFold3 | https://doi.org/10.1038/s41586-024-07487-w | https://deepmind.google/science/alphafold/alphafold-server/ |
| HDock Lite v1.1 | Huang Lab of Bioinformatics and Molecular Modeling | N/A |
| H++ | https://doi.org/10.1093/nar/gks375 | http://newbiophysics.cs.vt.edu/H++/index.php |
| Amber | The Amber Project | N/A |

## Methods and protocols

### hESC lines and culture

WT and YAP1 KO H1 hESCs were previously described (Estarás et al, 2017). The Nodal reporter (mCitrine::Nodal) hESC line was kindly provided by Dr. Aryeh Warmflash's laboratory and described elsewhere (Liu et al, 2022). The doxycycline-inducible Yap1-Myc-Bir2 H1 cell line was generated using a PiggyBac transposon system. In this process, YAP1 CDS, fused to a biotin ligase and Myc tag, was cloned into KA0717 plasmid, and co-transfected along the transactivator pPBCAG-rtTAM2-IN and

transposase pCAG-PBase plasmids. These backbone plasmids were a gift from Dr. Kenjiro (Estarás et al, 2017). Transfected cells were isolated through G418 selection. Resistant clones were isolated and screened to confirm lack of leaking and robust expression of the construct upon doxycycline treatment. The doxycycline-inducible Yap1-Myc-Bir2 clonal cells were treated with 25 pg/ml of Doxycycline to overexpress YAP1 without degradation. To confirm YAP1 deletion and overexpression, sanger sequencing and WB analysis were performed. All cell lines were tested regularly for mycoplasma using published procedures and primers (Siegl et al, 2023) and were maintained at 37 °C and 5% $CO_2$ in a humidity-controlled environment. All hESC lines were cultured in mTESR1 medium on Matrigel-coated tissue culture plates. The media was replaced every day. Colonies were split when they reached ~70% confluence at a 1:20 ratio using 0.5 M PBS/EDTA. For single cell dissociation experiments, all colonies were disaggregated using Accutase, cells were counted and seeded in the presence of Rock inhibitor for 24 h. For directed differentiation to mesoderm, cells were treated with 50 ng/ml Activin for 48 h in mTESR1.

## Mice strains and housing/husbandry conditions for mice

All animal work complied with ethical regulations for animal testing and research and was conducted in accordance with IACUC approval by Temple University and followed all AAALAC guidelines (IACUC protocol #4973). All husbandry practices were used as previously described (Abraham et al, 2024). In brief, mice were housed with bedding material, mouse chow, fresh water, and an enrichment item. All mice were obtained from Jackson Laboratory, including Yap1$^{fl/fl}$ (Yap1$^{tm1.1Dupa}$/J) and the Sox2-Cre mice (B6.Cg-Edil3<Tg(Sox2-cre)1Amc > /J). The generation of conditional knock-out of YAP1 under Sox2-Cre were previously described (Abraham et al, 2025). In brief, Yap1$^{flox/flox}$ mice were crossed with Sox2-Cre:Yap1$^{flox/+}$ mice to generate conditional Yap1 knockouts (Sox2-Cre:Yap1$^{fl/fl}$) along with heterozygous littermate controls (Yap1$^{fl/fl}$; Sox2-Cre-negative embryos). The Yap1$^{fl/fl}$ embryos, even they are Sox2-Cre-negative, are likely to be heterozygous for Yap1 in extraembryonic tissues due to paternal allele deletion, through Sox2-Cre:Yap1$^{flox/+}$ males. This may have implications for interpreting potential effects on extraembryonic compartments (see Discussion). Only males carrying the Sox2-Cre allele were used for breeding due to known maternal inheritance of Sox2-Cre (Hayashi et al, 2003).

## Embryo isolation and genotyping

We followed established protocol from our lab to isolate embryos and perform genotyping (Abraham et al, 2025; Abraham et al, 2024). Timed pregnancies were confirmed by the visualization of a vaginal plug at noon, which was considered E0.5. Embryos used in this study were isolated at E7-E9.5 for staining, qPCR, western blot and single-cell RNAseq analysis. The visceral yolk sac was collected for genotyping. Mice were genotyped using standard flox, cre, and SRY protocols (Table EV1).

## ChIP-qPCR and ChIP-seq experiments

ChIP experiments were conducted as previously described (Estarás et al, 2015) In brief, $1 \times 10^6$ million cells were double cross-linked with 2 mM di (N-succinimidyl) glutarate DSG (45 min) and 1%

formaldehyde (15 min) at RT. The cells were lysed and sonicated using a Qsonica Q700 prove sonicator (7 min ON, 1 min ON/OFF, 25% amp). Following sonication, the chromatin was centrifuged at max speed to remove the non-sonicated chromatin for 30 min. The chromatin extract incubated with the 1–5 µg primary antibody overnight at 4 °C. After overnight incubation, protein A/G magnetic beads were added to the chromatin extracts for 4 h at 4 °C. The IPs were washed and reverse cross-linked with elution buffer overnight at 65 °C. The next day, DNA purification was performed using the Qiaquick PCR purification kit. For ChIP-qPCR experiments, the amplification as performed on a Quant Studio 3 device using FAST SYBR green master mix or PowerUp SYBR green master mix for amplification. The DNA was eluted in 50–75 µl and 2 µl was used for each reaction. For ChIP-seq experiments, the DNA was quantified using a Qubit Flex and the libraries were prepared using the NEBNext DNA Library kit. The quality of the libraries was analyzed using the Agilent 4150 Tapestation. Samples were sequenced using a P2 flow cell and NextSeq 1000 instrument. Primers for ChIP-qPCR and ChIP antibodies are listed in Table EV2.

## Biotinylation assay

The doxycycline (dox)-inducible Yap1-Myc-Bir2 H1 cell line was used to perform the proximity ligation assay. We performed this experiment using four different experimental groups (minus dox minus biotin, minus dox plus biotin, plus dox minus biotin, and plus dox plus biotin). To induce the protein construct, 25 pg/ml doxycycline was added to the media for 24 h, followed by incubation with 50 mM biotin (or untreated) for 16 h to label all proteins in proximity to YAP1. To identify nuclear interactors of YAP1, the nuclear fraction was isolated for analysis of biotin-labeled proteins. Briefly, nuclear lysis buffer (1 M Tris-HCl pH 7, 5 M NaCl, 1 M $MgCl_2$, 10% NP-40, 1 M DTT, and 1x protease inhibitor) was added to the cells and incubated on ice for 10 min. The cells were then centrifuged at $500 \times g$ for 5 min at 4 °C. The supernatant was removed, and the pellet was washed with DPBS$-/-$ and centrifuged again at $500 \times g$ for 5 min at 4 °C. The supernatant was discarded, and the pellet was snap-frozen and stored at $-80$ °C before being shipped to the Proteomics Core at Sanford-Burnham-Prebys Medical Discovery Institute, where the pull-down and mass spectrometry were performed using an established protocol. Briefly, cells were lysed in 8 M urea, 50 mM ammonium bicarbonate (ABC) and benzonase, then centrifuged at $14,000 \times g$ for 15 min. Disulfide bridges were reduced with 5 mM tris(2-carboxyethyl)phosphine (TCEP) at 30 °C for 1 h and alkylated with 15 mM iodoacetamide (IAA) in the dark at room temperature for 30 min at room temperature. Streptavidin-based affinity purification was performed on the Bravo AssayMap platform (Agilent) using AssayMap streptavidin cartridges (Agilent). Biotin-enriched peptides were reconstituted with 2% ACN, 0.1% FA, quantified by NanoDrop, and analyzed by LC-MS/MS using a Proxeon EASY-nanoLC system (ThermoFisher) coupled to a Orbitrap Fusion Lumos Tribid mass spectrometer (Thermo Fisher Scientific). Spectra were processed with MaxQuant software version 1.6.11.0 and searched against the Homo Sapiens Uniprot protein sequence database (downloaded in Jan 2020) and GPM cRAP sequences (commonly known protein contaminants). Limma software analysis was applied to identified differential enriched

proteins in the four experimental conditions (Ritchie et al, 2015). An initial list of 311 hits was obtained from proteins significantly enriched (FC > 2, FDR < 10%, with MS values in at least 2 replicates), in samples expressing YAP1-MYC-BIR2 construct treated with biotin (+dox,+biotin) versus those without biotin (+dox, −biotin). To refine the list of candidate interactors, the 311 initial hits were filtered based on fold change (Log2FC > 1.5) to exclude proteins that remained biotinylated in control conditions lacking the YAP1-MYC-BIR2 construct (−Dox, +Biotin) or both YAP1-MYC-BIR2 and biotin (−Dox, −Biotin). This filtering yielded a final subset of 83 proteins, which were then subjected to in silico protein–protein docking analysis to evaluate potential interactions with YAP1.

## Protein docking

Computation protein docking screening was performed on the list of 83 potential interactors from BioID screening of YAP1 and three positive controls (TEAD3, AMOT, and LATS1). All protein structures were downloaded as PDB files from the AlphaFold human proteome dataset, which were shown to have a quality near to that of experimental methods as shown in the Critical Assessment of Techniques for Protein Structure Prediction (CASP14), making them a reliable option for in silico protein studies (Jumper et al, 2021; Kryshtafovych et al, 2021; Tunyasu-vunakool et al, 2021). HDock software was used to dock each protein to YAP1 (Huang and Zou, 2008, 2014; Yan et al, 2020; Yan et al, 2017a; Yan et al, 2017b). This software was chosen due to it being available as a standalone, downloadable package while also being one of the more accurate algorithms for protein docking according to the Critical Assessment of Predicted Interactions (CAPRI) (Wodak et al, 2023). HDOCK performs template-free, fast Fourier transform protein docking, which performs protein dock-ings ab initio without a homologous template. Since many of these complexes are novel interactions, template-free docking reduces possible bias from homologous templates. Fast Fourier transform protein docking converts the cartesian coordinates of each residue in both proteins into a three-dimensional grid with their associated values based on whether it is located on the surface, inside, or core of the protein (Katchalski-Katzir et al, 1992). One protein is kept static as the "receptor", while the other protein is rotated around it as the "ligand". HDock uses a 15-degree angle interval and a 1.2 Å translational interval, undergoing a total of 4392 rotations where one binding mode is kept per rotation. These output binding poses are then clustered with an RMSD of 5 Å, and the top ten structures are output as the final structures, in order of their scores from the HDock scoring function.

As a means of evaluating the stability of each of the docked structures, energy minimization calculations were performed using Amber software to provide an approximate ranking of these interactions (Case et al, 2023a). This software was used due to its efficiency and ability to carry out quality simulations, especially those involving proteins and explicit solvent (Case et al, 2005). To prepare the docked protein files for use with Amber, they were first run on the H + + server, which predicts the protonation states of the histidine residues as well as the overall charge of the protein at biological pH (Anandakrishnan et al, 2012). The prepared PDB file from H + + was loaded in Amber with the ff19SB force field (Tian et al, 2020). Ions were added to neutralize the system, then the

protein was solvated using the OPC water model, which uses explicit solvation to model the aqueous solvent (Izadi et al, 2014). Ions were also randomly placed using the OPC model at a concentration of 150 mM to mimic the cellular environment (Li et al, 2015). Once the system was solvated and buffered, the energy minimization was run, which outputs a relaxed structure in its lowest energy conformation as well as the potential energy associated with this structure. This potential energy takes into account protein–protein interactions, protein-solvent interactions, and solvent-solvent interactions. While it does not give the binding free energy of the proteins, it outputs a potential energy value that can yield an approximate ranking and reflects the relative stability of the docked complexes. A more negative potential energy value from these calculations indicates that a complex is more stable, and the complexes with the lowest potential energy are more likely to represent biologically relevant interactions to be studied further.

## Treatment of inhibitors and directed differentiation of hESCs

For hESC experiments involving TEAD inhibition, cells were seeded and treated with 1 µM GNE-7883 for 24 h before being collected for downstream analysis. For hESC experiments using YAP1 inhibitor DASATINIB, cells were seeded and treated with 0.5 µM for 72 h. For directed differentiation to mesoderm, cells were treated with 50 ng/ml Activin for 48 h in mTESR1.

## Immunostaining, imaging, and quantification of hESCs

Glass chamber slides were coated with 5 µg/ml of laminin-521 in PBS+/+ for 2 h at 37 °C. Cells were dissociated using Accutase and seeded onto the coated slides. Once cells reached 60–70% confluency, they were fixed with 4% formaldehyde for 15 min at room temperature.

Following fixation, cells were permeabilized with 0.5% Triton X-100 for 10 min and washed with PBS−/−. They were then incubated with blocking solution (0.5% PBS-Tween, 1%, BSA, 10% FBS) for 30 min at room temperature. Primary antibodies, diluted in blocking solution, were incubated overnight at 4 °C. The following day, cells were washed and incubated with secondary antibodies and DAPI for 20 min at room temperature. Finally, cells were washed PBS and mounted with a glass coverslip.

Fluorescent intensity analysis was performed using the *ImageJ 1.53t* program. Images were taken using an EVOS M7000 microscope at 20 and 63x. A region of interest (ROI) was drawn using the freehand selection tool around the nucleus of a single cell using the DAPI channel. This ROI was then copied onto the RFP channel in the same exact position. Within the ROI, the sum of the intensity of the pixels was then measured in the DAPI and RFP channels individually. The RFP raw intensity was then normalized to the DAPI intensity of the same cell and multiplied by 100 to attain percent positivity. The fluorescence of 25 randomly selected individual cells were measured for each replicate. Pearson correlation coefficient was analyzed using the ZEN 3.0 (blue edition) software. Airyscan images of single cells were taken using the Zeiss LSM900 with Airyscan 2. DAPI, GFP, and RFP were imaged using the 405 nm, 488 nm, and 561 nm lasers, respectively. The ZEN 3.0 (blue edition) software was used to analyze the Pearson correlation coefficient for the relationship between the

colocalization of the GFP and RFP signal. In order to set the crosshairs for accurate analysis of positive correlation, the background colocalization of both the 488 nm and 561 nm lasers were determined by their colocalization with the unused 640 nm laser for which no secondary antibody was utilized. Pearson correlation coefficient values above 0.7 are considered significant.

## RNA isolation, cDNA conversion, and qPCR

RNA extraction of hESCs was performed using the Quick RNA Extraction Zymo Kit. Mouse embryos were snap-frozen with Trizol in liquid nitrogen immediately after isolation. After genotyping, RNA extraction from mouse embryos was performed by pooling 3–4 embryos of each genotype. The Trizol procedure was applied following manufacture intstructions. In brief, following lysis with Trizol, chloroform and isopropanol were added to the samples for phase separation and RNA precipitation. Glycoblue was added to the RNA solution prior to resuspension in water. A total of 250–500 ng of RNA was reverse transcribed using the iScript Reverse Transcription Supermix. The cDNA samples were then amplified using FAST SYBR Green Master Mix or PowerUp SYBR Master Mix on a Quant Studio 3. All results were normalized to RPS23 and GAPDH unless otherwise stated, and the ΔΔCt method was used to calculate relative transcript abundance against the indicated references. Primers for cDNA amplification are listed in Table EV2.

## SiRNA transfections in hESCs

SiRNA transfections were performed using the Lipofectamine RNAiMAX transfection reagent following manufacturer's instructions. Briefly, cells were dissociated using Accustase, and a mix of SiRNA, lipofectamine, and Opti-mem was added to the cells along with culture media and ROCK inhibitor. Transfection efficiency and downstream experiments were performed 72 h post-transfection.

## Protein isolation, co-immunoprecipitation (Co-IP) and western blots

For hESCs, cells were lysed using RIPA buffer containing protease and phosphatase inhibitors and left on ice for 20 min. For mouse embryos, after isolation, the embryos were snap-frozen in liquid nitrogen. Once the genotypes were confirmed, 8–10 embryos per genotype were pooled and lysed using RIPA buffer. Samples with RIPA were centrifuged at max speed for 10 min, and the supernatant was quantified using BCA assays. For Co-IP experiment, WT hESCs were washed with PBS−/− and lysed on ice for 20 min in IPH buffer (50 mM Tris/HCl), 150 mM NaCl, 5 mM EDTA, 0.5% Igepal). Lysates were then centrifuged at $13,000 \times g$ for 10 min at 4 °C. The resulting supernatant was pre-cleared with agarose beads by rotating for 45 min at 4 °C. After separating the input, lysates were incubated overnight with primary antibodies at 4 °C with rotation. The following day, Protein A/G agarose beads were added and rotated for an additional 3 h at 4 °C. Beads were washed three times with IPH buffer, and sample loading buffer was added. Samples were boiled at 95 °C for 10 min and then loaded onto a gel. Following quantification, SDS-PAGE electrophoresis was performed with 10–40 µg of protein per sample for hESCs and

50–70 µg of protein per sample for mouse embryos. Wet transfer was carried out using 0.45 µm PVDF membranes. The membranes were blocked with 2% BSA in TBS-T for 1 h at room temperature. Primary antibodies were diluted in 2% BSA in TBS-T and incubated overnight at 4 °C. HRP-conjugated secondary antibodies and SuperSignal West Pico chemiluminescent substrate were used for protein detection on a BioRad Chemi-Doc Touch Detection System and LiCor Odyssey DLx Imager.

## Fast protein liquid chromatography (FPLC)

Around 3 million cells of WT hESCs were lysed on ice for 30 min in Pierce™ IP Lysis Buffer (Thermo Fisher) with 1× SIGMAFAST protease inhibitor cocktail. Lysates were centrifuged at $13,000 \times g$ for 10 min at 4 °C and the protein concentration in the supernatant was determined using Pierce™ 660 nm Protein Assay Reagent (Thermo Fisher). 2 mg of protein were fractionated using an FPLC (ÄKTA pure FPLC, GE Healthcare), using a 1MDa Superose 6 Increase 10/300 GL column (#GE29-0915-96, Sigma-Aldrich) equilibrated in 1× PBS; 0.5-ml fractions were collected at a flow rate of 0.5 ml min$^{-1}$. Then, fractions were concentrated to 75 µl with 3 kDa MWCO Amicon Ultra 0.5 centrifugal filter devices (#UFC500396, Merck Millipore) and used for immunoblotting. The molecular masses of the FPLC fractions were calibrated using gel filtration standards (#1511901, Bio-Rad Laboratories). Western blot intensities were analyzed using the ImageJ software. The intensity of the protein of interest was quantified and normalized to the intensity of the housekeeping protein.

## Single-cell RNAseq of E7 embryos

Procedures for single-cell analysis of mouse embryos during gastrulation followed protocols previously described by the Estaras lab (Abraham et al, 2025; Abraham et al, 2024). For E7 analysis, multiple pregnancies were synchronized, and embryos were isolated at 7 am seven days following the visualization of the vaginal plugs. Following genotyping, 3 cYap1 KO embryos (Sox2-Cre:Yapfl/fl) and 4 heterozygous controls (Sox2-Cre:Yapfl/+) were pooled and dissociated with Tryple for 5 min. The 10x Chromium Next GEM Single Cell 3' GEM, Library & Gel Bead Kit v3.1 was used for library generation. Qubit and Agilent 4150 Tapestation were used for library quantification before pooling. Libraries were pooled and sequenced using NextSeq2000.

## Whole-mount immunostaining, imaging, and quantification of mouse embryos

After isolation, embryos were placed into a 48-well cell culture dish containing 4% formaldehyde and incubated for 45 min at room temperature. Following fixation, the embryos were washed three times with PBS and stored at 4 °C. Once genotyping was confirmed, embryos of the same genotype were pooled into the same well. The embryos were permeabilized with 1% SDS in PBS for 15 min at room temperature while rocking. Afterward, the embryos were washed three times with PBS, and 0.5% Triton X-100 in PBS was added for 15 min at RT. The embryos were then washed three more times with PBS, and primary antibodies were added in blocking solution (PBS-T and 10% FBS). The following day, embryos were washed three times with PBS, and secondary antibodies, along with

DAPI (1 mg/ml), were added for 2 h at RT while rocking. The embryos were then mounted on glass slides with mounting media. Images were captured using the EVOS M7000. Immunofluorescence images were analyzed using the line tool in MATLAB to quantify the fluorescence intensity along the by proximal-to-distal axis of the embryo (Stronati et al, 2022). For the proximal-to-distal axis quantification; three "cup shaped" lines were drawn from the proximal posterior part of the epiblast to the most distal anterior part of the epiblast. The fluorescence signal of the three lines was averaged for each embryo. Each embryo image was rescaled to the same size, so distances are comparable. For the posterior-to-anterior quantification; the extraembryonic tissue and embryonic were located by eye. The embryo was divided in six bins by drawing five parallel straight lines from the boundary between the extraembryonic tissue down to the distal tip of the epiblast. The average fluorescence signal of each bin was plotted in the graph (from posterior to anterior). The average of the intensity profiles from 3 embryos per condition were plotted. Brightfield images were also taken after each embryo isolation, and phenotyping analysis was conducted. The perimeter of the epiblast was quantified using ImageJ, and the number of cells per embryo was calculated by trypsinizing the cells and counting them using a Countess 3 device.

## Modeling AlphaFold proteins and protein–DNA complexes

The AlphaFold3 (AF3) and AlphaFold2 (AF2) programs were utilized (Abramson et al, 2024). Protein sequences and domain boundaries were refined from Uniprot entries for the following proteins: TEAD4_HUMAN (Q15561), YAP1_HUMAN (P46937), and QSER1_HUMAN (Q2KHR3). AF3 and AF2 modeling predicted QSER1 structure and identified domains. The AF3 program was utilized to model transcription factors bound to DNA forming a variety of complexes. Given the experimental evidence for transcription factor binding to the Nodal enhancer presented in this work, a 36 base-pair portion of this DNA region was chosen as it contained a single TEAD4 cognate binding site (sequence    TGCATTCCCCACTAACATCAAAAAGCCTGGGA-GAGC, cognate binding site underlined) To make a model for this specific portion of the enhancer, a TEAD4 protein monomer, and a single peptide of YAP1, and various portions of the QSER1 protein were utilized. In all cases, the models produced had the DNA-binding domain of TEAD4 positioned precisely to allow extensive binding contacts between the main DNA-binding helix to the major groove of the cognate DNA sequence. In comparing TEAD4 binding to its cognate sequence in known structures, the AF3 model had 8 hydrogen bonds to DNA compared to 10 hydrogen bonds found in the known X-ray crystal structure PDB code 5GZB, with 6 amino acids in common to both structures.

TEAD4 has an N-terminal DNA-binding domain followed by a flexible unstructured linker which connects to a second folded effector domain in the C-terminal half of the protein. The cognate DNA sequence strictly governed the position of the N-terminal TEAD4 DNA binding domain, while the flexible linker allowed many different positions for the C-terminal domains among the various models sampled. When YAP1 is present (especially residues 46–101) it wraps around the second TEAD4 domain and acts like "molecular glue" to bridge TEAD4 binding to the QSER1-bMERB

domain found at the C-terminus. A beta strand from YAP1 inserts between the TEAD4 domain and the QSER1-bMERB domain to form a tighter interface that AF3 predicts to be more stable. While previous crystal structures have shown mouse (3JUA) and human (8A8R) YAP1 wrapping around a single TEAD domain, these AF3 models demonstrate a YAP1 peptide inserting between TEAD4 and QSER1. Aligning and super-positioning of protein–DNA complexes, analysis of interfaces and contacts, and figure preparation were done with ChimeraX (Meng et al, 2023).

## Quantitative scoring and N-terminal trimming of proteins used in modeling

AF3 modeling was first performed with full-length canonical protein sequences from Uniprot. To optimize AF3 modeling and scores, subsequent models were built with protein sequences that trim away flexible unstructured N-terminal tails as follows: TEAD4(1-32), YAP1(1-45), and QSER1(1-1532). The final domain boundaries are shown as residue numbers in the color-coded labels of the model Fig. 5E. To quantitatively compare the scoring of protein complexes in the presence and absence of YAP1(46-101), the DNA was omitted and the ColabFold version (Mirdita et al, 2022) of AlphaFold2 was used with higher sampling ($n = 25$ models for each protein complex) and total ipTM scores were compared with and without the YAP1 peptide. The geometric mean of the ipTM scores and standard deviation (Excel STDEV.P function for the whole population) were calculated for each set of 25 models. Structural and conformational homogeneity was assessed by aligning the 25 models on TEAD4 domains, and visual inspection of the homogeneity of YAP1 and or QSER1 positioning.

## scRNAseq analysis of E7 embryos

Single-cell sequencing reads were counted individually for each sample using cellranger (v7.1.0) with default parameter on mm10 mice genome. Detection of doublet and cells contaminated with ambient RNA were assessed using scrublet and soupX (v1.6.2), in Python and R, respectively. Low-quality cells, including doublets, cells with over 10% mitochondrial content, and those expressing fewer than 5000 RNA features, were filtered out. Most analysis were performed using the Seurat R package (v4.3.0), including quality control plots generated with FeatureScatter and VlnPlot Seurat' functions. Samples were normalized and variance stabilized using the SCTransform function in Seurat, regressing out variables including nCount_RNA, percent.mt, percent.rb, S.Score, and G2M.Score. Principal component analysis (PCA) was run with 19 principal components for both samples. The samples were then integrated using Seurat's SelectIntegrationFeatures, PrepSCTIntegration, FindIntegrationAnchors, and IntegrateData functions with 3000 selected features and the SCTransform normalization method. The populations were annotated based on the differential expression of known markers tested against all other clusters combined. Selected markers with significant adjusted $p$-values in the assigned population were used to generate the heatmap in Fig. 1B. For example, Sox2 is significantly upregulated in the epiblast cluster relative to all other clusters. For visualization of gene expression in UMAP, and heatmap, the SCT, and RNA assay were used, respectively. Differential gene expression analysis for each cluster

between genotypes was performed on the RNA assay using the FindMarkers Seurat function after log normalization and scaling. Genes were considered differentially expressed when adjusted *p*-value, based on Bonferroni correction was below 0.05. Cell type count differences between genotypes for each cell types were assessed by randomly downsampling cYap1 KO samples 100 times to the same number of cells as the WT sample (788 cells). For each iteration, the number of cells in each cluster was counted. Chi-squared tests with Bonferroni correction were used to assessed significant differences in cell counts between genotypes across clusters. Pathway analysis was performed using the SCPA R package (v1.5.4) with curated gene lists selected from the MSigDB C2 database.

### ChIP-seq analysis

ChIP fragments were sequenced in an Illumina NextSeq 1000 sequencer in SE configuration. Adapter and poor-quality sequences were trimmed, and cleaned, using fastp (v0.23.2). Reads were aligned to the Human hg38 genome assembly (GRCh38) using bowtie2 (v2.2.5) with parameters '--phred33 -q --no-unal'. Reads with a mapping quality score (MAPQ) below 20, unmapped reads, secondary alignments, and reads failing quality checks were removed using SAMtools (v1.16.1), and duplicated reads were removed using Picard's MarkDuplicates function. Peaks were identified using getDifferentialPeaksReplicates.pl from HOMER (v4.11) with style '-factor' parameter and input sample as background reference. Peaks overlapping were assessed using SRplot (*ref: PMID: 37943830*). Differential peak enrichment between genotypes was conducted using rgt-THOR (*ref: PMID: 27484474*) with parameters '--merge --deadzones' to combine peaks, and exclude regions from the hg38 blacklist. In addition to differential peak calls, rgt-THOR generated a bigwig file for visualization. Peaks, and differential peaks, were assigned to genes using the nearest TSS method with ChIPseeker (v1.18.0). Two independent sets of histone modification ChIP-seq samples for H3K27me3, H3K4me1, H3K36me3, and H3K27ac were obtained from ENCODE, generated by the labs of Bing Ren (GSE16256) and Bradley Bernstein (GSE29611). Peak and bigwig coverage files were directly retrieved from ENCODE for downstream analysis. CpG islands were retrieved from the hg38 UCSC genome annotation database. Heatmaps of ChIP-seq signals were generated using deepTools' computeMatrix and plotHeatmap functions. Gene ontology analysis was performed using the enrichR R package (v3.2) using the GO_Biological_Process_2023 library and visualized with ggplot2.

## Data availability

ChIP-Seq and scRNAseq data were deposited in GEO under the accession nos. GSE280805 and GSE280804. The raw data associated to this study was deposited in BioStudies under accession link: https://www.ebi.ac.uk/biostudies/studies/S-BSST2084?key= 2ada598a-e585-4973-9479-d77256cb5ccd.

The source data of this paper are collected in the following database record: biostudies:S-SCDT-10_1038-S44319-026-00746-z.

## Peer review information

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

## Acknowledgements

We acknowledge previous members of the Estaras lab for discussions and experimental support, including Dr. Eleonora Stronati and Arantza Larrinaga-Zamanillo. Also, thanks to Dr. Simon, Dr. Kishore, and Dr. Garikipati's laboratories at Aging+Cardiovascular Discovery Center for discussions and feedback. Thanks to Dr. Kishore's lab for their generosity sharing reagents. Thanks to Alex Morris (Fox Chase Cancer Center) for initial analysis on scRNAseq datasets and discussion. Dr. Jonathan Whetstine's lab and the Genomics Facility at the Fox Chase Cancer Center for technical support for single-cell experiments. We acknowledge support from Fox Chase Molecular Modeling Facility, funded in part by the NIH Cancer Center Support Grant P30 CA006927. Thanks to Sanford Burnham Prebys Proteomics and Bioinformatics Core for help with the biotinylation assay and bioinformatics analysis, funded in part by the NIH Cancer Center Support P30 CA030199. This work was funded by the NIH/NICHD R01 HD106969 (to CE), NIH/NINDS R01NS119699 (to NA), NIH/NHLBI 5T32HL091804 (to EA and EM), NIH/NICHD F31HD113419 (to EA), and NIH/NIGMS T32 GM142606 (to AD).

## Author contributions

**Elizabeth Abraham**: Conceptualization; Data curation; Formal analysis; Funding acquisition; Validation; Investigation; Visualization; Methodology; Writing—review and editing. **Thomas Roule**: Data curation; Formal analysis; Investigation; Visualization. **Aidan Douglas**: Investigation. **Emily Megill**: Investigation. **Olivia M Pericak**: Investigation. **Jordan E Howe**: Formal analysis; Investigation; Visualization. **Carmen Choya-Foces**: Investigation. **Joanne F Garbincius**: Investigation. **Henry M Cohen**: Investigation. **Paula Roig-Flórez**: Investigation. **Mikel Zubillaga**: Investigation; Project administration. **Mark D Andrake**: Formal analysis; Investigation; Visualization. **Seonhee Kim**: Supervision. **John W Elrod**: Supervision. **Naiara Akizu**: Supervision. **Conchi Estaras**: Conceptualization; Resources; Data curation; Formal analysis; Supervision; Funding acquisition; Investigation; Visualization; Methodology; Writing—original draft; Project administration; Writing—review and editing.

Source data underlying figure panels in this paper may have individual authorship assigned. Where available, figure panel/source data authorship is listed in the following database record: biostudies:S-SCDT-10_1038-S44319-026-00746-z.

## Disclosure and competing interests statement

The authors declare no competing interests.

# Expanded View Figures

**Figure EV1.  Controls for *Yap1* cKO embryo analyses.**

(A) Gel shows expected bands from flox genotyping using a Yap1Flox:cre system. Adapted from Abraham et al, 2025. (B) Gels show the genotype of breeding partners. Note that only males carry the Sox2-cre allele to avoid maternal inheritance of Cre activity. (C) Same-day genotyping for flox and Cre for fresh-embryo sequencing was performed from the yolk sacs of 14 embryos, simultaneously isolated from 2 pregnant dams. Four controls, indicated in red triangles, and three *Yap1* cKO embryos (floxflox/cre +), shown in blue circles, were pooled and processed for scRNAseq. (D) Genotyping of SRY (sex identity) in the 14 embryos isolated for the experimental design of the scRNAseq experiment. (E) Violin plot of *Yap1* and *Wwtr1* (TAZ) from scRNAseq expression levels in all clusters comparing *Yap1* cKO to control. *Yap1* expression is significantly reduced in *Yap1* cKO cells across epiblast lineages, including epiblast (adjusted $p = 7.8 \times 10^{-58}$), primitive streak ($1.2 \times 10^{-20}$), nascent mesoderm ($4.2 \times 10^{-17}$), cardiac mesoderm ($1.2 \times 10^{-5}$), blood progenitors, and endoderm ($3.8 \times 10^{-4}$). Adjusted *p*-values were calculated using a Wilcoxon rank-sum test with Benjamini–Hochberg correction ($*p < 0.05$, $**p < 0.001$, $***p < 0.0001$). Each dot represents a single cell from E7 scRNAseq data. (F) Graphs show RT-qPCR analysis of *Yap1* and its target gene, Ccn2 (CTGF), in E7.5 *Yap1* cKO and control embryos ($n = 10$). Data are presented as mean ± SEM. Statistical analysis: Student's t-test, $**p = 0.0072$ and $***p < 0.0008$. (G) Graphs display cell cycle S and G2M scores in control and *Yap1* cKO embryos from scRNAseq analysis. Box-and-whisker plots indicate the median (center line), interquartile range (25th–75th percentiles; box), and minimum to maximum values (whiskers). Individual dots represent a single cell from E7 embryo scRNA-seq data. (H) Bright-field images of control and *Yap1* cKO E7 embryos. Graphs show cell number quantification per embryo (left) and the size of the epiblast (right) in control and *Yap1* cKO embryos ($n = 8$–10 embryos). Data are presented as mean ± SEM. Statistical analysis: Student's t-test. Scale bar 250 µm. (I) Single cell pathway analysis was applied to DEGs. Terms related to TGFb and Wnt signaling pathways significantly enriched (q-value > 1.4, adj. *p*-value < 0.05) in the epiblast are shown. (J) Full western blot of nuclear extracts of E7 embryos shown in Fig. 1G. C: control embryos and Y: *Yap1* cKO embryos. Red Arrows indicate bands shown in main Figure; SMAD2/3 (mw: 55 kDa), HISTONE H3 (mw: 15 kDa), GAPDH (mw: 37 kDa), B-CATENIN (mw: 90 kDa). (K) Western blot of whole embryo lysates of E7 control and *Yap1* cKO embryos. Pooled embryos numbers are indicated above each lane, along with the makers analyzed and on the right is the full blots. Red Arrows indicate bands that were cropped; SMAD2/3 (mw: 55 kDa), GAPDH (mw: 37 kDa), and B-CATENIN (mw: 90 kDa).

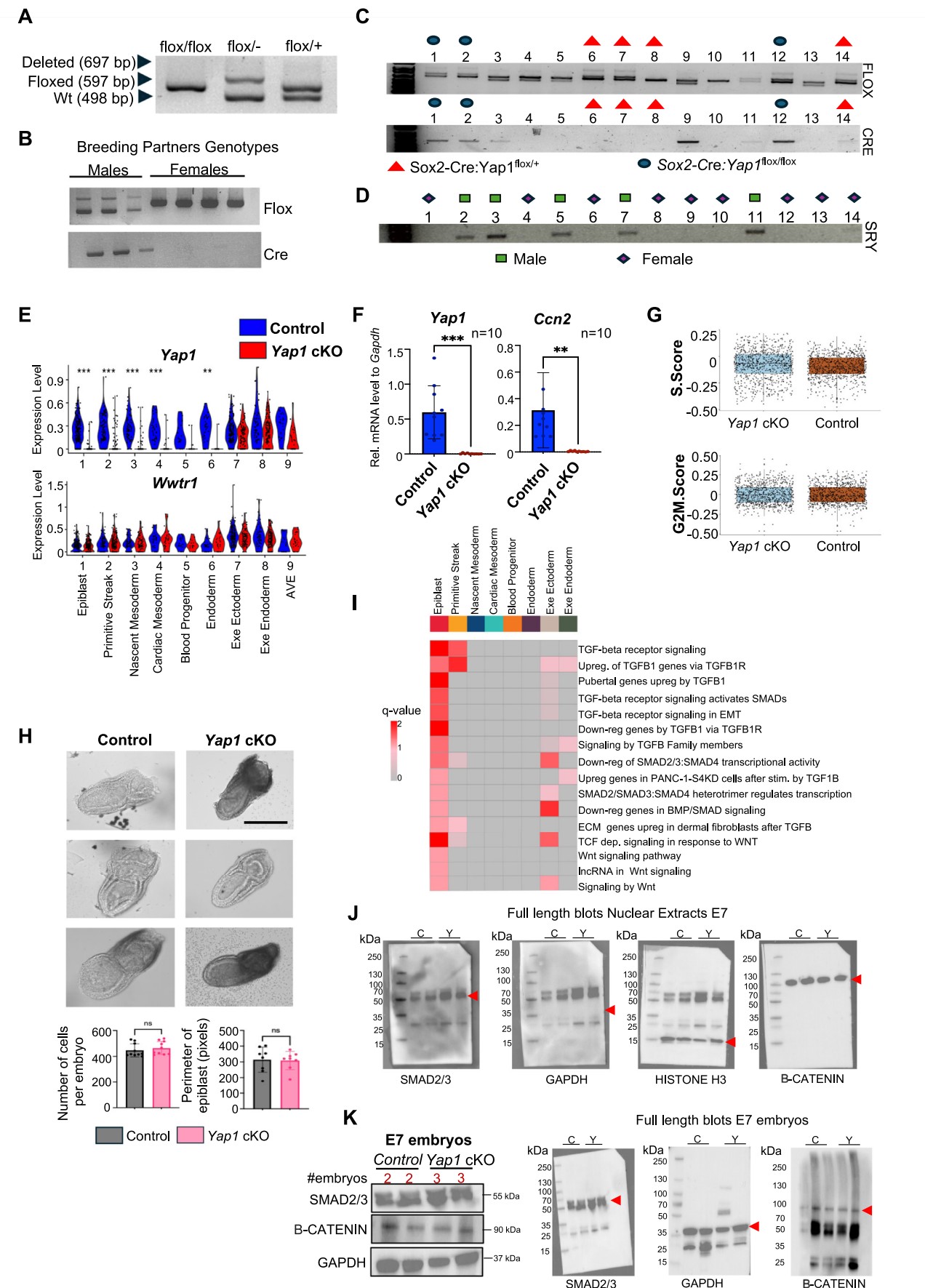

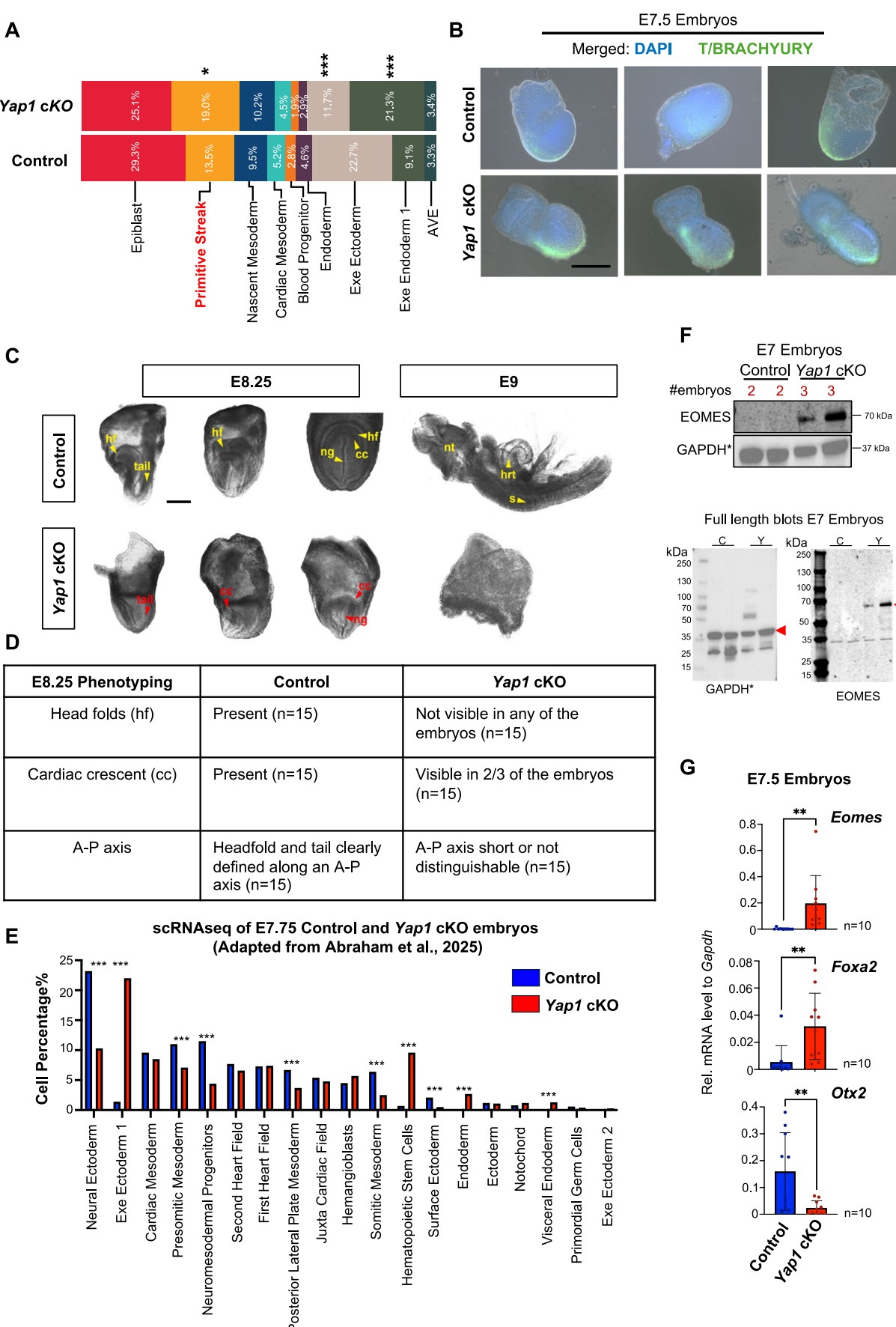

**Figure EV2.   Phenotyping analysis of *Yap1* cKO embryos reveals patterning defects.**

(A) Bar graph showing the percentage of cells assigned to each cluster of control and Yap1 cKO embryos. All populations are shown. Statistical analysis: Chi-test, * <0.05, ** <0.001,*** <0.0001. (B) Merged images of whole-mount immunostaining of BRACHYURY (T; green) and DAPI (blue) in E7.5 control and *Yap1* cKO embryos. The experiment was repeated three times with different litters with consistent results. See also Fig. 2B. Scale bar 250 μm. (C) Bright-field images of control and *Yap1* cKO at E8.25 and E9. hf: head fold, ng: neural groove, cc: cardiac crescent, nt: neural tube, hrt: heart, and s: somites. Scale bar 250 μm. (D) Table showing the phenotyping description of control and *Yap1* cKO embryos at E8.25 ($n = 15$). (E) Percentage of cells from E7.75 control and *Yap1* cKO scRNA-seq datasets previously published by our lab (Abraham et al, 2025). Clusters marked with an asterisk (*) are significantly different. Statistical analysis: Chi-test with Bonferroni correction run through 100 bootstrap iteration, ***$p < 0.0001$. (F) Western blot of whole embryo lysates of control and *Yap1* cKO at E7. Embryos pooled per lane are indicated, along with the markers. The uncropped blots are represented below. The red arrows highlight the molecular weight. C: control Y: *Yap1* cKO Eomes (mw: 70 kDa) and Gapdh (mw: 37 kDa) *Same western blot from Fig. EV1K. (G) qPCR of listed markers in E7.5 embryos ($n = 10$). Data are presented as mean ± SEM. Statistical analysis: Student's t-test, **$p = 0.0096$ (Eomes), 0.0068 (Foxa2), and 0.0089 (Otx2).

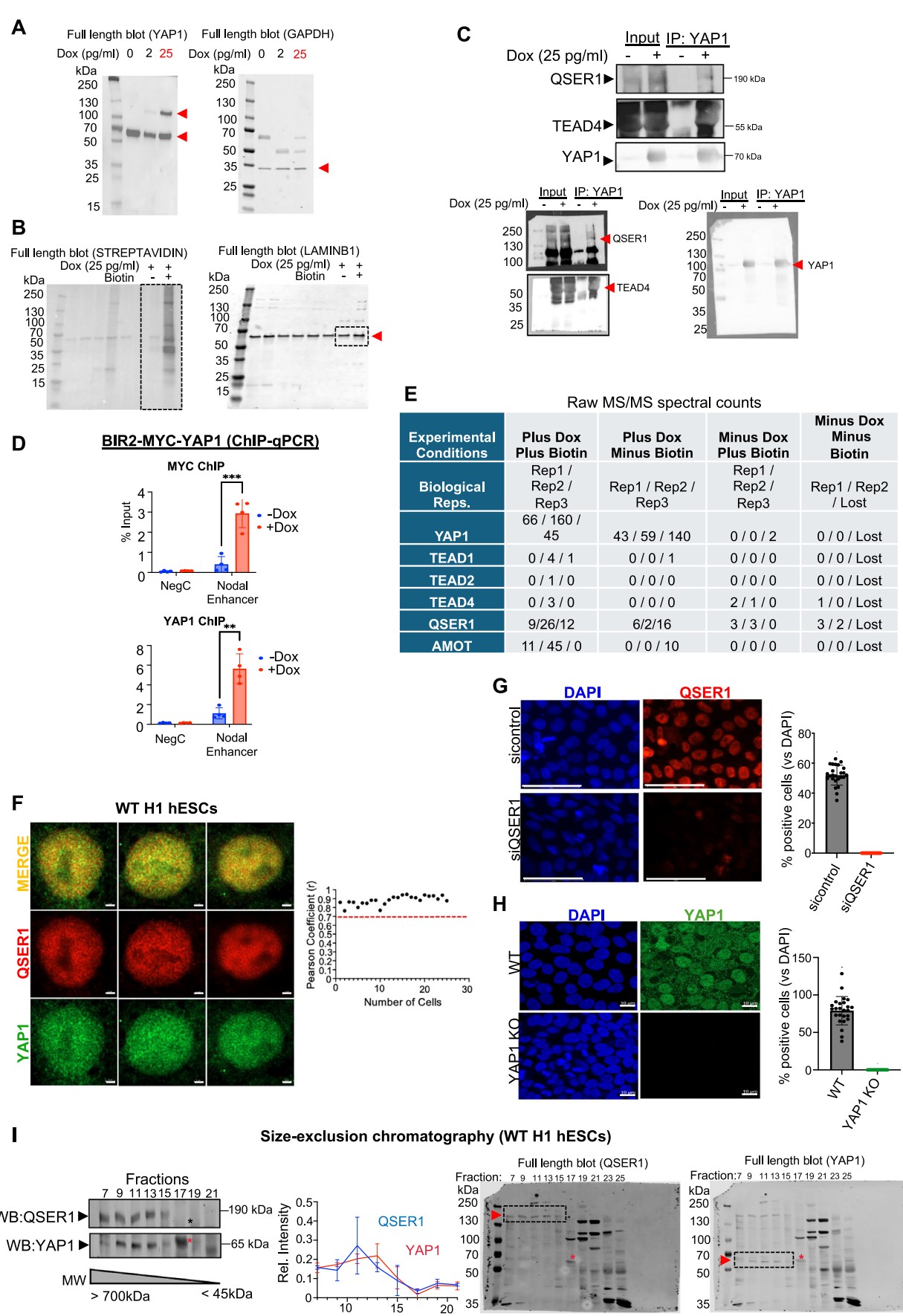

◄ **Figure EV3. YAP1 co-localizes with QSER1 in hESCs.**

(A) Uncropped western blot from Fig. 3B. Red arrowheads indicate bands shown in the main figure. (B) Uncropped western blot from Fig. 3C. The dotted square highlights the bands shown in the main figure. (C) Co-immunoprecipitation (Co-IP) experiment was performed in the doxycycline-inducible YAP1-Myc-Bir2 clonal hESC line in the presence and absence of Doxycycline, as indicated. The YAP1 antibody was used for immunoprecipitation, and western blot analysis was performed using a QSER1, TEAD4, and YAP1 antibodies. 10% of total lysate was loaded as input. Full western blot shown below. (D) ChIP-qPCR analysis was performed in the clonal YAP1-Myc-Bir2 hESC line, in the absence and presence of 25 pg/ml of Doxycycline (Dox). ChIPs using c-myc and YAP1 antibodies were carried out. The genomic regions analyzed are indicated at the bottom. NegC, negative control region ($n = 4$, independent biological replicates). Data are presented as mean ± SEM. Statistical analysis: Student's t-test, $^{**}p = 0.0056$ and $^{***}p = 0.0008$. (E) Table depicting experimental conditions used for the BioID2 assay and replicates. Raw MS/MS counts (without normalization) are shown for the indicated proteins in each replicate. Dox: doxycycline. (F) Immunostaining of endogenous YAP1 (green) and QSER1 (red) in WT hESCs. Representative 63x magnified images of single cells are shown. Scale bar 2 μm ($n = 25$). On the right, Pearson correlation coefficient (r) was calculated for YAP1 and QSER1 signals in each cell. A threshold of $r > 0.7$ (indicated by the red dashed line) shows a strong positive correlation. (G) Immunostaining of QSER1 (red) in sicontrol and siQSER1 in H1 hESCs and quantification of cells positive for QSER1, relative to DAPI (blue). Scale bar 50 μm (25 cells were counted across three images of one biological replicate). Data are presented as mean ± SEM. (H) Immunostaining of YAP1 (green) in WT and YAP1 KO H1 hESCs and quantification of cells positive for YAP1, relative to DAPI (blue). Scale bar 10 μm. (25 cells were counted across three images of one biological replicate). Data are presented as mean ± SEM. (I) Size-exclusion chromatography was performed on nuclear extracts from WT H1 hESCs. The elution fractions analyzed are indicated on the top. The approximate MW range covered by these fractions is indicated at the bottom. Western blot of QSER1 and YAP1 was performed. Relative intensity was quantified. Complete blots are shown on the right. * shows a band that appeared in both the QSER1 and YAP1 channels and was discarded from quantifications. Experiment was performed with two biological replicates. Data are presented as mean ± SEM.

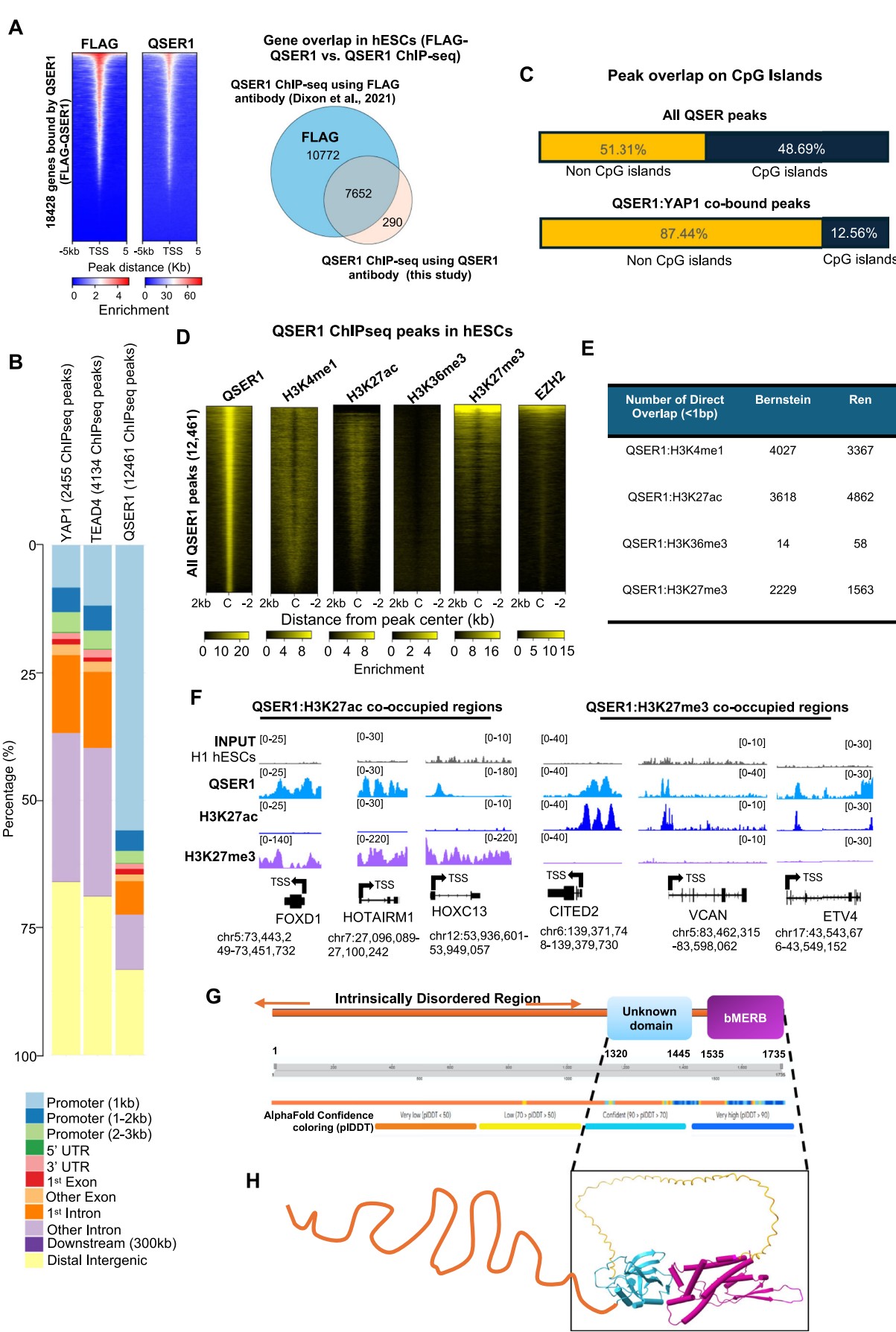

**Figure EV4. Genome-wide analysis of QSER1 occupancy in hESCs and protein structure.**

(A) Intersection analysis was carried out between our QSER1 ChIP-seq (using QSER1 antibody) and previously published Flag-tagged QSER1 ChIPseq in H1 hESCs (using FLAG antibody) (Dixon et al, 2021). Heatmaps and Venn Diagram show the binding correlation and number of co-bound genes in the two datasets. (B) Genomic distribution of QSER1, TEAD4, and YAP1 peaks in WT H1 hESCs from ChIP-seq datasets. (C) Graph shows the percentage of QSER1 peaks (all peaks) and QSER1:YAP1 co-bound peaks (199) associated to CpG islands in hESCs. (D) Heatmap shows correlation of QSER1 binding, indicated histone marks (source, ENCODE, Bernstein datasets), and EZH2. The heatmap is ranked by QSER1 peak signal and the number of peaks are shown (C = center of the peak, $+/-2$ kb). (E) Table shows the number of overlapping QSER1:H3K4me1, QSER1:H3K27ac, QSER1:H3K36me3, and QSER1:H3K27me3 peaks, genome-wide, using two different histone ChIP-seq datasets (Ren and Bernstein). (F) IGV genome browser captures show QSER1, H3K27me3, and H3K27ac at indicated genes TSS: transcription start site. (G) QSER1 predicted structure based on AF modeling. Most of the protein comprises an intrinsically disordered region (IDR), especially the first three quarters (residues 1 to 1319, orange). Following the large IDR, there are two folded domains. The first is a domain of unknown function (light blue) that is mostly composed of ß-strands, and the second has been designated as a "bivalent Mical/EHBP Rab binding" or bMERB domain (magenta, residues 1535–1735) and is largely helical with three ß-strands in the middle. These two folded domains are separated by 90 residues of a flexible disordered linker (also orange). Below the domain arrangement is a scale bar to denote residue numbers and a bar that represents the AlphaFold2 confidence score (pIDDT) with a color key below that shows that darker blue reflects higher confidence. (H) Schematic representation of IDR and AlphaFold predictions of QSER1 folded domain structures. Each is color-coded to match the domains shown above. While the relative orientation of the two domains is the most frequently predicted conformation, the linker between them is highly flexible.

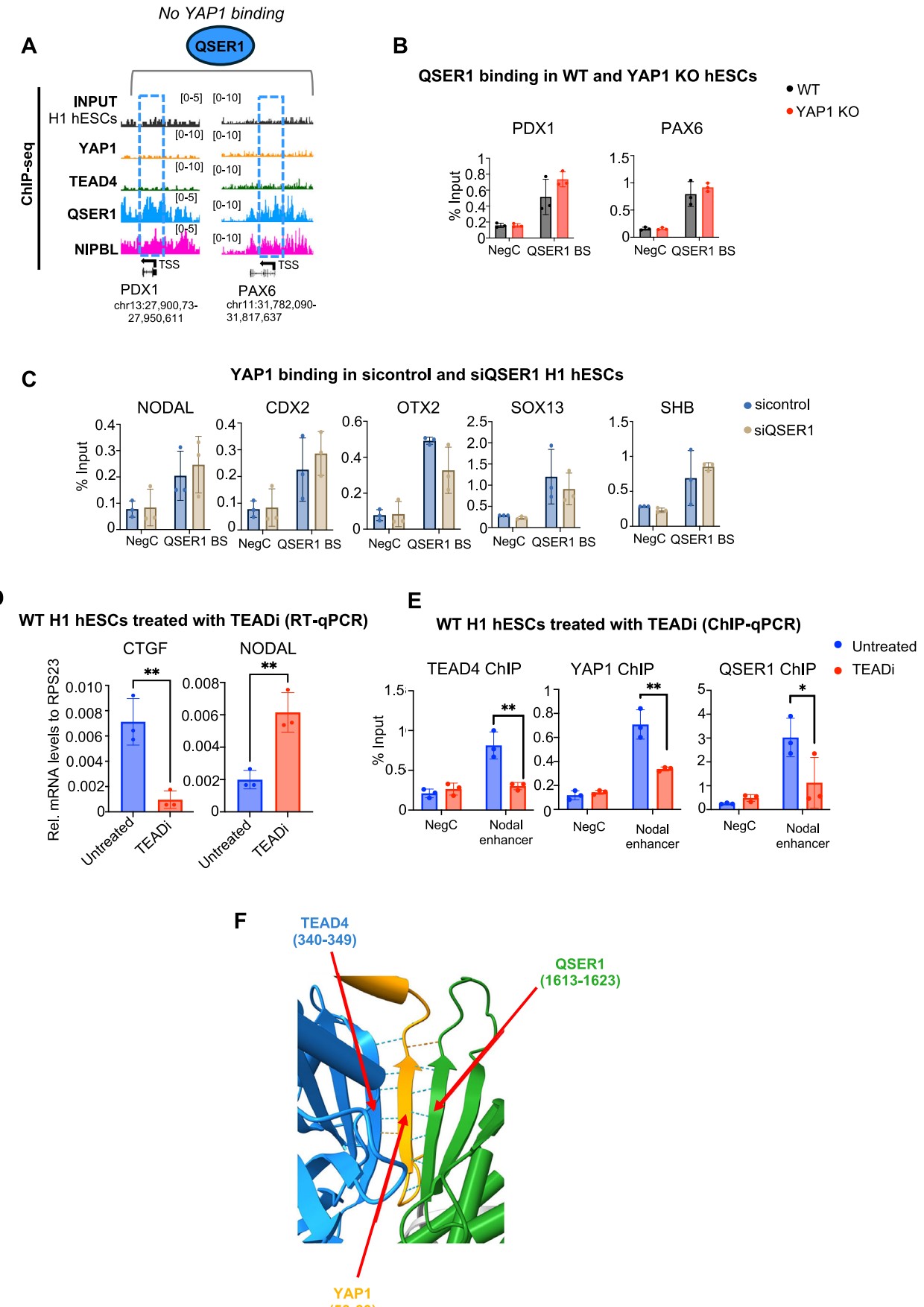

◀ **Figure EV5. QSER1 does not affect YAP1 binding to TEAD4.**

(A) IGV genome browser snapshots show more examples of distribution of QSER1, YAP1, TEAD4, and NIPBL on indicated genes. (B) Graphs show ChIP-qPCR analysis of QSER1 protein on the indicated genomic regions in WT and YAP1 KO hESCs. QSER1 BS: QSER1 binding site. NegC: Negative control region ($n = 3$, independent biological replicates). Data presented as mean ± SEM. Statistical analysis: Student's t-test. (C) Graphs show ChIP-qPCR analysis of YAP1 protein on the indicated genomic regions and conditions in sicontrol and siQSER1 conditions. NegC: Negative control region ($n = 3$, independent biological replicates). Data presented as mean ± SEM. Statistical analysis: Student's t-test. (D) RT-qPCR of gene expression of CTGF (downstream gene of the Hippo signaling pathway) and NODAL in WT H1 hESCs treated with or without 5 μM GNE-7883 TEAD inhibitor (TEADi) ($n = 3$, independent biological replicates). Data presented as mean ± SEM. Statistical analysis: Student's t-test, **$p = 0.0056$ (CTGF) and **$p = 0.0059$ (NODAL). (E) Graph of ChIP-qPCR of TEAD4, YAP1, and QSER1 at enhancer of the NODAL gene in untreated and TEADi treated cells. NegC: Negative control region ($n = 3$, independent biological replicates). Data presented as mean ± SEM. Statistical analysis: Student's t-test, *$p = 0.0148$, **$p = 0.0064$ (YAP1), and **$p = 0.0066$ (TEAD4). (F) Molecular modeling of TEAD4 (blue), YAP1 (orange), and QSER1 (green) using AlphaFold3 showing that YAP1 residues 50–60 are tightly bound to QSER1 residues 1613–1623 (7 hydrogen bonds) and TEAD4 residues 340–349 (5 hydrogen bonds, shown as dotted lines). Top ipTM scores for this complex are 0.68, reflecting a high confidence in the conformation of this model.

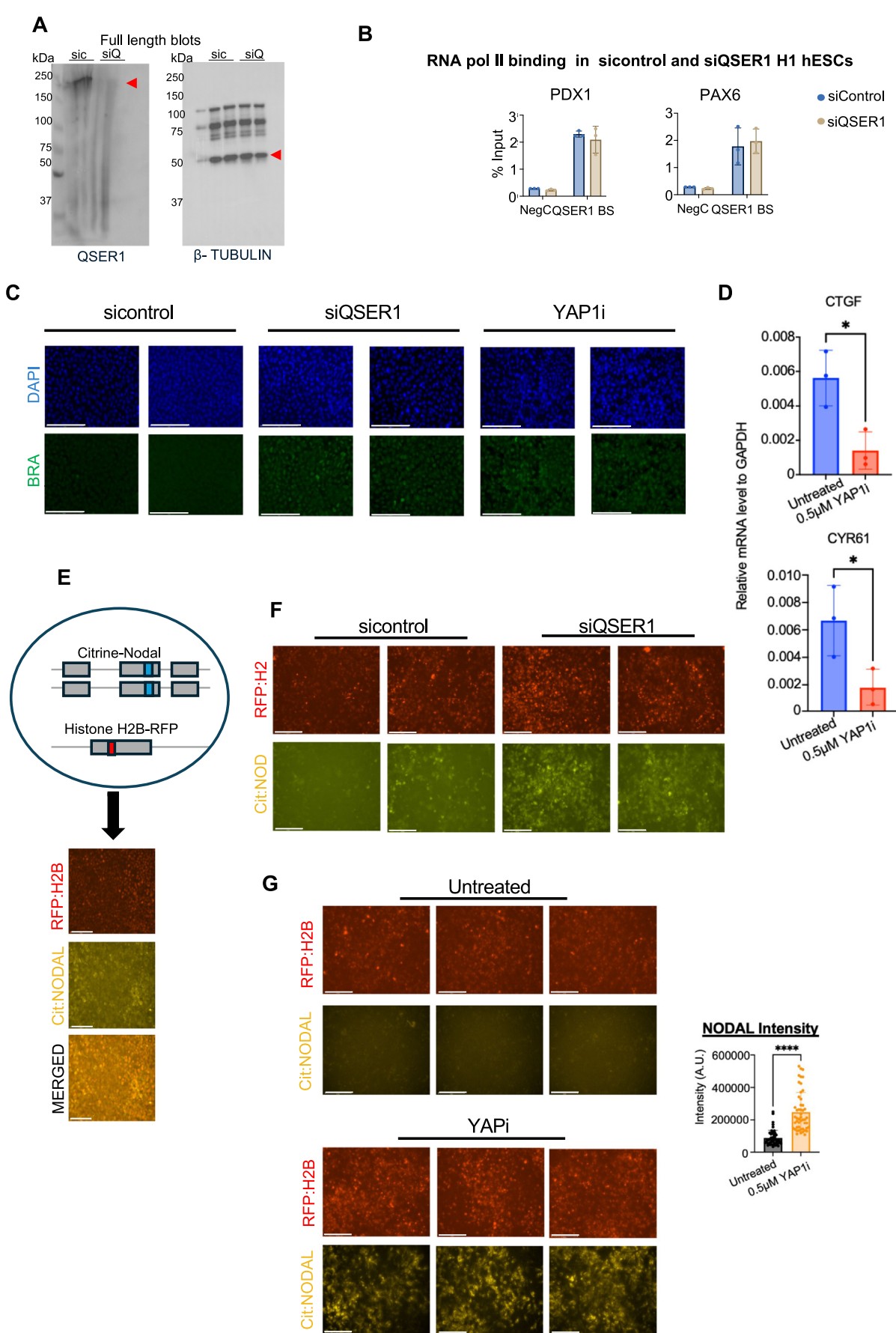

**Figure EV6.  QSER1 deletion partially phenocopies loss of YAP1 in hESCs.**

(A) Full uncropped blot of Fig. 6B. Blotted against QSER1 (mw: 190 kDa) and beta-TUBLIN (mw: 50 kDa). Red arrow indicates the band that was cropped. Sic: sicontrol and SiQ: siQSER1. (B) Graphs show ChIP-qPCR analysis of RNA polymerase II protein on the indicated genomic regions in sicontrol and siQSER1 hESCs. NegC: Negative control region and QSER1 BS: QSER1 binding site ($n = 3$, independent biological replicates). Data presented as mean ± SEM. Statistical analysis: Student's t-test. (C) Additional images of hESC treated with Activin and stained for BRA shown in Fig. 6E. (D) Graphs show RT-qPCR analysis of YAP1-target genes CTGF and CYR61 in hESCs untreated and treated with the YAP1 inhibitor (YAPi) DASATINIB for 72 h treatment ($n = 3$, independent biological replicates). Data presented as mean ± SEM. Statistical analysis: Student's t-test, CTGF: *$p = 0.0367$ and CYR61: *$p = 0.0490$. (E) Scheme of the Nodal-citrine: H2B-RFP hESC construct with representative fluorescent images of hESCs under basal conditions. (F) Additional images of hESC treated with Activin and NODAL shown in Fig. 6G. (G) Representative images of untreated and YAP1i treated hESCs treated with Activin (50 ng/mL) for 48 h, NODAL protein expression was visualized using an engineered dual-reporter line expressing NODAL-citrine and H2B-RFP (Liu et al, 2022). Scale bar, 125 μm. Graph shows quantification of fluorescence intensity per cell (50 cells were quantified from three biological replicates). Data presented as mean ± SEM. Statistical analysis: Student's t-test, ****$p < 0.0004$.

