## [Peer Review File · EMBO Reports]

YAP1 and QSER1 are Key Modulators of Embryonic Signaling Pathways in the Mammalian Epiblast

Elizabeth Abraham, Thomas Roule, Aidan Douglas, Emily Megill, Olivia Pericak, Jordan Howe, Carmen Choya-Foces, Joanne Garbincius, Henry Cohen, Paula Roig-Florez, Mikel Zubillaga, Mark Andrade, Seonhee Kim, John Elrod, Naiara Akizu, and Conchi Estarás

Corresponding author(s): Conchi Estarás (conchi.estaras@temple.edu)

Review Timeline:

Submission Date:	3rd Nov 24
Editorial Decision:	16th Dec 24
Revision Received:	13th Jun 25
Editorial Decision:	8th Aug 25
Revision Received:	26th Sep 25
Editorial Decision:	12th Jan 26
Revision Received:	25th Jan 26
Accepted:	2nd Mar 26

Editor: *Martina Rembold*

Transaction Report:

Dear Dr. Estarás

Thank you for the submission of your manuscript to our journal. I apologize for the delay in handling your manuscript, but we have now received the full set of referee reports that is copied below.

As you will see, while the referees agree that the study is potentially interesting, they also all point out that it requires significant revision before it can be considered for publication here. The referees consider the functional link between YAP1 and QSER and to Nodal signalling not fully convincing at this stage of the analysis. They also have a number of significant technical concerns, such as the quality and analysis of the ChIP and BioID2 data. In addition, the potential compensation of YAP1 function by TAZ as well as the contribution of TEADs need to be considered.

From the referee comments it is clear that, as it stands, the technical quality of the study is low/unacceptable and publication of the manuscript in our journal can therefore not be considered at this stage. On the other hand, given the potential interest of your findings, I would like to give you the opportunity to address the concerns and would be willing to consider a revised manuscript with the understanding that the referee concerns must be fully addressed and their suggestions (as detailed above and in their reports) taken on board.

Should you decide to embark on such a revision, acceptance of the manuscript will depend on a positive outcome of a second round of review and I should also remind you that it is EMBO reports policy to allow a single round of revision only and that, therefore, acceptance or rejection of the manuscript will depend on the completeness of your responses included in the next, final version of the manuscript.

We realize that it is difficult to revise to a specific deadline. In the interest of protecting the conceptual advance provided by the work, we recommend a revision within 3 months (March 16, 2025). Please discuss the revision progress ahead of this time with the editor if you require more time to complete the revisions.

I am also happy to discuss the revision further via e-mail or a video call, if you wish.

*****IMPORTANT NOTE:

We perform an initial quality control of all revised manuscripts before re-review. Your manuscript will FAIL this control and the handling will be delayed IN CASE the following APPLIES:

- 1) A data availability section providing access to data deposited in public databases is missing. If you have not deposited any data, please add a sentence to the data availability section that explains that.
- 2) Your manuscript contains statistics and error bars based on $n=2$. Please use scatter blots in these cases. No statistics should be calculated if $n=2$.

When submitting your revised manuscript, please carefully review the instructions that follow below. Failure to include requested items will delay the evaluation of your revision. *****

- 1) a .docx formatted version of the manuscript text (including legends for main figures, EV figures and tables). Please make sure that the changes are highlighted to be clearly visible.
- 2) individual production quality figure files as .eps, .tif, .jpg (one file per figure). Please download our Figure Preparation Guidelines (figure preparation pdf) from our Author Guidelines pages <https://www.embopress.org/page/journal/14693178/authorguide> for more info on how to prepare your figures.
- 3) a .docx formatted letter INCLUDING the reviewers' reports and your detailed point-by-point responses to their comments. As part of the EMBO Press transparent editorial process, the point-by-point response is part of the Review Process File (RPF), which will be published alongside your paper.
- 4) a complete author checklist, which you can download from our author guidelines (<<https://www.embopress.org/page/journal/14693178/authorguide>>). Please insert information in the checklist that is also reflected in the manuscript. The completed author checklist will also be part of the RPF.
- 5) Please note that all corresponding authors are required to supply an ORCID ID for their name upon submission of a revised manuscript (<<https://orcid.org/>>). Please find instructions on how to link your ORCID ID to your account in our manuscript

tracking system in our Author guidelines

(<<https://www.embopress.org/page/journal/14693178/authorguide#authorshipguidelines>>)

6) We replaced Supplementary Information with Expanded View (EV) Figures and Tables that are collapsible/expandable online. A maximum of 5 EV Figures can be typeset. EV Figures should be cited as "Figure EV1, Figure EV2" etc... in the text and their respective legends should be included in the main text after the legends of regular figures.

7) The accession numbers and database should be listed in a formal "Data Availability " section (placed after Materials & Method) that follows the model below (see also <<https://www.embopress.org/page/journal/14693178/authorguide#dataavailability>>).

Please note that we also require URLs that resolve directly at the deposited datasets. Please also provide public access to the mass spectrometry dataset.

Data availability

Additional information on source data and instruction on how to label the files are available <<https://www.embopress.org/page/journal/14693178/authorguide#sourcedata>>.

10) Figure legends and data quantification:

- the name of the statistical test used to generate error bars and P values,
 - the number (n) of independent experiments (please specify technical or biological replicates) underlying each data point,
 - the nature of the bars and error bars (s.d., s.e.m.)
- If the data are obtained from n {less than or equal to} 5, show the individual data points in addition to the SD or SEM.
- If the data are obtained from n {less than or equal to} 2, use scatter blots showing the individual data points.

11) Our journal encourages inclusion of *data citations in the reference list* to directly cite datasets that were re-used and obtained from public databases. Data citations in the article text are distinct from normal bibliographical citations and should

directly link to the database records from which the data can be accessed. In the main text, data citations are formatted as follows: "Data ref: Smith et al, 2001" or "Data ref: NCBI Sequence Read Archive PRJNA342805, 2017". In the Reference list, data citations must be labeled with "[DATASET]". A data reference must provide the database name, accession number/identifiers and a resolvable link to the landing page from which the data can be accessed at the end of the reference. Further instructions are available at <<https://www.embopress.org/page/journal/14693178/authorguide#referencesformat>>.

12) All Materials and Methods need to be described in the main text using our 'Structured Methods' format. According to this format, the Methods section includes a Reagents and Tools Table (listing key reagents, experimental models, software and relevant equipment and including their sources and relevant identifiers) followed by a Methods and Protocols section describing the methods, ideally using a step-by-step protocol format. The aim is to facilitate adoption of the methodologies across labs. Please download and fill our Reagents and Tools Table template (.docx), which you can find in our author guidelines:

13) As part of the EMBO publication's Transparent Editorial Process, EMBO Reports publishes online a Review Process File to accompany accepted manuscripts. This File will be published in conjunction with your paper and will include the referee reports, your point-by-point response and all pertinent correspondence relating to the manuscript.

Yours sincerely,

=====

Referee #1:

Review Abraham et al.

The manuscript by Abraham and colleagues addresses the functions of the transcriptional regulator YAP1 in the gastrulation stage mouse embryo. The authors apply mouse genetics to conditionally delete YAP1 exclusively in the epiblast, followed by the analysis of the resulting phenotype by scRNA-seq. Molecular approaches are used to identify interacting protein partners, and functional roles of YAP1 and the associated protein QSER1 on chromatin.

The topic of this study, YAP1 functions in pluripotent cells and during early differentiation, was previously already addressed, including studies by the same group (Hsu et al. 2018; Estaras et al., 2017; Estaras et al. 2015, recent BioRxiv-manuscript by Abraham et al.). In the current work, the authors aim for providing a new angle to YAP functions by the analysis of cooperative functions of YAP with the QSER1, a putative negative regulator of DNA methylation, and their effects on Nodal/Smad2/3 signalling. Certainly, to date the role of HIPPO/YAP/TAZ signalling during gastrulation in mouse still requires additional studies and is far from understood. With their current study the authors try to add another piece in the puzzle, however, the presented manuscript presents with several conceptual and experimental weaknesses that require major revision and clarification to convincingly allow for the conclusions drawn by the authors.

Major concerns:

1. One major conclusion of the manuscript is the negative regulation of Nodal-signalling levels by YAP1. Accordingly, the

authors aim to demonstrate the hyperactivation of Nodal-signalling in YAP1 conditionally deleted embryos. However, the data to show that Nodal-signalling levels are increased are not convincing. Please show cumulative expression of Nodal in epiblast cells of the scRNA-seq analysis (Fig. 1F can't be read/interpreted), such as by violin-plots. Fig. 1I showing the RNAscope of Nodal doesn't show the expected expression of Nodal, so it is not clear what generates the signal, that appears rather unspecific. The Western Blot of nuclear Smad2/3 doesn't allow for the conclusion that signalling levels are elevated since it lacks critical controls, such as for the purity of the nuclear purification vs. the cytoplasm. Most likely the cytoplasmic pool exceeds the nuclear, so that impurities in the purification might disturb the outcome (H3 is reduced in the YAP-KO). The authors should use pSMAD2 as readout and normalize to a) a loading control and b) total SMAD2. This seems the more common way to test for Nodal/SMAD2 pathway activity. The pathway analysis (Fig. 1L) is uninformative and doesn't allow to look at the actual data. Do the authors expect the reader to decode the different numbers?

2. Most importantly, it remains unclear why the authors used T/Brachyury as readout for Nodal signalling. Most likely, T/Brachyury isn't directly regulated by increased levels of Nodal, but rather by canonical Wnt-signalling downstream of Wnt3a. Potential readouts for Nodal would be Smad2/3 regulated genes of anterior primitive streak derivatives, such as DE, node, notochord, prechordal plate. These would include Eomes, Foxa2, Cer1, Lhx1, Gsc, and others. The elevated expression of T/Brachyury rather hints at elevated Wnt-levels, as already previously reported by the same authors. I would expect that increased Nodal levels would also impact the proportion of different cell types in the gastrulating embryo, which, according to the authors, is not the case. What is the morphogenetic defect in the YAP1 cKO embryos that indicate hyperactivation of Nodal-signalling. In the discussion the authors compare the lethal phenotype of YAP1 mutants, with the conditional mutants of the current study. However, no data are shown concerning the later phenotype. Does it reflect the hyperactivation of Nodal-signalling, as described for TGIF1/2 double mutants? Please clarify and add the data about later occurring phenotypes, lethality etc, with a specific view on Nodal signalling readouts (loss/specification of certain cell types)

3. The functional link between YAP1 and QSER seems weak and difficult to interpret. Can the authors give further insight on what lead to their decision to select QSER1 among the many significant hits of the biotinylation assay as an interesting target of further analyzation? It is stated that „it has been only partially characterized" as the only reason justifying their further research. Have the authors looked into any other co-factors of Myc-YAP1-Bir2 or was their decision influenced by reasons not explained in the manuscript?

4. The quality of presented ChIP data and the data analysis is poor. Why are QSER1 peaks not shown in Fig. 3F, when all other marks and EZH2 are shown in relation to QSER1 peaks? Please always indicate number of sites at the y-axis (No. of QSER1 peaks in this example). The overlap with H3K4me1 and H3K27ac seems very weak/minor. This would need a backup on genome browser views to convincingly demonstrate that there is significant enhancer-binding of QSER1. What is the regulation by QSER1? Is this related to methylation? Can this be shown? How do the authors envision how YAP1 could regulate QSER1 binding in cases where they don't overlap on the same regions? There only seems a very loose interaction, which is only shown in 2 examples (please also read below).

Additional issues:

For future work, may I suggest to use the annotation of line numbers which massively help during reviewing.

1. At several places I suggest re-phrasing to match commonly used nomenclature, or clarify phrasing

"whole-body YAP1 knockout" (constitutive gene-deletion),

"cYAP1KO" (YAP1cKO),

"mid-end gastrulation stage (E8)": This is incorrect! Gastrulation proceeds much longer until the formation of the body axis is completed (rather E12)

"spatially and temporally differentiates": unclear what this means

2. Please reference the Sox2:Cre deleter strain in the text. To my knowledge this is a transgenic strain, so that the nomenclature should reflect this (Sox2Cre/+ is incorrect). Also, the nomenclature of the alleles in Fig.1A and text might be misleading. YAP1 is always deleted by the Sox2:Cre deleter in the male, so that the male mice are heterozygous, and most likely also the extraembryonic tissues in YAP1flox/flox, Sox2 +/+ embryos, if there is not wt allele present. This might be important since there are effects on extraembryonic tissues that may result from haploinsufficiency.

3. Fig. 1E: Please state absolute No. of DEGs.

4. Fig. 2B: The quantification of fluorescent intensities is surely problematic. Generally, the brightness in mutants seems elevated, also in regions that lack T/Brachyury in the anterior. How can this be explained. How were the images normalized?

5. Is the fusion protein of YAP1-Bir functional? Was this experimentally tested? It is surprising, that no TEAD was found in the MS analysis. Is this due to steric hindrance? Please clarify. Experiments could involve co-IP between Myc-YAP1-Bir2 and TEAD. Is dox-inducible Myc-YAP1-Bir2 expressed at comparable levels in comparison to endogenous YAP1?

6. Please explain the rationale for RNA-PolIII ChIP on enhancers. In the introduction, it is stated that YAP1 and QSER control RNA-PolIII activity. Activity was not checked and tested.

7. In the text and in the Volcano Plot of MS data, CTNBL1 is mentioned and shown. To my knowledge this is NOT beta-Catenin, and not involved in canonical Wnt-signalling as stated in the text.

8. Fig. 3D: Why are there no proteins detected on the left side of the volcano plot? Is this a technical artefact? It is rather unusual that there is no enrichment in the negative control?

9. Figure S3B: the label of y-axis is "genes, the label of x-axis is "TSS" and gene distance (bp). I guess that all these labels need correction: regions (indicate numbers), peak summit, peak distance in kb. Are all regions ranked in the same order? If so, this would indicate a remarkable similarity of both ChIP-experiments. Please clarify.

10. Figure S3D: It is rather unclear what this figure represents.
11. Figure S3E: To spread ChIP peaks over 20 kb(!!) is unusual and doesn't allow to interpret where the peaks are located on genomic scale and if they overlap. Indicate number of regions shown (y-axis) and include QSER1 and YAP1 in heatmap representation.
12. Figure 3H: What is the value on the x-axis? Is this FDR-adjusted p-value, q-value, or indeed FDR? Please indicate No. of positive terms per pathway. The representation seems unusual. Why do authors use different representation for Figs. 1L and 3H?
13. The PEE element of Nodal was defined in mouse, as referenced in the text. However, the authors use hESCs. Please adjust the reference, and/or show that the indicated PEE is synonymous in human.
14. Text and Figures occasionally differ: Fig. 4A and text don't fit. Figure shows SMAD2, but text mentions INHBA.
15. The author analyze QSER1 binding in YAP1-KO via ChIP-qPCR focusing on two genes for each previously defined group (Fig. 4B). It is stated that „199 out of the 1,192 genes show a perfect overlap of QSER1 and YAP1 binding at the same regulatory region", Can the authors show more than 2 out of these 199 genes that show the same decrease of QSER1-binding upon YAP1-KO if they want to make the claim that YAP1 controls DNA accessibility (I assume they mean DNA binding) of QSER1 on coregulated genes.
16. The authors state that a variety of genes related to TGF-beta signaling are co-bound by YAP1 and QSER1 however it remains unclear, whether these genes are actually regulated by these two proteins (eg. SMAD3, ACVR2A) or however play any physiological role in early development of mammals at all (eg. INHBA). The GO pathway analysis depicted in Fig. 3H does not give any further insights into this issue since among the enrichment in TGF-beta/Nodal and Wnt-signalling are completely unrelated terms like „ionotropic glutamate receptor" or „angiogenesis".

In summary, the current manuscript requires major revisions, that include both, major text revisions and corrections, and also a substantial revision on experiments and data analysis to substantiate the major claims of the study, namely increased Nodal-signalling levels and functional interactions with QSER1.

Referee #2:

The manuscript entitled "YAP1 and QSER1 are Key Modulators of Embryonic Signaling Pathways in the Mammalian Epiblast" by Abraham et al. investigates the function of the mouse and human YAP1 protein in the epiblast and pluripotent stem cell differentiation. The authors perform single cell RNAseq and can show that YAP1 is a major regulator of Nodal signals and use conditional mutation in the epiblast to infer expanded expression of primitive streak markers. Conversely in human pluripotent stem cell derived gastruloid cultures elevated mesodermal differentiation is observed. Molecularly, the study shows that YAP1 acts both as activator and repressor and specifically interacts with QSER1 on enhancers. This interaction is independent and distinct from previously reported QSER1 function to restrain DNA methylation on Polycomb sites. Biochemical interaction is demonstrated by engineering a Bio2-tagged YAP1 and mass spectrometric identification of streptavidin bead bound proteins from nuclear extracts. The study is of interest for researchers in early mammalian development and cell signaling and contains data of high quality. The main advance is of extending of YAP1 function in early embryonic lineages and demonstrating a requirement in the epiblast which is distinct to YAP1 function in TE specification. Several points should be addressed to improve the impact and clarity of the study.

Specific points

1. The introduction raises the question of an involvement of Hippo signaling in cell growth and organ size control. The current version seems to rule out embryonic growth regulation but does not consider TAZ (*Wwtr1*) which could provide a redundant function. Although, a *Wwtr1* mutation in mice is compatible with epiblast development the absence of a discussion is surprising as TAZ is a separate gene but the introduction would make one believe it is a synonym. It would be interesting to know to what extent the absence of YAP1 would be compensated by TAZ or if there is reciprocal regulation or dependence. A few sentences should be added to the discussion to orient the reader and possibly the RNAseq data might be interrogated for expression changes in *Wwtr1*.
2. The authors mention that non-cell autonomous effects in YAP1 conditional mutant epiblast might also affect PS formation. It would be interesting to discuss if AVE development is affected - maybe secondarily from signals from the epiblast. Absence of the AVE would likely also explain expanded posterior fates and elevated signaling as this becomes a major source of signal inhibition in the anterior region. It would be interesting to see if scRNAseq provides evidence for AVE formation or a AVE defect can be ruled out on grounds of embryonic timing.

Minor points

- a) Results page 3 bottom: The authors argue that human pluripotent stem cells and mouse epiblast are comparable. A certain equivalence indeed exists but this would overlook substantial differences in streak formation, eg. in human there seems no DVE-AVE but early extraembryonic mesoderm formation. Also the human epiblast separates from the amniotic ectoderm which is not formed in mice until after gastrulation. Maybe a brief description of the relevant signaling events shared could be informative for the reader.

b) Results page 4, second para: The absence of TEAD(1-4) in the masspec data is surprising. Is it clear if the authors' methodology could recover TEAD interactions? It would be interesting to compare other biochemical studies that have identified TEAD proteins as interactors to consider differences - did the nuclear lysates (input) used by the authors would contain TEAD proteins and these were not biotinylated?

c) Discussion, end of the first line: "aka" could be replaced by "and"

Referee #3:

The manuscript by Abraham et al. reports the genetic interaction between YAP1 and QSER1 in modulating key signaling pathways (such as nodal) in mammalian epiblast.

Overall the paper is interesting for two main reasons: 1) as the authors mention, the role of YAP is extremely tissue-specific as YAP can control a plethora of diverse target genes and 2) QSER1 is a relatively poorly characterized modulator of DNA methylation.

However, I believe that the paper can be strengthened by addressing some key technical and biological questions:

- The authors often refer to YAP1 as if it is acting independently of TEADs. I do believe this is a key point. I suggest the author to test some of their phenotypes and transcriptional readouts with TEAD inhibitors. Similarly, I would encourage the authors to add CHIP-seq tracks for TEAD in their IGV plots for completeness. It would not be surprising that, even if the observed regulation of Nodal is TEAD independent, still TEAD would act as an anchor for YAP on chromatin
- Along those lines, the authors should report if canonical target genes of hippo signaling are modulated in YAP cKO cells.
- The authors identify QSER1 from a BioID2-MS experiment. On one side, the cutoff used for statistically significant enrichment in figure 2D seems quite lenient ($-\log_{10}pvalue > 1$). On the other, and most importantly, the data distribution seems highly asymmetric (quite odd if the analysis was performed using a standard "differential abundance" method). I would suggest the authors to add in the Material and Methods the computational analysis used for Mass Spectrometry data.
- Related to the previous point, I cannot find a validation of the interaction between YAP1 and QSER1. A simple co-IP from nuclear fraction might strengthen the evidence of the interaction (given the small number of co-occupied genomic regions). It will also overcome the claimed technical limitations of BioID2 in detecting the interaction with TEADs.
- The authors should show the absolute numbers of cells for each cell types retrieved by scRNA-seq. Moreover, since the cell annotation was done by manually curated list of markers, plots demonstrating selective expression of such markers should be included in the manuscript.
- At the end of the 4th section of the results the authors claim that YAP1 prevents epiblast differentiation by restricting Noval and WNT pathways. I would argue that, in absence of functional data (i.e. rescue of phenotypes by genetically/chemically perturbing Nodal/WNT), their data are correlative. I would rather tone down such sentence.

Minor points:

- The word "instead" in row 8 of the abstract rows 8 to me is not the best to introduce the sentence: since the sentence in row 8 is not in contrast to the previous one
- In page 2, a reference should be assigned to the study reporting that mice *Alfp-Cre YAP fl/fl* in liver reach normal organ size.
- I am not sure I would use the word "unbiased screening assay" for a BioID2-MS experiment. It could simply being referred to a proteomic experiment.

Referee #1:

Review Abraham et al.

The manuscript by Abraham and colleagues addresses the functions of the transcriptional regulator YAP1 in the gastrulation stage mouse embryo. The authors apply mouse genetics to conditionally delete YAP1 exclusively in the epiblast, followed by the analysis of the resulting phenotype by scRNA-seq. Molecular approaches are used to identify interacting protein partners, and functional roles of YAP1 and the associated protein QSER1 on chromatin.

The topic of this study, YAP1 functions in pluripotent cells and during early differentiation, was previously already addressed, including studies by the same group (Hsu et al. 2018; Estaras et al., 2017; Estaras et al. 2015, recent BioRxiv-manuscript by Abraham et al.). In the current work, the authors aim for providing a new angle to YAP functions by the analysis of cooperative functions of YAP with the QSER1, a putative negative regulator of DNA methylation, and their effects on Nodal/Smad2/3 signalling. Certainly, to date the role of HIPPO/YAP/TAZ signalling during gastrulation in mouse still requires additional studies and is far from understood. With their current study the authors try to add another piece in the puzzle, however, the presented manuscript presents with several conceptual and experimental weaknesses that require major revision and clarification to convincingly allow for the conclusions drawn by the authors.

We appreciate the reviewer's careful consideration of our prior studies and the recognition that the role of HIPPO/YAP1 signaling during gastrulation remains incompletely understood. In the revised manuscript, we have clarified and expanded on the novel aspect of our study, the role of YAP1 in the epiblast, and the cooperative function between YAP1 and QSER1 in regulating expression of genes involved in gastrulation signaling. In addition, we have strengthened the presentation of our experiments in the Figures and extensively rewritten sections of the results to more clearly support the conclusions drawn. We believe the changes we incorporated in response to the reviewers' suggestion strengthen our study.

Major concerns:

1. One major conclusion of the manuscript is the negative regulation of Nodal-signalling levels by YAP1. Accordingly, the authors aim to demonstrate the hyperactivation of Nodal-signalling in YAP1 conditionally deleted embryos. However, the data to show that Nodal-signalling levels are increased are not convincing. Please show cumulative expression of Nodal in epiblast cells of the scRNA-seq analysis (Fig. 1F can't be read/interpreted), such as by violin-plots.

We acknowledge that some aspects of the data presentation related to Nodal signaling regulation—particularly in Figure 1—could have been clearer and more accurately represented in our initial submission. In response, we have carefully revised **Figure 1** to strengthen both the experimental

evidence and the clarity of its presentation. We highlight below the key improvements included in the revised version:

- Replaced UMAPs with violin plots that include expression levels of Nodal and other relevant genes, along with statistical comparisons across all relevant populations (**Figure 1E**).
- Included total Smad2/3 protein levels (nuclear and total) alongside GAPDH and Histone H3 controls (**Figure 1H**).
- Replaced the representation of the Single-Cell Pathway Analysis (SCPA) to clearly highlight Nodal and Wnt pathway activities in the epiblast of cKO compared to controls (**Figure 1G**).

Overall, the revised version presents convincing evidence from three independent approaches—scRNA-seq, RT-qPCR, and WB on embryos at E7—demonstrating increased Nodal signaling genes upregulated in *Yap1* cKO embryos. In the revised version, we also emphasize that direct targets of SMADs, such as *Id1*¹⁻³ and *Nanog*⁴⁻⁶, are overexpressed in the epiblast of *Yap1* cKO embryos (**Revised Figure 1 and Results section; page 5, lane 20-25**).

1.2. Fig. 1I showing the RNAscope of Nodal doesn't show the expected expression of Nodal, so it is not clear what generates the signal, that appears rather unspecific.

We agree with the reviewer's concern. We observed a consistent increase in signal intensity in the KO embryos across different experimental replicates, which initially led us to believe the RNAscope signal was specific. To address this concern, we consulted directly with the RNAscope technical support team. They recommended an updated protocol and suggested switching from fluorescence to a colorimetric detection method to enhance spatial resolution and minimize background. After implementing the colorimetric approach, with guidance from the regional field scientist, we observed a more regionally restricted Nodal signal in WT embryos, as expected. However, in the *Yap1* cKO embryos, the signal remained diffuse and widespread, with increased intensity and ectopic staining in the extraembryonic tissues (**Reviewer Figure 1**). Despite these adjustments, we were unable to confidently improve the specificity or interpretability of the staining. Given these limitations, and to maintain the rigor of the manuscript, we have chosen to remove the RNAscope data from the final version.

We believe this does not alter the conclusions presented in the figure, as the upregulation of *Nodal* is consistently supported by RT-qPCR and scRNA-seq analyses, both of which show increased *Nodal* mRNA levels in the epiblast of YAP1-deficient embryos.

Reviewer Figure 1:

(A) RNAscope ISH analysis of Nodal probe in control and *Yap1* cKO embryos using a chromogenic red assay. This detection method provides bright, permanent red dots and high contrast with high sensitivity, compared to fluorescent methods. Compared to control embryos, which show Nodal staining from the primitive streak (PS) to the distal epiblast, *Yap1* cKO embryos display widespread signal that seems nonspecific throughout the embryo, including extra-embryonic tissues.

1.3. The Western Blot of nuclear Smad2/3 doesn't allow for the conclusion that signalling levels are elevated since it lacks critical controls, such as for the purity of the nuclear purification vs. the cytoplasm. Most likely the cytoplasmic pool exceeds the nuclear, so that impurities in the purification might disturb the outcome (H3 is reduced in the YAP-KO). The authors should use pSMAD2 as readout and normalize to a) a loading control and b) total SMAD2. This seems the more common way to test for Nodal/SMAD2 pathway activity.

We appreciate the reviewer's suggestion and agree that proper normalization and confirmation of nuclear purity are essential for interpreting Smad2/3 localization. In response to the reviewer's concern, we assessed nuclear purity by probing for GAPDH in the nuclear fractions. We did not detect any GAPDH signal, suggesting minimal cytoplasmic contamination (**Figure 1H**).

Furthermore, we included new analysis using total embryonic extracts and observed a trend toward an increase in total Smad2/3 protein levels, although this was less pronounced than the nuclear accumulation (**Figure EV1K**). We believe these analyses support the interpretation that the increased intensity of SMAD2/3 bands in nuclear extracts of *Yap1* cKO embryos reflects enhanced nuclear translocation, rather than differences in sample processing or fractionation.

Additionally, we probed for β -CATENIN, the WNT pathway effector, and observed only a slight increase in the nuclei of the mutants compared to controls, further suggesting that the observed increase in nuclear SMADs is specific (**Figure 1H and Results section; page 5, lane 31 and page 6, lane 1-7**).

Due to the limited material from E7.0 embryos—each of which contains only a few thousand cells—we face significant technical constraints. To obtain sufficient protein, we pool ~8-10 embryos from at least six different litters over different days and flash-freeze them, which precludes the isolation of matched cytoplasmic and nuclear fractions post-thaw. As a result, only nuclear or total protein extracts can be reliably processed for Western blot, while the cytoplasmic fraction cannot.

Following the reviewer’s suggestion, we also assessed pSMAD2 levels by Western blot. While we successfully detected pSMAD2 in human H1 cells, we were unable to detect a clear signal in mouse embryos. This likely reflects the limited amount of starting material and the transient or spatially restricted nature of SMAD2 phosphorylation in embryos. These limitations restrain our analysis of the pSMAD2 (**Reviewer Figure 2**).

We conclude that, although we were unable to reliably detect pSmad2 by WB in these low-input samples, the combined data from scRNA-seq, RT-qPCR, and nuclear extract analyses consistently point toward increased TGF β /NODAL signaling activity in the absence of YAP1.

Reviewer Figure 2:

(A) Full western blot of E7 WT mouse embryos and WT H1 hESCs. pSMAD2 is detected (red arrow) in hESCs but is absent in E7 mouse embryos

1.4. The pathway analysis (Fig. 1L) is uninformative and doesn't allow to look at the actual data. Do the authors expect the reader to decode the different numbers?

We agree that the original presentation of pathway enrichment in Figure 1L (**Revised Figure EV11**) could be made more intuitive. However, we believe the analysis is informative. The heatmap represents a Single Cell Pathway Analysis (SCPA), which scores differences in pathways activity across analyzed cell populations. This format, which visualizes pathway enrichment using color-coded q-values, a format widely used in recent publications^{7,8} and aligns with the guidelines provided by the official SCPA Visualization Tutorial and GitHub Repository. We agree that it was a mistake to show in the main Figure the terms codes and in the supplemental figure the associated full names. In the revised version, we have moved this representation to supplemental, where it now includes the names of all the significant terms retrieved, positioned right next to the heatmap (**Figure EV11**).

To better address the reviewer's concern and to improve clarity, we replaced the original heatmap with a UMAP-based visualization, where pathway enrichment scores (q-values) for the most significant TGF β /NODAL and WNT-related terms are overlaid as color gradients across cell populations. This format allows readers to intuitively assess spatial and population-specific enrichment patterns (**Figure 1G**). Notably, the revised figure clearly shows that TGF β /NODAL and WNT pathways activities are enriched in the epiblast population of *Yap1* cKO embryos, supporting our conclusion that *Yap1* loss enhances Nodal (and WNT) signaling in these cells.

2. Most importantly, it remains unclear why the authors used T/Brachyury as readout for Nodal signalling. Most likely, T/Brachyury isn't directly regulated by increased levels of Nodal, but rather by canonical Wnt-signalling downstream of Wnt3a.

Our analysis of T/Bra was not intended to serve as a direct readout of Nodal activity alone, but rather to validate the expansion of the primitive streak (PS) domain in *Yap1* cKO embryos, as predicted in the scRNAseq analysis. T/Bra transcription has a conserved role in mesoderm differentiation and is one of the first markers of nascent mesoderm. *T/Bra* is initially expressed in the posterior embryo just before the emergence of the PS. As gastrulation progresses, its expression domain expands to include the PS, notochord, and later the tailbud NMP population⁹⁻¹². Therefore, since expression levels and distribution are reliable readouts of nascent mesoderm/PS formation, we chose it for analysis. We apologize for not explaining this correctly in the first submission, we clarify this in the revised text (**Introduction section: page 3, line 5-8. Results section, page 6, line 18-21**).

On the other hand, we respectfully disagree with the comment that T/Brachyury is not directly regulated by increased levels of Nodal. T/Brachyury expression is well-established to require both Nodal (SMAD/3) and canonical Wnt (β -CATENIN) signaling inputs, as demonstrated in multiple

prior studies^{13,14,15}. And it is also important to clarify that we observed enrichment of both WNT-related genes and NODAL in the epiblast of *Yap1* cKO embryos. To strengthen this point—which may not have been sufficiently clear in our initial submission—we have moved relevant Wnt-related data from the supplemental materials into the main figure (see **Figure 1G and Figure EV1I**).

We believe that the expansion of the T/Bra-positive domain supports the scRNA-seq analysis, which shows an increased number of cells in the primitive streak population in *Yap1* cKO embryos. We think this phenotype is consistent with the observed upregulation of both *Nodal* and Wnt signaling pathways in the epiblast.

2.2. Potential readouts for *Nodal* would be Smad2/3 regulated genes of anterior primitive streak derivatives, such as DE, node, notochord, prechordal plate. These would include *Eomes*, *Foxa2*, *Cer1*, *Lhx1*, *Gsc*, and others. The elevated expression of T/Brachyury rather hints at elevated Wnt-levels, as already previously reported by the same authors.

To strengthen our analysis, we examined the expression of NODAL target genes, including *Eomes* and *Foxa2*, both of which show consistent upregulation in mutant embryos compared to controls. We also observed a reduction in the ectodermal marker *Otx2*, consistent with a shift in epiblast cell fate toward primitive streak derivatives at the expense of ectoderm lineages (**Figure EV2F-2G and Results section; page 7, lane 10-14**).

2.3. I would expect that increased *Nodal* levels would also impact the proportion of different cell types in the gastrulating embryo, which, according to the authors, is not the case.

We apologize for not explaining this clearly in the original submission. We revised the text to explain this clearer. Our scRNA-seq analysis at E7 revealed that *Yap1* cKO gastrulas display a significant increase in the proportion of primitive streak cells, along with a slight decrease in epiblast cells (**Figure 2A**). These findings are consistent with our immunofluorescence analysis of *T/Bra*, which shows an expansion of the PS domain, and RT-qPCR results demonstrating upregulation of *Eomes* and *Foxa2* genes (**Figure 2B-D and EV2F-G**). This, along with studies at later timepoints included during the revision (see points below), demonstrate that *Yap1* deletion affect the proportion of gastrulation populations. (**Result section: page 6, lane 26-30 and page 7, lane 1-19**).

2.4. What is the morphogenetic defect in the YAP1 cKO embryos that indicate hyperactivation of *Nodal*-signalling. In the discussion the authors compare the lethal phenotype of YAP1 mutants, with the conditional mutants of the current study. However, no data are shown concerning the later phenotype. Does it reflect the hyperactivation of *Nodal*-signalling, as described for TGIF1/2 double mutants? Please clarify and add the data about later occurring phenotypes, lethality etc, with a specific view on *Nodal* signalling readouts (loss/specification of certain cell types).

We appreciate this important comment. In the revised version, we included new analysis of the *Yap1* cKO embryos at later developmental stages, focusing on the differential loss or gain of specific cell populations. We also added the lethality data for the Sox2-Cre conditional *Yap1* KO embryos, which similar to constitutive *Yap1* KO embryos, failed to progress beyond E8.5^{16,17}. Overall, the new analyses helped to clarify the phenotypic consequences of *Yap1* deletion in the epiblast. In the revised paper, we also discussed these phenotypes in the context of their resemblance to increased NODAL and WNT signaling activity in the epiblast (**Discussion section; page 16, lane 12-31**).

Specifically, our analysis older *Yap1* cKO embryos display severely underdeveloped headfolds and neural plate structures that are not compatible with life passed E8.5^{16,17} (**Figure EV2C-D**). These abnormalities are consistent with impaired ectoderm development and disruption of anterior structures¹⁸. To further determine the developmental consequence of epiblast *Yap1* deletion, we analyzed the proportion of cell populations in the scRNAseq of E7.75 embryos we previously generated¹⁶. Remarkably, a comparison of epiblast-derived populations between control and *Yap1* cKO at E7.75 revealed an increase in the proportion of hematopoietic stem cell and endoderm populations alongside a significant reduction in neural and surface ectoderm populations in *Yap1* cKO embryos (**Figure EV2E**). Other mid-posterior PS populations, including presomitic and somitic mesoderm, were also underrepresented in the *Yap1* cKO embryos (**Figure EV2E**). To confirm these results, we performed qPCR and Western blot analyses of the anterior primitive streak marker *Eomes*¹⁹, endoderm marker *Foxa2*²⁰ and neuroectoderm marker *Otx2*²¹ in E7.75 embryos. Consistent with the morphological analysis and scRNAseq data, *Eomes* and *Foxa2* were upregulated and *Otx2* downregulated in *Yap1* cKO (**Figure EV2F-G**) (**Results section; page 7, lane 4-14**).

Overall, we conclude that conditional *Yap1* deletion leads to imbalanced differentiation of the epiblast. Specifically, *Yap1* loss in the epiblast promotes the expansion of anterior primitive streak derivatives while compromising neural committed cells. This shift is consistent with increased NODAL and WNT signaling activities in the mutant embryos, as suggested by our transcriptomic analysis of the epiblast at E7. As mentioned above, these phenotypes are discussed in the revised manuscript (**Discussion section; page 16, lane 12-31**).

3. The functional link between YAP1 and QSER seems weak and difficult to interpret. Can the authors give further insight on what lead to their decision to select QSER1 among the many significant hits of the biotinylation assay as an interesting target of further analyzation? It is stated that „it has been only partially characterized" as the only reason justifying their further research. Have the authors looked into any other co-factors of Myc-YAP1-Bir2 or was their decision influenced by reasons not explained in the manuscript?

The concern is valid—we did not fully explain our rationale for selecting QSER1 for further investigation in the initial version of the manuscript. Our initial interest in QSER1 stemmed from

its identification as a significant hit in our BioID2 assay and from its limited characterization in the context of early development. To strengthen the rationale, we included additional analysis in the revised version. Specifically, we used protein-protein interaction energy prediction tools to evaluate potential YAP1-interactions among the BioID2 hits in our assay. Among the 83 BioID2 hits, QSER1 ranked within the top five predicted interactors of YAP1, with a more favorable interaction energy score than even known partners such as TEAD3 or AMOT (**Figure 3E**). This suggested QSER1 as a top interactor among the Biotinylated hits.

Thus, in the revised version we include this analysis and clearly articulate this rationale:

“Considering its predicted binding stability, along with QSER1’s prominent yet poorly understood role in pluripotency²², we chose to investigate this potential functional interaction in greater detail” (**Results section; page 9, lane 18-20**).

4. The quality of presented ChIP data and the data analysis is poor. Why are QSER1 peaks not shown in Fig. 3F, when all other marks and EZH2 are shown in relation to QSER1 peaks?

We apologize for not realizing that QSER1 peaks were not displayed in the heatmap panel. The revised Figure includes it (**Figure EV4D: all QSER1 peaks, ranked by QSER1 peaks**) and **Figure 4C** (QSER1 and YAP co-bound genes, ranked by YAP1 peaks).

4.2. Please always indicate number of sites at the y-axis (No. of QSER1 peaks in this example).

Agreed. We updated the presentation of the ChIP-seq data across the manuscript to indicate the number of peaks at the Y axis. (See **Figure 4C** and **Figure EV4A and EV4D**).

4.3. The overlap with H3K4me1 and H3K27ac seems very weak/minor. This would need a backup on genome browser views to convincingly demonstrate that there is significant enhancer-binding of QSER1.

We acknowledge that our original presentation may not have convincingly conveyed the extent of overlap. However, the overlap between QSER1 and enhancer-associated histone marks is not modest. In fact, a substantial number of QSER1 peaks co-localize with H3K27ac and H3K4me1 peaks—specifically, see table below, using two available datasets from Dr. Bernstein and Ren groups:

Number of Direct Overlap (<1bp)	Bernstein	Ren
QSER1:H3K4me1	4027	3367
QSER1:H3K27ac	3618	4862
QSER1:H3K36me3	14	58
QSER1:H3K27me3	2229	1563

The limited visualization of this colocalization was partly due to differences in signal intensity across the ChIP-seq datasets, which affected the apparent overlap. In the revised version, we adjusted the signal scale for each ChIP-seq dataset to more clearly highlight the co-occupied regions. These updated visualizations better illustrate the correlations of histone marks and QSER1 binding (See revised **Figure 4C** and **Figure EV4C**).

As the reviewer suggested, to illustrate this more clearly, we now include genome browser snapshots at representative loci in **Figure 4E** and **Figure EV4F**.

4.4. What is the regulation by QSER1? Is this related to methylation? Can this be shown?

QSER1 has previously been shown to interact with TET proteins and regulate DNA methylation at H3K27me3/EZH2-bound regions²². Therefore, initially, we hypothesized that QSER1 might influence methylation at YAP-bound regulatory sites. However, upon analysis of our ChIP-seq data, we found minimal overlap between QSER1:YAP co-bound regions and H3K27me3 occupancy (**Figure 4C**), arguing against a role in protecting methylation at these specific loci.

Furthermore, QSER1 protects methylation by interacting with TET proteins²², which were not enriched in our biotinylating assay. This suggests the existence of different QSER1 chromatin complexes and that QSER1 has additional functions beyond protecting methylation on YAP1 enhancers.

To gain insight into QSER1's potential activities, during the revision, we used the professional services of a Molecular Modeling facility to analyze QSER1's predicted structure through AlphaFold tools²³, since there is no available QSER1 structure in PDB. The results revealed that QSER1 is largely intrinsically disordered, with only two folded domains in the C-terminus: one of unknown function and a "bivalent Mical/EHBP Rab binding" (bMERB) domain²⁴, a mostly helical domain, which is likely to mediate protein-protein interactions, including those with Rab GTPases (**Figure EV4G-H**). None of these domains are predicted to have enzymatic functions or catalytic activities. Notably, we found no evidence of a canonical DNA-binding domain, suggesting that QSER1 is likely recruited to chromatin through interactions with other DNA-bound proteins.

Given QSER1's extensive intrinsically disordered regions and lack of predicted catalytic or DNA-binding domains, we propose that QSER1 functions as a scaffold or co-regulator that facilitates protein-protein interactions within chromatin-associated complexes (see **updated Discussion; page 17, lane 20-31**).

Based on these features, we hypothesize that QSER1 may act by stabilizing recruitment of transcriptional regulators, to fine-tune transcriptional levels, in accordance with the roles described for other intrinsically disordered proteins^{25,26}. This analysis helped us to build a rationale for why we examined RNA Polymerase II binding, as a readout of transcriptional regulation, instead of DNA methylation. Our analysis are consistent with a role of QSER1 in restraining transcriptional activity of YAP-bound enhancers, which is also in line with a recent report on cancer cells reporting that QSER1 acts as a repressor²⁷ (**Figure 6A-C**).

Therefore, we did not assess DNA methylation and instead focused on alternative mechanisms consistent with the enhancer binding profile and guided by the predicted structure of QSER1.

4.5. How do the authors envision how YAP1 could regulate QSER1 binding in cases where they don't overlap on the same regions? There only seems a very loose interaction, which is only shown in 2 examples (please also read below).

This is another fair comment that needed clarification, as the reviewer pointed out.

Indeed, YAP1 and QSER1 do not always co-occupy the exact same genomic regions. However, we believe that these regions, while separated linearly by several kilobases, are brought into spatial proximity via cohesin-mediated chromatin looping, allowing functional interactions between QSER1 and YAP1-bound elements.

To support this model, we conducted an integrative analysis of QSER1, YAP1, and cohesin (NIPBL) ChIP-seq data in the revised manuscript. Among the 1192 genes that show both QSER1 and YAP1 occupancy, 1098 genes display cohesin peaks (**Figure 4D-E**). Furthermore, among them, 899 NIPBL peaks perfectly overlap with YAP1 (while 525 NIPBL peaks overlap with QSER1), as highlighted in the examples shown in **Figure 5A** and **5C**. This analysis suggests that YAP1 is frequently positioned within chromatin loops where physical contact with QSER1-bound regions is possible, even without direct peak overlap.

To further assess this possibility, we applied molecular modeling analysis to model QSER1, YAP1, and TEAD4 binding to the DNA sequence of the NODAL enhancer (**Figure 5E**). This prediction supports a structural model in which YAP1 serves as a bridging factor to facilitate QSER1 contacts to TEAD enhancers. In the absence of YAP1, QSER1 predicted binding stability to these regions is reduced, likely due to the loss of a stabilizing interface (**Figure 5F and EV5F**).

In summary, although QSER1 and YAP1 do not always co-bind to the exact same DNA sequences, we propose that their functional interaction is mediated by chromatin looping and protein-protein

contacts facilitated by YAP1, enabling coordinated regulation of target genes (**Figure 5E-F** and see **Discussion; page 18, lane 1-14**).

Furthermore, in the revised version, we extended our ChIP-qPCR analysis to 12 genes. Among them, 9 show the same trend in QSER1 reduction in the absence of YAP1 (**Figure 5B and 5D**).

Additional issues:

For future work, may I suggest to use the annotation of line numbers which massively help during reviewing.

We sincerely apologize for that. We added the numbers in the revised manuscript.

1. At several places I suggest re-phrasing to match commonly used nomenclature, or clarify phrasing "whole-body YAP1 knockout" (constitutive gene-deletion), "cYAP1KO" (YAP1cKO), "mid-end gastrulation stage (E8)": This is incorrect! Gastrulation proceeds much longer until the formation of the body axis is completed (rather E12) "spatially and temporally differentiates": unclear what this means.

- We have revised our nomenclature throughout the manuscript to align with commonly used conventions. Specifically, we changed "*cYap1* KO" to "*Yap1* cKO", and "whole-body YAP1 knockout" to "constitutive *Yap1* KO", as suggested.
- To avoid ambiguity in developmental stage descriptions, we no longer use terms like "mid" or "late gastrulation." Instead, we refer to specific embryonic days (e.g., E7.0, E7.75, E8.5) to more accurately indicate the developmental stage.
- The phrase "spatially and temporally differentiates" has been changed to "the epiblast differentiates" to avoid confusion (**Introduction; page 2, lane 27-28**).

2. Please reference the Sox2:Cre deleter strain in the text. To my knowledge this is a transgenic strain, so that the nomenclature should reflect this (Sox2Cre/+ is incorrect). Also, the nomenclature of the alleles in Fig.1A and text might be misleading. YAP1 is always deleted by the Sox2:Cre deleter in the male, so that the male mice are heterozygous, and most likely also the extraembryonic tissues in YAP1flox/flox, Sox2 +/+ embryos, if there is not wt allele present. This might be important since there are effects on extraembryonic tissues that may result from haploinsufficiency.

- The Sox2-Cre deleter strain was referenced in the original submission. In the revised version, we referenced it in **Result section; page 4, lane 9**.
- We agree that the use of "Cre/+" was inappropriate and corrected the nomenclature to reflect that Sox2-Cre is a transgenic line. In the revised manuscript, we consistently refer to this line as Sox2-Cre.

- The reviewer is also correct that *Yap1* is deleted in the male germline by Sox2-Cre, resulting in heterozygous embryos from these breeders. We were aware of this, and we mentioned it in the initial submission. However, we now emphasize it more clearly in the revised text and have clarified the consequences of this breeding strategy in the Methods section (**Page 19, lane 28-29**), and mentioned in the discussion. In particular, we explain that the analyzed *Yap1*^{flox/flox} embryos (regardless of Sox2-Cre presence) are heterozygous for *Yap1* in extraembryonic tissues due to paternal allele deletion, which may have implications for interpreting potential effects on extraembryonic compartments (**Discussion; Page 17, lane 1-19**).

3. Fig. 1E: Please state absolute No. of DEGs.

Done. Please see revised **Figure 1C**.

4. Fig. 2B: The quantification of fluorescent intensities is surely problematic. Generally, the brightness in mutants seems elevated, also in regions that lack T/Brachyury in the anterior. How can this be explained. How were the images normalized?

We apologize for a mistake in the axis of Figure 2B (Revised **Figure 2C**) that generated the confusion. The X-axis of the graph represents the signal intensity in the proximal-to-distal axis, not the posterior-to-anterior, as previously indicated. In the revised version, we also include the actual posterior-to-anterior quantification (**Figure 2C**).

For the proximal-to-distal axis quantification; immunofluorescence images were analyzed using the line tool in MATLAB. Three “cup-shaped” lines were drawn from the proximal posterior part of the epiblast to the most distal anterior part of the epiblast. The fluorescence signal of the three lines was averaged for each embryo. Each embryo image was rescaled to the same size. The average of the intensity profiles from multiple embryos was plotted in the graph shown in **Figure 2C**.

For the revised posterior-to-anterior quantification; The extraembryonic and embryonic parts of the embryo were located by eye. The embryonic part was divided in six bins by drawing five parallel straight lines from the boundary between the extraembryonic tissue down to the distal tip of the epiblast. The average fluorescence signal of each bin was plotted in the graph (from posterior to anterior). The average of the intensity profiles from multiple embryos was plotted in the graph shown in **Figure 2C**.

See also revised **Methods (Page 27, lane 10-23)**.

5. Is the fusion protein of YAP1-Bir functional? Was this experimentally tested?

Yes, the YAP1-Bir2 fusion construct is functional. In our initial submission, biotinylation activity was confirmed by Western blot analysis in the presence and absence of the construct, as shown in the revised **Figure 3B**.

In the revised version, we included additional validations:

- We included new ChIP-qPCR analysis of the Yap1-Bir2 construct in the presence and absence of Doxycycline, and detected binding at the Nodal enhancer, compared to negative control region, consistent with functional Yap1-Bir2 binding to TEAD4 sites on the chromatin (**Figure EV3D**).
- Additionally, in the revised version, we incorporate co-immunoprecipitation experiments to show that the YAP1-Bir2 fusion protein retains the ability to interact with TEAD4 (**Figure EV3C**).

These results support that the YAP1-Bir2 construct is biochemically active and engages in its expected protein–protein and chromatin interactions.

5.2. It is surprising, that no TEAD was found in the MS analysis. Is this due to steric hindrance? Please clarify.

We agree that the apparent absence of TEAD proteins in our BioID mass-spectrometry (MS) data seemed counter-intuitive, and we failed to explain this clearly in the original submission. In the revised version, we explain in more detail our data. Along with the additional experiments performed in the revision, we hope we provide enough evidence to validate the functionality of the Bir2 construct and the BioID2 assay.

We had four experimental conditions in our biotinylation assay and three biological replicates (see Table below). Few TEADs peptides were detected in only one or two of the three “+Dox +Biotin” replicates. The enrichment did not reach our statistical threshold (FC>2, FDR<10%, with detectable MS counts in at least 2 of the replicates), but the pattern is consistent with a trend enrichment in the presence of YAP and Biotin, compared to the other samples.

Protein	MS/MS spectral counts											
	Plus Biotin			Dox +			Minus Biotin			Dox +		
	Rep1	Rep2	Rep3	Rep1	Rep2	Rep3	Rep1	Rep2	Rep3	Rep1	Rep2	Rep3
YAP1	66	160	45	43	59	140	0	0	2	0	0	Lost
TEAD1	0	4	1	0	0	1	0	0	0	0	0	Lost
TEAD2	0	1	0	0	0	0	0	0	0	0	0	Lost
TEAD4	0	3	0	0	0	0	2	1	0	1	0	Lost

Published BioID/TurboID datasets report similarly low TEAD recovery when YAP was used as a bait. For example, a YAP-BioID screen in *Mol Cell* (2019)²⁸ retrieved ~5 TEAD peptides versus >70 peptides for other cofactors, in trend with our datasets. In the revised text, we mention factors that may contribute to this:

1. Limited accessible lysines on TEAD proteins. Proximity ligases biotinylate solvent-exposed lysines. The structured TEAD bound to the DNA contains relatively few such residues, reducing the probability of tagging²⁸⁻³⁰.
2. Geometric constraints due to the tight YAP:TEAD interactions. The orientation of the Bir2 enzyme (cloned at the N-terminal of YAP) in the YAP protein along with the rigid conformation of the YAP-TEAD complexes, may cause TEAD to hide from the effective “biotin cloud”, whereas more flexible or transient partners are labelled efficiently.

We included this discussion in the text (**Result section; page 8, lane 30-31 and page 9, lane 1-4**) along with the table in **Figure EV3E**.

5.3. Experiments could involve co-IP between Myc-YAP1-Bir2 and TEAD.

In the revised manuscript, and as we mentioned above, we include co-immunoprecipitation experiments demonstrating that the Myc-Bir2-YAP1 fusion retains the ability to interact with TEAD4 (**Figure EV3C**). Additionally, we provide ChIP-qPCR data showing that the construct binds TEAD4-bound regions (**Figure EV3D**).

5.4. Is dox-inducible Myc-YAP1-Bir2 expressed at comparable levels in comparison to endogenous YAP1?

Yes, the Western blot shown in **Figure 3B** using the YAP1 antibody demonstrates that the expression of Myc-Bir2-YAP1 is similar to the endogenous YAP1 levels under the conditions used for the experiments (25pg/ml Dox).

6. Please explain the rationale for RNA-PolII ChIP on enhancers. In the introduction, it is stated that YAP1 and QSER control RNA-PolII activity. Activity was not checked and tested.

We agree with the reviewer that we did not directly assess RNAPII activity—such measurements would require methods like nuclear run-on or RNAPII phosphorylation state analysis, which are part of future projects in the lab-. Instead, our experiments focused on RNAPII recruitment to enhancer regions in the presence or absence of QSER1 to test a potential role of QSER1 in transcriptional regulation. We hope that answer above (**Reviewer 1 main concerns, number 4.4**) provided above support the rationale to investigate how QSER1 controls RNAPII recruitment in YAP:QSER1 binding sites.

7. In the text and in the Volcano Plot of MS data, CTNNBL1 is mentioned and shown. To my knowledge this is NOT beta-Catenin, and not involved in canonical Wnt-signalling as stated in the text.

We apologize for this mistake and thanks the reviewer for catching it up. Indeed, CTNNBL1 is not related to Wnt signaling. CTNNBL1 (Catenin Beta-Like 1) functions in RNA splicing, cell cycle regulation, and DNA damage response. We did not mention it in the text in the revised version.

8. Fig. 3D: Why are there no proteins detected on the left side of the volcano plot? Is this a technical artefact? It is rather unusual that there is no enrichment in the negative control?

The asymmetry of the volcano plot is not a technical artifact but an expected outcome of the experimental design. In the revised version we explained this in more detail. The volcano plot in revised **Figure 3D** shows the hits significantly enriched ($FC > 2$, $FDR < 10\%$, with MS values in at least 2 replicates) in samples expressing Bir2-YAP1 construct treated with biotin (+dox,+biotin) versus those without biotin (+dox, -biotin). Thus, we do expect a positive enrichment of hits in the plus versus minus biotin condition, as represented in the Volcano plot (pink dots versus grey dots).

Also, in the revised version, we clarify that we had a second layer of filtering, that is now reflected in the Volcano plot by different colors (red dots versus pink dots). From the ~300 hits significantly enriched (pink dots), we filtered out those that were still biotinylated in the absence of the Bir2-Yap1 construct (-dox, +biotin) and in the absence of the Bir2-Yap1 construct and biotin (-dox, -biotin). After these filters, we obtained a subset of 83 hits (red dots), as bonafide biotinylated targets of Yap1. (See **Results section; page 8, lane 10-19** and **Methods section: page 21, lane 13-24**).

Previously published studies also have similar representations of volcano plots to display biotinylated hits³¹⁻³⁴.

9. Figure S3B: the label of y-axis is "genes, the label of x-axis is "TSS" and gene distance (bp). I guess that all these labels need correction: regions (indicate numbers), peak summit, peak distance in kb. Are all regions ranked in the same order? If so, this would indicate a remarkable similarity of both ChIP-experiments. Please clarify.

Indeed, there is a remarkable similarity in both ChIP-seq experiments. The axes were correct, but we clarified the labels (number of genes) in the revised paper. To clarify, we also explained how this comparison was carried out. To generate the heatmap, we reanalyzed the FLAG-QSER1 ChIP-seq dataset from Dixon et al.,²² using the same HOMER pipeline applied to our datasets. This allowed consistent annotation of peaks and genes in the two datasets. The Dixon's dataset

identified 50,175 peaks associated with 18,424 genes. The heatmap shown in **Figure EV4A** is the ranking of the signal of Flag-QSER1 peaks annotated near the transcription start site (TSS) of the 18,428 genes. We then compared the signal of our QSER1 ChIP-seq dataset on these genes, which yielded fewer peaks (14,461 peaks), annotated to a total of 7,942 genes. Thus, the two heatmaps show the TSS (± 5 kb window) of 18,424 Flag-QSER1 genes, ranked in the same order. The ranked heatmaps revealed a highly overlapping pattern between both datasets, as reviewer noted. In the revised version, we also included a Venn Diagram that highlights the overlapping genes between the two datasets (**Figure EV4A**).

The degree of similarity is expected, as both datasets were generated using the same H1 human embryonic stem cell line. Importantly, our results using an antibody against QSER1 yield fewer peaks than the FLAG antibody used against the tagged QSER1. However, this analysis strengthens the confidence in the binding distribution of our QSER1 ChIP-seq analysis, as most peaks were identified by two independent laboratories.

10. Figure S3D: It is rather unclear what this figure represents.

We have updated the figure and figure legend for clarity. Original Figure S3D (**Revised Figure 4B**) was a schematic representation summarizing the distribution of QSER1, YAP1, and EZH2 ChIP-seq peaks on associated genes. Each row indicates number of genes defined by the distribution of peaks for the indicated factors. Then, overlapping circles represent peaks that are closely co-localized in that set of genes. Separated circles indicate peaks bound to different regulatory regions annotated to the same gene.

For example, we identified 1,192 genes with both QSER1 and YAP1 peaks. Of these, 199 genes show co-binding of QSER1 and YAP1 proteins to the same regulatory region (presumably the same enhancer), while in 993 genes, QSER1 and YAP1 bind to different regions. The same logic applied to QSER1 and EZH2 co-binding analysis.

To simplify this and maintain the focus on the paper, we removed EZH2 from the scheme and only display co-localization of QSER1 and YAP1 peaks. The simplified scheme is in **Figure 4B**.

11. Figure S3E: To spread ChIP peaks over 20 kb(!) is unusual and doesn't allow to interpret where the peaks are located on genomic scale and if they overlap. Indicate number of regions shown (y-axis) and include QSER1 and YAP1 in heatmap representation.

We appreciate the reviewer's feedback. The original heatmap in Figure S3E displayed signal intensity within a ± 10 kb window centered on the peak summits. While this broader range is less common, it was intended to illustrate that the ChIP signal is highly enriched at the peak center

with minimal background signal in the surrounding genomic regions—particularly useful when visualizing a limited number of regions, as in our case.

Nonetheless, in response to the reviewer’s concern, we have updated the figure to show a narrower window of ± 2 kb around the peak center. We also now indicate the number of regions included on the y-axis and have incorporated both QSER1 and YAP1 into the heatmap for a more informative comparison.

The updated heatmap displays 1,538 YAP peaks located at the 1,192 genes co-bound by YAP and QSER1, ranked by YAP1 peak intensity. The X-axis represents a ± 2 kb window centered on each of the YAP1 peaks. The signal from QSER1 ChIP is also shown (see revised **Figure 4C**).

12. Figure 3H: What is the value on the x-axis? Is this FDR-adjusted p-value, q-value, or indeed FDR? Please indicate No. of positive terms per pathway. The representation seems unusual. Why do authors use different representation for Figs. 1L and 3H?

Original Figure 3H showed the results of a Reactome Pathway analysis, while Figure 1L illustrated a single-cell pathway analysis (SCPA). Reactome pathway identifies biological pathways significantly enriched in bulk datasets. SCPA is a method that scores pathway activity on heterogeneous cell populations. The SCPA uses Reactome pathways as input, but the representations are different, since the SCPA will show enrichment of pathways (i.e q values) across different cell populations identified in the datasets. For instance, in our case, the SCPA analysis show significant enrichment of TGFb/NODAL and WNT terms in the epiblast population, compared to other populations in the sample.

To avoid confusions, we used a KEGG pathway analysis for the panel in **Figure 4F**, which is a more standardized method to represent pathways enrichment in a dataset³⁵. The resulting categories are similar to the ones obtained with the Reactome Pathway analysis. All significant terms are shown and ranked by adj p value, as indicated in the X axis. Some of the genes inside each term are highlighted in the revised **Figure 4F**.

13. The PEE element of Nodal was defined in mouse, as referenced in the text. However, the authors use hESCs. Please adjust the reference, and/or show that the indicated PEE is synonymous in human.

The reviewer is correct. We clarify this in the revised text as follows: “In mouse embryos, a regulatory region known as the Proximal Epiblast Enhancer (PEE) is critical for NODAL expression in the PS during gastrulation³⁶. Our previous studies identified that the PEE region is conserved in the human genome and that YAP1 represses this enhancer in hESCs³⁷. In this study,

we identified that QSER1 co-occupies this key enhancer of *NODAL*, located ~13 kb upstream of the TSS in human cells”.

14. Text and Figures occasionally differ: Fig. 4A and text don't fit. Figure shows SMAD2, but text mentions INHBA.

We could not find this inconsistency in our paper. The original **Figure 4A** (revised **Figure 5C**) showed both SMAD2 and INHBA genes, and we mentioned both in the text. In any case, we revised text and figures to make sure they fit.

15. The author analyze QSER1 binding in YAP1-KO via ChIP-qPCR focusing on two genes for each previously defined group (Fig. 4B). It is stated that „199 out of the 1,192 genes show a perfect overlap of QSER1 and YAP1 binding at the same regulatory region", Can the authors show more than 2 out of these 199 genes that show the same decrease of QSER1-binding upon YAP1-KO if they want to make the claim that YAP1 controls DNA accessibility (I assume they mean DNA binding) of QSER1 on coregulated genes.

We have expanded our analysis to include **12 additional genes** from the group of overlapping QSER1 and YAP1 peaks. These additional ChIP-qPCR results are now included in the revised figure and confirm that QSER1 binding is consistently reduced upon YAP1 knockout at multiple co-bound sites (**Figure 5B** and **5D**).

Additionally, we agree that "DNA accessibility" was not the most precise term in this context. We have updated the manuscript text to more accurately refer to "DNA binding of QSER1" to avoid confusion.

16. The authors state that a variety of genes related to TGF-beta signaling are co-bound by YAP1 and QSER1 however it remains unclear, whether these genes are actually regulated by these two proteins (eg. SMAD3, ACVR2A) or however play any physiological role in early development of mammals at all (eg. INHBA). The GO pathway analysis depicted in Fig. 3H does not give any further insights into this issue since among the enrichment in TGF-beta/Nodal and Wnt-signalling are completely unrelated terms like „ionotropic glutamate receptor" or „angiogenesis".

We agree that binding does not imply transcriptional regulation or functional involvement in early development, and we ensured to clarify this in the result section (**Page 12, lane 6-8**). In the text associated to **Revised Figure 4** (previous Figure 3), we describe the genes co-bound by QSER1 and YAP1. These include several well-established components of the TGF- β /Nodal signaling pathway, such as *NODAL*, *SMAD2/3*, and *ACVR2A*, which are known to play critical roles in early mammalian development. We acknowledge that other co-bound genes have less clearly defined roles in early embryogenesis. However, given that our co-bound gene list includes 1,192

genes, we expect a degree of functional heterogeneity, and not all targets will be directly involved in the same developmental process. The analysis is intended to provide an unbiased overview of the biological categories represented in the co-bound gene set, which likely reflect genes with multiple functions. For example, “Pluripotency pathways” and “Pathways in cancer” share many genes, including those for TGFb/Nodal and Wnt signaling.

On the other hand, our functional analysis shows that QSER1 deletion increases RNAPII binding in a set of the YAP:QSER1 co-bound genes (**Figure 6A-C**) and that QSER1 is a negative regulator of Nodal signaling in hESC-derived 2D gastruloids. These findings are further supported by qPCR analysis of NODAL and WNT3 genes (**Figure 6D**), and a shift toward mesoendodermal fates in the gastruloids (**Figure 6E-G**), consistent with elevated Nodal pathway activity.

Overall, our findings support that YAP1 and QSER1 co-bind multiple *Nodal* signaling genes (among others), and that their presence is required to maintain low transcriptional activity in at least some of these targets and restrain overall pathway activation in hESCs.

In summary, the current manuscript requires major revisions, that include both, major text revisions and corrections, and also a substantial revision on experiments and data analysis to substantiate the major claims of the study, namely increased Nodal-signalling levels and functional interactions with QSER1.

We appreciate the reviewer’s thorough evaluation. In response, we have carefully revised the manuscript to address all concerns raised. This includes:

Experimental revision, such as the addition of key controls (e.g., total SMAD2/3 analysis, inclusion of GAPDH in nuclear extracts), RT-qPCR analysis of additional Nodal-target genes in vivo, expanded ChIP-qPCR analysis on YAP1:QSER1 co-bound genes, and validation of BioID assay by multiple approaches, including in-silico analysis and co-immunoprecipitation experiments.

Figure revision, including replacing UMAP plots with violin plots for better data representation, revising axis labels, and clearly displaying all relevant controls.

Text revision, including ensuring consistency with nomenclature guidelines (e.g., proper designation of mouse strains) and refining language to accurately reflect findings (e.g., avoiding implications of causality where only correlation is shown).

Collectively, we believe this revision strength our main conclusions:

-YAP1 deletion in the epiblast leads to activation of signaling pathway genes (including Nodal) involved in the differentiation of the epiblast toward PS lineages.

-YAP1 cooperates with QSER1 to repress Nodal signaling during hESC differentiation.

Referee #2:

The manuscript entitled "YAP1 and QSER1 are Key Modulators of Embryonic Signaling Pathways in the Mammalian Epiblast" by Abraham et al. investigates the function of the mouse and human YAP1 protein in the epiblast and pluripotent stem cell differentiation. The authors perform single cell RNAseq and can show that YAP1 is a major regulator of Nodal signals and use conditional mutation in the epiblast to infer expanded expression of primitive streak markers. Conversely in human pluripotent stem cell derived gastruloid cultures elevated mesodermal differentiation is observed. Molecularly, the study shows that YAP1 acts both as activator and repressor and specifically interacts with QSER1 on enhancers. This interaction is independent and distinct from previously reported QSER1 function to restrain DNA methylation on Polycomb sites. Biochemical interaction is demonstrated by engineering a Bio2-tagged YAP1 and mass spectrometric identification of streptavidin bead bound proteins from nuclear extracts. The study is of interest for researchers in early mammalian development and cell signaling and contains data of high quality. The main advance is of extending of YAP1 function in early embryonic lineages and demonstrating a requirement in the epiblast which is distinct to YAP1 function in TE specification. Several points should be addressed to improve the impact and clarity of the study.

Thank you for the appreciation of our study. We addressed reviewers concerns to improve the impact and quality of our study.

Specific points

1. The introduction raises the question of an involvement of Hippo signaling in cell growth and organ size control. The current version seems to rule out embryonic growth regulation but does not consider TAZ (Wwtr1) which could provide a redundant function. Although, a Wwtr1 mutation in mice is compatible with epiblast development the absence of a discussion is surprising as TAZ is a separate gene but the introduction would make one believe it is a synonym. It would be interesting to know to what extent the absence of YAP1 would be compensated by TAZ or if there is reciprocal regulation or dependence. A few sentences should be added to the discussion to orient the reader and possibly the RNAseq data might be interrogated for expression changes in Wwtr1.

We thank the reviewer for this important observation. To address this point, we have introduced clarifications in both the Introduction and Discussion that TAZ (encoded by the *Wwtr1* gene) is a homolog of YAP1 but plays a less prominent role during early development. This is supported by genetic studies showing that *Wwtr1* knockout mice are viable, while *Yap1* knockout results in early embryonic lethality (**Introduction section; page 2, lane 12-14**).

In addition, we used our scRNA-seq dataset to assess *Wwtr1* expression levels in control and *Yap1* cKO embryos. Our analysis shows no significant difference in *Wwtr1* mRNA expression across all cell populations in E7.0 embryos (**Figure EV1E**), suggesting that TAZ does not compensate at the transcriptional level for the loss of YAP1 in this developmental context. These points have been incorporated into the revised manuscript to provide readers with appropriate context regarding potential redundancy and the distinct roles of YAP1 and TAZ.

2. The authors mention that non-cell autonomous effects in YAP1 conditional mutant epiblast might also affect PS formation. It would be interesting to discuss if AVE development is affected - maybe secondarily from signals from the epiblast. Absence of the AVE would likely also explain expanded posterior fates and elevated signaling as this becomes a major source of signal inhibition in the anterior region. It would be interesting to see if scRNAseq provides evidence for AVE formation or a AVE defect can be ruled out on grounds of embryonic timing.

We appreciate this comment, which encouraged us to look closer at the extraembryonic populations.

Upon closer examination of the ExE populations of our scRNAseq datasets, we determined that population 9 (initially labeled as "ExE endoderm 2") expresses key markers of the anterior visceral endoderm (AVE), including *Lefty1*, *Cer1*, *Dkk1*, and *Hhex*³⁸⁻⁴⁰. This expression profile is consistent with AVE identity (see **Reviewer Figure 3**). Accordingly, we have updated its annotation in the revised Figures (**Figure 1B and 1C**).

We also compared whether there are differences in AVE cell number and gene expression in our cKO compared to controls. We do not observe significant changes in the proportion of cells annotated to the AVE population between *Yap1* cKO and control embryos (revised **Figure EV2A**), and only one gene was differentially expressed within this cluster (**Figure 1C**). Therefore, without additional data, it is difficult to speculate on whether AVE defects contribute to the phenotype of *Yap1* cKO embryos.

However, we did detect a substantial change in the proportion of other extraembryonic populations, especially in the extraembryonic ectoderm. This suggests, according to the reviewer's comment, that YAP1 activity in the epiblast may indirectly influence the formation of some extraembryonic tissues. In the revised paper, we discuss the potential involvement of altered

extraembryonic compartments in contributing to the patterning phenotypes in YAP-deficient embryos. We also mention the limitations in assessing changes in these populations in our mutant embryos due to our breeding strategy, which generates heterozygous embryos for the extraembryonic compartment, due to parental transmission of one floxed allele (See Discussion; page 17, lane 1-19).

Markers of AVE

Reviewer Figure 3:

(A) UMAP plot of E7 mouse embryo single-cell data with annotated clusters. The cluster originally labeled as "Exe ectoderm 2" has been renamed to "AVE" (anterior visceral endoderm) based on the expression levels of indicated markers in the UMAP representation.

Minor points

a) Results page 3 bottom: The authors argue that human pluripotent stem cells and mouse epiblast are comparable. A certain equivalence indeed exists but this would overlook substantial differences in streak formation, eg. in human there seems no DVE-AVE but early extraembryonic mesoderm formation. Also the human epiblast separates from the amniotic ectoderm which is not formed in mice until after gastrulation. Maybe a brief description of the relevant signaling events shared could be informative for the reader.

We completely agree with the differences highlighted by the reviewer. To acknowledge these distinctions while maintaining the rationale for using hESCs, we have revised the sentence as follows:

"Although key differences exist between mouse and human development at this stage—particularly regarding the formation of extraembryonic tissues⁴¹⁻⁴³, in vitro cultured human pluripotent stem cells share important functional similarities with the epiblast cells of the mouse, including their pluripotent state and capacity to give rise to all three germ layers^{44,45}."

b) Results page 4, second para: The absence of TEAD(1-4) in the masspec data is surprising. Is it clear if the authors' methodology could recover TEAD interactions? It would be interesting to compare other biochemical studies that have identified TEAD proteins as interactors to consider differences - did the nuclear lysates (input) used by the authors would contain TEAD proteins and these were not biotinylated?

This point was also raised by Reviewer 1, and we have addressed it in detail in our response to that comment (see **Reviewer 1, Minor Points 5, 5.2, 5.3 and 5.4**).

Briefly, we detected a few TEAD peptides in one of the BioID2-YAP1 replicates but did not reach statistical significance. We clarified this in the revised version. We also discussed potential reasons for the low TEAD recovery in the biotinylation assay (**Results section, page 8, lane 30-31 and page 9, lane 1-4**).

Furthermore, to further validate the functionality of the Bir2-YAP construct, we confirmed the interaction of the BioID2-YAP1 construct with endogenous TEAD4 by co-immunoprecipitation and confirmed chromatin binding to TEAD-bound enhancers by ChIP-qPCR analysis (**Figure EV3C-D**).

Overall, we conclude that the low mass spectrometry counts of TEAD proteins are a technical limitation of the biotinylation assay, and that the activity of the Bir2-YAP1 construct is similar to the endogenous YAP1.

c) Discussion, end of the first line: "aka" could be replaced by "and"
Changed, thanks.

Referee #3:

The manuscript by Abraham et al. reports the genetic interaction between YAP1 and QSER1 in modulating key signaling pathways (such as nodal) in mammalian epiblast.

Overall the paper is interesting for two main reasons: 1) as the authors mention, the role of YAP is extremely tissue-specific as YAP can control a plethora of diverse target genes and 2) QSER1 is a relatively poorly characterized modulator of DNA methylation.

However, I believe that the paper can be strengthened by addressing some key technical and biological questions:

We appreciate the positive feedback. We performed a thorough revision of experiments, texts, and figures to address all technical and biological questions.

1. The authors often refer to YAP1 as if it is acting independently of TEADs. I do believe this is a key point. I suggest the author to test some of their phenotypes and transcriptional readouts with TEAD inhibitors. Similarly, I would encourage the authors to add ChIP-seq tracks for TEAD in their IGV plots for completeness. It would not be surprising that, even if the observed regulation of Nodal is TEAD independent, still TEAD would act as an anchor for YAP on chromatin

We appreciate this insightful comment and fully agree that the role of TEADs in mediating YAP1 function deserves further attention. Based on the reviewer's suggestion, we have now expanded our analysis to incorporate TEAD4.

We have added TEAD4 ChIP-seq tracks to the revised IGV plots and included Venn diagrams (see revised **Figure 4A, Figure 5A, 5C and EV5A**), which show the expected substantial overlap between YAP1 and TEAD4 peaks, including on the YAP1:QSER1 co-occupied sites. These results support the notion that TEAD4 acts as a chromatin anchor for YAP1, genome-wide.

To directly test the role of TEADs in the regulatory mechanisms described, we performed new experiments using a pan-TEAD inhibitor (GNE-7883⁴⁶), which allosterically blocks interactions between YAP/TAZ and all four TEAD paralogs by binding to the TEAD lipid pocket. As shown in revised **Figure EV5D**, treatment with GNE-7883 led to upregulation of *NODAL* expression, recapitulating the transcriptional effects observed upon YAP1 or QSER1 depletion. These findings support that the YAP1-TEAD interaction is critical for this regulation.

We performed ChIP-qPCR analysis to test whether TEAD inhibition affects QSER1 binding to the *NODAL* enhancer. The results (**Figure EV5E**) show that TEAD inhibition reduces both YAP1

and QSER1 occupancy at this site, supporting a model in which YAP1:TEAD4 chromatin complexes facilitate QSER1 recruitment to their sites.

Prompted by the reviewer's suggestion, we also explored the structural basis of this interaction. In collaboration with Dr. Mark Andrade (Fox Chase Cancer Center), we used AlphaFold 3 to model the QSER1:YAP1:TEAD4 complex bound to the *NODAL* enhancer DNA. The predicted trimeric complex yielded a strong ipTM score of 0.68, indicating a high-confidence interaction. In this model, YAP1 is positioned between the bMERB domain of QSER1 and the TEAD4 DNA-binding domain, acting as a molecular bridge (**Figure 5E and EV5F**). In contrast, modeling of QSER1 and TEAD4 in the absence of YAP1 resulted in a much lower ipTM score (0.31), indicating weaker and more heterogeneous interfaces.

These structural predictions align with our ChIP-qPCR findings: YAP1 deletion (**Figure 5A**) or inhibition to TEAD binding (**Figure EV5E**) markedly reduces QSER1 chromatin binding to YAP:QSER1 co-occupied sites, while QSER1 loss does not affect YAP1 occupancy, which remains anchored via TEAD4 (**Figure EV5A and C**).

We thank the reviewer for raising this important point, which led us to perform additional experiments and analyses that significantly strengthened our mechanistic model. These revisions are now included in the main text (**Results section, page 13, lane 1-11**).

2. Along those lines, the authors should report if canonical target genes of hippo signaling are modulated in YAP cKO cells.

Agreed. We observed downregulation of typical YAP1 target genes, such as *CCN1*(CYR61) and *CCN2* (CTGF), along with Hippo pathway regulatory genes, including *AMOTL2*, *PTPN14*, and *WWC2*. We include the following description in the text:

“Consistent with reduced YAP activity in the embryos, we observed typical YAP1 target genes and Hippo regulators, transcriptionally affected. These include **CCN2 (CTGF)**, which was significantly downregulated across the primitive streak and nascent mesoderm, and **CCN1 (CYR61)**, which was reduced in the epiblast. Regulators of the Hippo pathway, including **AMOTL2**⁴⁷, **PTPN14**⁴⁸, and **WWC2**⁴⁹, were also downregulated across multiple lineages, such as the epiblast, primitive streak, and cardiac mesoderm (**Table EV1**). These results support effective YAP1 depletion and disruption of Hippo signaling in the cKO embryo”. (**Results section, page 4, lane 24-30**)

3. The authors identify QSER1 from a BioID2-MS experiment. On one side, the cutoff used for statistically significant enrichment in figure 2D seems quite lenient (-log₁₀pvalue 1). On the other, and most importantly, the data distribution seems highly asymmetric (quite odd if the analysis was

performed using a standard "differential abundance" method). I would suggest the authors to add in the Material and Methods the computational analysis used for Mass Spectrometry data.

The BioID2 assay is a proximity-based labeling technique that excels in capturing in situ interactions but is also known to generate noisy datasets. For this reason, many published studies prioritize fold-change (FC) enrichment over strict p-value cutoffs when identifying candidate interactors in this screening^{28,50-52}. However, in our analysis, we did apply a standard statistical filter: we used an adjusted p-value threshold (FDR < 10%) and a fold-change cutoff of FC > 2 to compare biotinylated proteins in biotin-treated Bir2-YAP1-expressing cells (+dox +biotin) versus untreated Bir2-YAP1-expressing cells (+dox -biotin). This first comparison yielded an initial list of 311 enriched proteins. Next, we filtered the 311 hits by comparing their enrichment in two additional conditions: (+biotin/-Dox) and (-biotin/-Dox), and retained only those proteins with a Log₂FC > 1.5 relative to both. This yielded a final list of 83 high-confidence proximity interactors, including QSER1.

In the revised version, to predict probabilities of potential YAP1 interactors among the hits identified in the BioID2, we performed computational docking to evaluate whether these proteins could plausibly form stable complexes with YAP1. This analysis allowed us to rank candidate interactors based on predicted structural compatibility, with lower energies associated with more stable complexes (see **Methods; page 21 and lane 13-24**). Supporting the validation of our biotinylated target list, around half of the hits are predicted to form complexes more stable than YAP1 with TEADs (**Figure 3E**). Notably, QSER1 ranked among the top five most energetically favorable interactors of YAP1. (**Figure 3E and Results; page 9, lane 7-27**).

Regarding the asymmetry of the volcano plot, this issue was also noted by Reviewer 1. We now clarify the distribution and underlying data characteristics in the revised text (**please see Reviewer 1, minor comment 8**). A detailed description of the mass spectrometry data analysis pipeline has been added to the Materials and Methods section (**Page 20, lane 22-31 and page 21, lane 1-24**).

We believe these revisions clarify our approach to the biotinylation assay analysis and support the identification of a robust hit list.

4. Related to the previous point, I cannot find a validation of the interaction between YAP1 and QSER1. A simple co-IP from nuclear fraction might strengthen the evidence of the interaction (given the small number of co-occupied genomic regions).

We thank the reviewer for this important suggestion. In the revised manuscript, we have included several complementary approaches to validate the interaction between YAP1 and QSER1:

- Co-immunoprecipitation (co-IP) from nuclear extracts confirms the physical association between Myc-Bir2-YAP1 and endogenous QSER1 (**Figure EV3C**).
- Immunofluorescence analysis demonstrates nuclear co-localization of YAP1 and QSER1 (**Figure EV3F-H**).
- Size-exclusion chromatography shows the co-elution of YAP1 and QSER1 in fractions of similar molecular weight (**Figure EV3I**).
- In silico structural modeling predicts a high-confidence interaction interface between YAP1 and QSER1, with a favorable binding energy score (**Figure 3E** and **Figure 5E**).

Regarding the “small number of co-occupied genomic regions” pointed out by the reviewer., we would like to clarify that the number of contacts detected in the ChIP-seq might be underestimated, which is further supported by the analysis of cohesin binding performed for the revision.

Indeed, YAP1 and QSER1 do not always co-occupy the exact same genomic sequence. However, we think that these binding sites are brought into spatial proximity via cohesin-mediated chromatin looping, allowing functional interactions between QSER1 and YAP1-bound elements.

To support this model, we conducted an integrative analysis of QSER1, YAP1, and cohesin (NIPBL) ChIP-seq data in the revised manuscript. Among the 1,192 genes that show both QSER1 and YAP1 occupancy, 1098 genes display cohesin peaks (**Figure 4C-D**). Furthermore, among them, 899 NIPBL peaks perfectly overlap with YAP1 (while 525 NIPBL peaks overlap with QSER1), which is highlighted in the examples shown in **Figure 5A** and **5C**. This analysis suggests that YAP1 is frequently positioned within chromatin loops, where physical contact with QSER1-bound regions is possible, even without direct peak overlap.

This model is also supported by our molecular modeling predictions, as explained above (**Reviewer 3, comment 1**), in which YAP1 serves as a bridging factor—stabilizing the interaction between QSER1 and TEAD4, on YAP1:TEAD enhancer regions (**Figure 5E-F**).

We discuss this new data and the interpretation of this modeling in the revised discussion (**Discussion, page 17, lane 20-31 and page 18, lane 1-14**).

5. It will also overcome the claimed technical limitations of BioID2 in detecting the interaction with TEADs.

In addition, to address the technical limitation of the BioID2 assay in capturing TEAD proteins, we performed co-IP with the Myc-Bir2-YAP1 construct and confirmed its ability to interact with endogenous TEAD4 (**Figure EV3C**). This supports the functionality of the construct and reinforces the proximity labeling results.

6. The authors should show the absolute numbers of cells for each cell types retrieved by scRNA-seq. Moreover, since the cell annotation was done by manually curated list of markers, plots demonstrating selective expression of such markers should be included in the manuscript.

Due to space constraints, we used a heatmap (**Figure 1B**) to summarize the selective enrichment of population-specific markers. These markers were selected based on their well-established roles in lineage specification, and their expression patterns were validated by differential expression analysis. For each marker gene, we performed differential expression testing against all other clusters combined. The adjusted p-values (shown above each plot) indicate significant enrichment in the assigned population. For example, Sox2 is significantly upregulated in the epiblast cluster relative to all other clusters. In the revised version, we clarified the selection of the markers used to annotate populations (**Methods section, page 29, lane 19-23**).

We kept the original heatmap in revised **Figure 1B**, but to further address the reviewer's concern, we provide individual violin plots (**see Reviewer Figure 4**) showing the expression of key markers across all clusters. These plots clearly demonstrate the selective expression patterns used to annotate each population.

Following the reviewer's suggestion, in the revised version, we included the absolute cell number of each population retrieved in the scRNAseq (**see revised Figure 1C**).

- Epiblast: *Sox2*
- Primitive Streak: *Nodal, Eomes, Fgf8*
- Nascent Mesoderm: *Mesp1*
- Cardiogenic mesoderm: *Hand, Kdr*
- Blood Progenitor: *Tal1, Sox7*
- Endoderm: *Sox17, Foxa1*
- ExE Ectoderm: *Elf5, Gata1*
- ExE Endoderm: *Fabp1*

Reviewer Figure 4:

(A) Violin plots showing the expression of marker genes used to define scRNA-seq clusters. Sox2 marks the epiblast; Nodal, Eomes, and Fgf8 mark the primitive streak; Mesp1 marks nascent mesoderm; Hand1 and Kdr mark cardiac mesoderm; Tal1 and Sox7 mark blood progenitors; Sox17 and Foxa1 mark endoderm; Elf5 and Gata1 mark extraembryonic ectoderm; and Fabp1 mark extraembryonic endoderm.

7. At the end of the 4th section of the results the authors claim that YAP1 prevents epiblast differentiation by restricting Noval and WNT pathways. I would argue that, in absence of functional data (i.e. rescue of phenotypes by genetically/chemically perturbing Nodal/WNT), their data are correlative. I would rather tone down such sentence.

Agreed. We correct the sentence to “ Overall, we conclude that conditional *Yap1* deletion leads to imbalanced differentiation of the epiblast(...). This shift is consistent with increased NODAL and WNT signaling activities in the mutant embryos, as suggested by our transcriptomic analysis of the epiblast at E7 (**Results section, page 7, lane 15-19**).

Minor points:

- The word "instead" in row 8 of the abstract rows 8 to me is not the best to introduce the sentence: since the sentence in row 8 is not in contrast to the previous one.

Agreed. Removed, thanks.

- In page 2, a reference should be assigned to the study reporting that mice *Alfp-Cre YAP fl/fl* in liver reach normal organ size.

Added. Apologies for this omission, and thanks. See **Introduction section, page 2, lane 17-19**.

- I am not sure I would use the word "unbiased screening assay" for a BioID2-MS experiment. It could simply being referred to a proteomic experiment.

Agreed. We removed the "unbiased screening assay" to proximity biotinylation assay across the text.

REFERENCES

1. Yu MS, Spiering S, Colas AR. Generation of First Heart Field-like Cardiac Progenitors and Ventricular-like Cardiomyocytes from Human Pluripotent Stem Cells. *J Vis Exp*. 06 2018;(136)doi:10.3791/57688
2. Malaguti M, Migueles RP, Blin G, Lin CY, Lowell S. Id1 Stabilizes Epiblast Identity by Sensing Delays in Nodal Activation and Adjusting the Timing of Differentiation. *Dev Cell*. Aug 19 2019;50(4):462-477.e5. doi:10.1016/j.devcel.2019.05.032
3. Cunningham TJ, Yu MS, McKeithan WL, et al. Id genes are essential for early heart formation. *Genes Dev*. Jul 1 2017;31(13):1325-1338. doi:10.1101/gad.300400.117
4. Wang Z, Oron E, Nelson B, Razis S, Ivanova N. Distinct lineage specification roles for NANOG, OCT4, and SOX2 in human embryonic stem cells. *Cell Stem Cell*. Apr 2012;10(4):440-54. doi:10.1016/j.stem.2012.02.016
5. Vallier L, Mendjan S, Brown S, et al. Activin/Nodal signalling maintains pluripotency by controlling Nanog expression. *Development*. Apr 2009;136(8):1339-49. doi:10.1242/dev.033951
6. Xu RH, Sampsel-Barron TL, Gu F, et al. NANOG is a direct target of TGFbeta/activin-mediated SMAD signaling in human ESCs. *Cell Stem Cell*. Aug 2008;3(2):196-206. doi:10.1016/j.stem.2008.07.001
7. Bibby JA, Agarwal D, Freiwald T, et al. Systematic single-cell pathway analysis to characterize early T cell activation. *Cell Rep*. Nov 22 2022;41(8):111697. doi:10.1016/j.celrep.2022.111697
8. Zheng L, Qin S, Si W, et al. Pan-cancer single-cell landscape of tumor-infiltrating T cells. *Science*. Dec 17 2021;374(6574):abe6474. doi:10.1126/science.abe6474
9. Rivera-Pérez JA, Magnuson T. Primitive streak formation in mice is preceded by localized activation of Brachyury and Wnt3. *Dev Biol*. Dec 15 2005;288(2):363-71. doi:10.1016/j.ydbio.2005.09.012
10. Bulger EA, Muncie-Vasic I, Libby ARG, McDevitt TC, Bruneau BG. TBXT dose sensitivity and the decoupling of nascent mesoderm specification from EMT progression in 2D human gastruloids. *Development*. Mar 15 2024;151(6)doi:10.1242/dev.202516
11. Stott D, Kispert A, Herrmann BG. Rescue of the tail defect of Brachyury mice. *Genes Dev*. Feb 1993;7(2):197-203. doi:10.1101/gad.7.2.197
12. Wilkinson DG, Bhatt S, Herrmann BG. Expression pattern of the mouse T gene and its role in mesoderm formation. *Nature*. Feb 15 1990;343(6259):657-9. doi:10.1038/343657a0
13. Funa NS, Schachter KA, Lerdrup M, et al. β -Catenin Regulates Primitive Streak Induction through Collaborative Interactions with SMAD2/SMAD3 and OCT4. *Cell Stem Cell*. Jun 4 2015;16(6):639-52. doi:10.1016/j.stem.2015.03.008
14. Estarás C, Benner C, Jones KA. SMADs and YAP compete to control elongation of β -catenin:LEF-1-recruited RNAPII during hESC differentiation. *Mol Cell*. Jun 2015;58(5):780-93. doi:10.1016/j.molcel.2015.04.001
15. Gadue P, Huber TL, Paddison PJ, Keller GM. Wnt and TGF-beta signaling are required for the induction of an in vitro model of primitive streak formation using embryonic stem cells. *Proc Natl Acad Sci U S A*. Nov 07 2006;103(45):16806-11. doi:10.1073/pnas.0603916103
16. Abraham E, Kostina A, Volmert B, et al. A retinoic acid:YAP1 signaling axis controls atrial lineage commitment. *Cell Rep*. May 07 2025;44(5):115687. doi:10.1016/j.celrep.2025.115687
17. Morin-Kensicki EM, Boone BN, Howell M, et al. Defects in yolk sac vasculogenesis, chorioallantoic fusion, and embryonic axis elongation in mice with targeted disruption of Yap65. *Mol Cell Biol*. Jan 2006;26(1):77-87. doi:10.1128/MCB.26.1.77-87.2006
18. Stiles J, Jernigan TL. The basics of brain development. *Neuropsychol Rev*. Dec 2010;20(4):327-48. doi:10.1007/s11065-010-9148-4
19. Arnold SJ, Hofmann UK, Bikoff EK, Robertson EJ. Pivotal roles for eomesodermin during axis formation, epithelium-to-mesenchyme transition and endoderm specification in the mouse. *Development*. Feb 2008;135(3):501-11. doi:10.1242/dev.014357
20. Burtscher I, Lickert H. Foxa2 regulates polarity and epithelialization in the endoderm germ layer of the mouse embryo. *Development*. Mar 2009;136(6):1029-38. doi:10.1242/dev.028415
21. Acampora D, Mazan S, Lallemand Y, et al. Forebrain and midbrain regions are deleted in Otx2^{-/-} mutants due to a defective anterior neuroectoderm specification during gastrulation. *Development*. Oct 1995;121(10):3279-90. doi:10.1242/dev.121.10.3279

22. Dixon G, Pan H, Yang D, et al. QSER1 protects DNA methylation valleys from de novo methylation. *Science*. Apr 09 2021;372(6538)doi:10.1126/science.abd0875
23. Abramson J, Adler J, Dunger J, et al. Accurate structure prediction of biomolecular interactions with AlphaFold 3. *Nature*. Jun 2024;630(8016):493-500. doi:10.1038/s41586-024-07487-w
24. Rai A, Oprisko A, Campos J, et al. bMERB domains are bivalent Rab8 family effectors evolved by gene duplication. *Elife*. Aug 23 2016;5doi:10.7554/eLife.18675
25. Ferrie JJ, Karr JP, Tjian R, Darzacq X. "Structure"-function relationships in eukaryotic transcription factors: The role of intrinsically disordered regions in gene regulation. *Mol Cell*. Nov 3 2022;82(21):3970-3984. doi:10.1016/j.molcel.2022.09.021
26. He J, Huo X, Pei G, et al. Dual-role transcription factors stabilize intermediate expression levels. *Cell*. May 23 2024;187(11):2746-2766.e25. doi:10.1016/j.cell.2024.03.023
27. Zhao X, Fang K, Liu X, et al. QSER1 preserves the suppressive status of the pro-apoptotic genes to prevent apoptosis. *Cell Death Differ*. Mar 2023;30(3):779-793. doi:10.1038/s41418-022-01085-x
28. Zhu C, Li L, Zhang Z, et al. A Non-canonical Role of YAP/TEAD Is Required for Activation of Estrogen-Regulated Enhancers in Breast Cancer. *Mol Cell*. 08 2019;75(4):791-806.e8. doi:10.1016/j.molcel.2019.06.010
29. Minde DP, Ramakrishna M, Lilley KS. Biotin proximity tagging favours unfolded proteins and enables the study of intrinsically disordered regions. *Commun Biol*. Jan 22 2020;3(1):38. doi:10.1038/s42003-020-0758-y
30. Mesrouze Y, Bokhovchuk F, Izaac A, et al. Adaptation of the bound intrinsically disordered protein YAP to mutations at the YAP:TEAD interface. *Protein Sci*. Oct 2018;27(10):1810-1820. doi:10.1002/pro.3493
31. Hémono M, Haller A, Chicher J, Duchêne AM, Ngondo RP. The interactome of CLUH reveals its association to SPAG5 and its co-translational proximity to mitochondrial proteins. *BMC Biol*. Jan 10 2022;20(1):13. doi:10.1186/s12915-021-01213-y
32. Wright MH, Paape D, Storck EM, Serwa RA, Smith DF, Tate EW. Global analysis of protein N-myristoylation and exploration of N-myristoyltransferase as a drug target in the neglected human pathogen *Leishmania donovani*. *Chem Biol*. Mar 19 2015;22(3):342-54. doi:10.1016/j.chembiol.2015.01.003
33. Yamanaka S, Horiuchi Y, Matsuoka S, et al. A proximity biotinylation-based approach to identify protein-E3 ligase interactions induced by PROTACs and molecular glues. *Nat Commun*. Jan 10 2022;13(1):183. doi:10.1038/s41467-021-27818-z
34. Uçkun E, Wolfstetter G, Anthonydhasan V, et al. In vivo Profiling of the Alk Proximitome in the Developing *Drosophila* Brain. *J Mol Biol*. Nov 19 2021;433(23):167282. doi:10.1016/j.jmb.2021.167282
35. Kanehisa M, Sato Y, Kawashima M, Furumichi M, Tanabe M. KEGG as a reference resource for gene and protein annotation. *Nucleic Acids Res*. Jan 4 2016;44(D1):D457-62. doi:10.1093/nar/gkv1070
36. Vincent SD, Dunn NR, Hayashi S, Norris DP, Robertson EJ. Cell fate decisions within the mouse organizer are governed by graded Nodal signals. *Genes Dev*. Jul 2003;17(13):1646-62. doi:10.1101/gad.1100503
37. Stronati E, Giraldez S, Huang L, et al. YAP1 regulates the self-organized fate patterning of hESC-derived gastruloids. *Stem Cell Reports*. Jan 03 2022;doi:10.1016/j.stemcr.2021.12.012
38. Takaoka K, Yamamoto M, Hamada H. Origin and role of distal visceral endoderm, a group of cells that determines anterior-posterior polarity of the mouse embryo. *Nat Cell Biol*. May 29 2011;13(7):743-52. doi:10.1038/ncb2251
39. Weberling A, Siriwardena D, Penfold C, Christodoulou N, Borovjak TE, Zernicka-Goetz M. Primitive to visceral endoderm maturation is essential for mouse epiblast survival beyond implantation. *iScience*. Jan 17 2025;28(1):111671. doi:10.1016/j.isci.2024.111671
40. Torres-Padilla ME, Richardson L, Kolasinska P, Meilhac SM, Luetke-Eversloh MV, Zernicka-Goetz M. The anterior visceral endoderm of the mouse embryo is established from both preimplantation precursor cells and by de novo gene expression after implantation. *Dev Biol*. Sep 1 2007;309(1):97-112. doi:10.1016/j.ydbio.2007.06.020
41. Handford CE, Junyent S, Jorgensen V, Zernicka-Goetz M. Topical section: embryonic models (2023) for Current Opinion in Genetics & Development. *Curr Opin Genet Dev*. Feb 2024;84:102134. doi:10.1016/j.gde.2023.102134
42. Rossant J, Tam PPL. Early human embryonic development: Blastocyst formation to gastrulation. *Dev Cell*. 01 24 2022;57(2):152-165. doi:10.1016/j.devcel.2021.12.022
43. Ghimire S, Mantziou V, Moris N, Arias AM. Human gastrulation: The embryo and its models. *Dev Biol*. Jan 2021;doi:10.1016/j.ydbio.2021.01.006

44. Brons IG, Smithers LE, Trotter MW, et al. Derivation of pluripotent epiblast stem cells from mammalian embryos. *Nature*. Jul 12 2007;448(7150):191-5. doi:10.1038/nature05950
45. Warriar S, Van der Jeught M, Duggal G, et al. Direct comparison of distinct naive pluripotent states in human embryonic stem cells. *Nat Commun*. Apr 21 2017;8:15055. doi:10.1038/ncomms15055
46. Hagenbeek TJ, Zbieg JR, Hafner M, et al. An allosteric pan-TEAD inhibitor blocks oncogenic YAP/TAZ signaling and overcomes KRAS G12C inhibitor resistance. *Nat Cancer*. Jun 2023;4(6):812-828. doi:10.1038/s43018-023-00577-0
47. Hirate Y, Hirahara S, Inoue K, et al. Polarity-dependent distribution of angiotensin II localizes Hippo signaling in preimplantation embryos. *Curr Biol*. Jul 08 2013;23(13):1181-94. doi:10.1016/j.cub.2013.05.014
48. Blakely WJ, Hatterschide J, White EA. HPV18 E7 inhibits LATS1 kinase and activates YAP1 by degrading PTPN14. *mBio*. Oct 16 2024;15(10):e0181124. doi:10.1128/mbio.01811-24
49. Hermann A, Wu G, Nedvetsky PI, et al. The Hippo pathway component Wwc2 is a key regulator of embryonic development and angiogenesis in mice. *Cell Death Dis*. Jan 22 2021;12(1):117. doi:10.1038/s41419-021-03409-0
50. Garbincius JF, Salik O, Cohen HM, et al. TMEM65 regulates and is required for NCLX-dependent mitochondrial calcium efflux. *Nat Metab*. Apr 2025;7(4):714-729. doi:10.1038/s42255-025-01250-9
51. Pronobis MI, Zheng S, Singh SP, Goldman JA, Poss KD. In vivo proximity labeling identifies cardiomyocyte protein networks during zebrafish heart regeneration. *Elife*. Mar 25 2021;10doi:10.7554/eLife.66079
52. Golchoubian B, Brunner A, Bragulat-Teixidor H, et al. Reticulon-like REEP4 at the inner nuclear membrane promotes nuclear pore complex formation. *J Cell Biol*. Feb 7 2022;221(2)doi:10.1083/jcb.202101049

Dear Conchi,

Thank you for clarifying the concerns regarding the composition of Figure 6E and Figure EV6D, for sending the tiled images of the gastruloids (Nodal and H2B signals) and the modified figure panels.

Thank you also for clarifying that the Western blot in Figure EV3I has been reprobbed with the YAP1 antibody.

I have meanwhile received the report from one of the referees whom I had asked to assess the revised version (referee #3) but not yet from referee #1.

You informed me that you are currently repeating the gastruloid experiments and hope to have them finished within the next weeks. To allow you to correct the mistakes made during image composition, to replace the data in question with the repeats and to ensure a transparent review process on the corrected data, I have decided to make a preliminary decision at this point.

I have recorded the positive report from referee #3 and now invite you to revise your study and update the imaging data and figure panels in question. Once the revised version including the associated raw imaging and metadata is in and given that the concerns have been adequately addressed and resolved, I will complete the review process, contacting referee #1.

I hope you agree with this way forward, which I feel is more streamlined than having partial replacements of figures.

Please do not hesitate to contact me if you have questions or further comments.

Kind regards,

Martina

=====

Referee #3:

The authors have addressed my points satisfactorily. I commend the manuscript for publication.

Dear Dr. Rembold,

I would like to give you an update on the experiments. We were able to reproduce the results using hESC-directed differentiation approaches, where Nodal and BRA markers are consistently upregulated in siQSER1 compared to controls.

However, the micropattern chips have not been cooperating. In our attempts, the cells either failed to attach, and controls did not express expected markers, likely due to inconsistency in the formation of colonies/poor cell attachment. We have successfully used this system extensively in the past and we are unsure of the origin of the problem. At the moment, we have run out of chips, and since they must be imported from France (CYTOO comp.), it will take at least another 6-8 weeks before we can continue.

As an alternative, I would propose replacing the micropattern images with the hESC-directed differentiation staining and quantifications. The biological message on QSER1 function during differentiation would remain the same, only the experimental system would be slightly different. Please have a look at the new images for Figure 6. We can complete the figure and the text and resubmit the updated manuscript before the end of the week.

I am also sharing with you the link for the raw files of the new images for Figure 6: LINK removed

Please let me know if you agree with this change so we can move forward.
Thank you very much,

Best regards,

Conchi

Conchi Estaras, PhD
Assistant Professor
Aging+Cardiovascular Discovery Center (ACDC)
Lewis Katz School of Medicine
Temple University
3500 North Broad St.
950 MERB
Philadelphia, PA 19140

Dear Conchi,

Thank you once more for the submission of your revised manuscript to EMBO reports. I follow up on my e-mail from December 31, 2025 with a formal decision letter.

As you know, referee #1, despite having raised significant concerns, was not available anymore. I have therefore asked referee #2 to assess the revised version and s/he considers all concerns adequately addressed and supports publication.

Before we can proceed with the official acceptance of your study, I need you to address the following editorial points:

- A Disclosure Statement & Competing Interests section is needed after Acknowledgments.
- Regarding the Author Contributions, we now use CRediT to specify the contributions of each author in the journal submission system. Therefore, please remove the Author Contributions from the manuscript file and make sure that the author contributions in our online manuscript tracking system are correct and up-to-date. The information you specified in the system will be automatically retrieved and typeset into the article. You can enter additional information in the free text box provided, if you wish. See also our guide to authors <https://www.embopress.org/page/journal/14693178/authorguide#authorshipguidelines>.
- The references need to be alphabetical, not numerical; et al needs to be used after 10 author names; DOIs should only be used for preprints and datasets that have not been published yet.
- Reference 44, 92, and 126 are preprints. Please cite them as (preprint: NAME et al, YEAR) in the text and add the tag [PREPRINT] in the reference list (at the end of the reference).
- Please add the following funding information in the online submission system. The information in the system and the manuscript text must be congruent: NIH Cancer Center Support Grant P30 CA006927 and NIH Cancer Center Support P30 CA030199.
- Tables EV1-EV4 are datasets so the titles in the online submission system and callouts in the manuscript need to be updated to Dataset EV1-EV4. The source file names are correctly labeled but the legends in each file need correction; the legends are also in the manuscript and need to be removed from there.
- Tables EV5 and EV6 can remain EV tables (Table EV1, Table EV2). Please update this information/nomenclature in the manuscript and also in the Reagents and Tools table and Author Checklist.
- Please add a description of housing and husbandry conditions for mice in the Methods section.
- The Synopsis image needs to be exactly 550 pixels wide x 200-600 pixels high and the resolution could be improved.
- Materials and Methods should be Methods.
- The nomenclature of EV figure legends and individual EV Figure files needs to be changed to be Figure EV1, etc. instead of Expanded View 1, etc.
- Some of the figure panels, in particular those showing quantifications have low resolution: Figure 1F/E/H, Figure 6F/J, EV1E-G, EV2E/G, EV3G/D/H, EV5,B/D/E. Please carefully check all panels showing quantification data and improve the resolution.
- Figure EV6C and EV6F are also of rather low resolution. It also appears as if the images in EV6F had been distorted, if compared to the raw data, i.e., not resized while preserving the x:y ratio and proportion. They look compressed in the x-axis. Please check whether this was the case. The same is true for EV6G.
- Please provide all figures in high-resolution.
- All scale bars must be clearly visible and their size be defined in the legend, not in the figure/image. In Figure EV3F-H, as an example, the scale bars can hardly be seen and there is low-resolution text associated with the bar. Please carefully check all imaging data and their scale bars.
- Please add MW markers for all Western blots.
- Please provide the specific URLs for GSE280804, GSE280805 datasets in the data availability statement, i.e., links that resolve directly to the datasets.
- Please make sure that the link to the raw data on BioStudies is up-to-date.

- Please address the following comments from our data editors in the figure legends:

- a) Please define the annotated p values ****/***/**/* as well as provide the exact p-values for the same in the legend of figure 1E, EV1 E as appropriate.
- b) Please provide the exact p values in the legends of figures 1F; 2C, D; 5D, 6A, C, D, F, G, I, J; EV1 F; EV2 E, G; EV3 D, G, H; EV5 D, E; EV6 E (unless $p < 0.0001$).
- c) Please indicate the statistical test used for data analysis in the legends of figures 1E, 3D; EV1 E
- d) Please define the box plots in terms of minima, maxima, centre, bounds of box and whiskers, and percentile in the legend of figure EV1 G
- e) Please note that information related to n is missing in the legends of figures 1E, 3D, 6I; EV1 E, EV1 G, EV3 I. In addition to specifying "n" we also need information on whether these are technical or biological replicates.
- f) Please define the error bars in the legends of figures 1H; EV3 I

- We perform routine data integrity checks on figures and quantification. In this context I noticed that the values for GAPDH in the source data file for Figure EV6D are the same for the CTGF and CYRR61 quantification. Could you please clarify whether these data are from the same experiment and the use of the same control values is thus justified? Please see the attached color-coded file.

- Please write the abstract in present tense. I copy my suggestion with minor modifications below my signature.

With kind regards,

Martina

=====

Referee #2:

The revised version of the manuscript entitled "YAP1 and QSER are key modulators of embryonic signaling pathways in the mammalian epiblast" by Abraham et al. contains new data and a rewritten text that have further strengthened the conclusions of the study. In particular, the authors have now included an analysis of extraembryonic expression changes in embryos with Yap1 mutant epiblast showing that the AVE appears unaffected but changes are observed in other extraembryonic tissues. Although, an analysis of phospho-SMADs was inconclusive due to limited material from embryos, YAP1 binding to a Nodal enhancer and regulation of Nodal target genes support the conclusion of a role of YAP1 in signalling regulation in the epiblast. The limited TEAD peptides in the YAP1 interaction proteomics data seems to have a technical explanation and is consistent with earlier work. In conclusion, my earlier concerns have been addressed in a satisfactory manner in the revised version. The epiblast specific function of YAP1 and the new data on QSER1 will be of interest to researchers studying early mammalian development, cell signaling, and stem cell biology.

Minor points

- a) On page 7, line 4 "with life past E8.5" or "beyond"; line 12 mid-posterior streak could be defined better
- b) On page 8, line 5 "clonal inducible ESC line" (also consider including the H1 cell line name); line 29: WAPL appears to be the current gene name.
- c) On page 10, line4 "role of QSER in protecting from DNA methylation"; it is curious that both YAP1 and QSER would have implications in activation and preventing repression, respectively. Yet the authors identify a repressive function. This seemingly opposite roles could be pointed out in a brief sentence in the discussion.
- d) Figure 3: The revised version explains the selection of QSER for follow up studies by its predicted strong interaction with YPA1 which is supported by panel E. Could a brief statement included for the two top interactors with $< -1.2 \times 10^6$ kcal/mol or twice the binding energy of QSER1 are expected and support the specificity of the proteomics analysis. Alternative, CALD1 and GOLGA5 could just be technical predictions by the software. A brief clarification could serve to orient the reader.

=====

Abstract

YAP1 signaling is essential for development but its specific roles in early embryogenesis remain poorly understood. To shed light on this, we analyzed YAP1's role in regulating the pluripotency of the mammalian epiblast, using scRNAseq approaches. Conditional deletion of *Yap1* in the mouse epiblast (*Sox2-Cre*) alters the expression of signaling genes, including *Nodal*, *Wnt3*, and *Fgf8*. Accordingly, *Yap1* loss leads to enhanced differentiation of the epiblast toward primitive streak lineages, as evidenced by the upregulation of *T/Brachyury* and *Eomes* genes. A proximity labeling assay in human pluripotent stem cells, followed by biochemical assays and molecular modeling predictions, reveals that YAP1 cooperates with QSER1 protein to regulate lineage genes. Our analysis shows that YAP1:TEAD4 enhancers recruit QSER1 to prevent RNA Polymerase II recruitment. QSER1 depletion, similar to YAP1, increases NODAL gene expression and leads to hyperactive NODAL signaling during human embryonic stem cells differentiation. Overall, our findings define a role of YAP1 in the epiblast *in vivo* and uncover an interplay with QSER1 controlling the activity of developmental signaling pathways in pluripotent cells.

Before we can proceed with the official acceptance of your study, I need you to address the following editorial points:

- A Disclosure Statement & Competing Interests section is needed after Acknowledgments.

Added.

- Regarding the Author Contributions, we now use CRediT to specify the contributions of each author in the journal submission system. Therefore, please remove the Author Contributions from the manuscript file and make sure that the author contributions in our online manuscript tracking system are correct and up-to-date. The information you specified in the system will be automatically retrieved and typeset into the article. You can enter additional information in the free text box provided, if you wish. See also our guide to authors <https://www.embopress.org/page/journal/14693178/authorguide#authorshipguidelines>.

Done.

- The references need to be alphabetical, not numerical; et al needs to be used after 10 author names; DOIs should only be used for preprints and datasets that have not been published yet.

Done.

- Reference 44, 92, and 126 are preprints. Please cite them as (preprint: NAME et al, YEAR) in the text and add the tag [PREPRINT] in the reference list (at the end of the reference).

Done.

- Please add the following funding information in the online submission system. The information in the system and the manuscript text must be congruent: NIH Cancer Center Support Grant P30 CA006927 and NIH Cancer Center Support P30 CA030199.

Done.

- Tables EV1-EV4 are datasets so the titles in the online submission system and callouts in the manuscript need to be updated to Dataset EV1-EV4. The source file names are correctly labeled but the legends in each file need correction; the legends are also in the manuscript and need to be removed from there.

The names have been changed to datasets, and the file names have been changed. The legends have also been removed from the manuscript.

- Tables EV5 and EV6 can remain EV tables (Table EV1, Table EV2). Please update this information/nomenclature in the manuscript and also in the Reagents and Tools table and Author Checklist.

The nomenclature in the manuscript and file have change. Also edited in the reagents table and author checklist.

- Please add a description of housing and husbandry conditions for mice in the Methods section.

A description was added.

- The Synopsis image needs to be exactly 550 pixels wide x 200-600 pixels high and the resolution could be improved.

Changed and uploaded.

- Materials and Methods should be Methods.

Changed.

- The nomenclature of EV figure legends and individual EV Figure files needs to be changed to be Figure EV1, etc. instead of Expanded View 1, etc.

Changed.

- Some of the figure panels, in particular those showing quantifications have low resolution: Figure 1F/E/H, Figure 6F/J, EV1E-G, EV2E/G, EV3G/D/H, EV5,B/D/E. Please carefully check all panels showing quantification data and improve the resolution.

Changed.

- Figure EV6C and EV6F are also of rather low resolution. It also appears as if the images in EV6F had been distorted, if compared to the raw data, i.e., not resized while preserving the x:y ratio and proportion. They look compressed in the x-axis. Please check whether this was the case. The same is true for EV6G.

Changed.

- Please provide all figures in high-resolution.

Changed.

- All scale bars must be clearly visible and their size be defined in the legend, not in the figure/image. In Figure EV3F-H, as an example, the scale bars can hardly be seen and there is low-resolution text associated with the bar. Please carefully check all imaging data and their scale bars.

Changed.

- Please add MW markers for all Western blots.

Changed.

- Please provide the specific URLs for GSE280804, GSE280805 datasets in the data availability statement, i.e., links that resolve directly to the datasets.

Added into the manuscript.

- Please make sure that the link to the raw data on BioStudies is up-to-date.

Done

- Please address the following comments from our data editors in the figure legends:

a) Please define the annotated p values ****/***/**/* as well as provide the exact p-values for the same in the legend of figure 1E, EV1 E as appropriate. Done.

b) Please provide the exact p values in the legends of figures 1F; 2C, D; 5D, 6A, C, D, F, G, I, J; EV1 F; EV2 E, G; EV3 D, G, H; EV5 D, E; EV6 E (unless $p < 0.0001$). Done

c) Please indicate the statistical test used for data analysis in the legends of figures 1E, 3D; EV1 E Done.

d) Please define the box plots in terms of minima, maxima, centre, bounds of box and whiskers, and percentile in the legend of figure EV1 G Done.

e) Please note that information related to n is missing in the legends of figures 1E, 3D, 6I; EV1 E, EV1 G, EV3 I. In addition to specifying "n" we also need information on whether these are technical or biological replicates. Done.

f) Please define the error bars in the legends of figures 1H; EV3 I Done.

- We perform routine data integrity checks on figures and quantification. In this context I noticed that the values for GAPDH in the source data file for Figure EV6D are the same for the CTGF and CYRR61 quantification. Could you please clarify whether these data are from the same experiment and the use of the same control values is thus justified? Please see the attached color-coded file.

Yes, these data are from the same experiment, and the use of the same housekeeping gene was intended for that purpose.

- Please write the abstract in present tense. I copy my suggestion with minor modifications below my signature.

Have incorporated the changes.

Reviewer 2

Minor points

a) On page 7, line 4 "with life past E8.5" or "beyond"; line 12 mid-posterior streak could be defined better

Page 7, line 12. Typo corrected: "with life beyond E8.5"

Page 7, line 20. We removed "mid-posterior" to avoid confusion. Now it reads "other PS populations, including presomitic and somitic mesoderm..."

b) On page 8, line 5 "clonal inducible ESC line" (also consider including the H1 cell line name); line 29: WAPL appears to be the current gene name.

Page 8, line 15. Now it reads: "we created an H1 inducible hESC clonal line..."

Page 9, line 11. Typo corrected: WAPL.

c) On page 10, line 4 "role of QSER in protecting from DNA methylation"; it is curious that both YAP1 and QSER would have implications in activation and preventing repression, respectively. Yet the authors identify a repressive function. This seemingly opposite roles could be pointed out in a brief sentence in the discussion.

Agreed. We clarify this in the revised discussion. Discussion; **page 18; lines 30-31, page 19; line 1-10.**

d) Figure 3: The revised version explains the selection of QSER for follow up studies by its predicted strong interaction with YPA1 which is supported by panel E. Could a brief statement included for the two top interactors with $< -1.2 \times 10^6$ kcal/mol or twice the binding energy of QSER1 are expected and support the specificity of the proteomics analysis. Alternative, CALD1 and GOLGA5 could just be technical predictions by the software. A brief clarification could serve to orient the reader.

Agreed. Sentence added on **page 10, lines 6-10.**

Dr. Conchi Estarás
Temple University
Department of Cardiovascular Sciences, Aging + Cardiovascular Discovery Center, Temple University, Lewis Katz School of
Medicine
Philadelphia
United States

Dear Conchi,

Thank you for sending the further revised manuscript file and revised Figure EV3, which I uploaded for you.

I am now very pleased to accept your manuscript for publication in the next available issue of EMBO reports. Thank you for your contribution to our journal.

You may qualify for financial assistance for your publication charges - either via a Springer Nature fully open access agreement or an EMBO initiative. Check your eligibility: <https://link.springer.com/journal/44319/how-to-publish-with-us>

Kind regards,

Martina

>>> Please note that it is EMBO Reports policy for the transcript of the editorial process (containing referee reports and your response letter) to be published as an online supplement to each paper. If you do NOT want this, you will need to inform the Editorial Office via email immediately. More information is available here: <https://link.springer.com/partners/embo-press/editorial-policies#Peer%20review>